# From Intents to Actions: Agentic AI in Autonomous Networks

## Abstract

Telecommunication networks are increasingly expected to operate autonomously while supporting heterogeneous services with diverse and often conflicting *intents*—that is, performance objectives, constraints, and requirements specific to each service. However, transforming high-level intents—such as ultra-low latency, high throughput, or energy efficiency—into concrete control actions (i.e., low-level actuator commands) remains beyond the capability of existing heuristic approaches. This work introduces an Agentic AI system for intent-driven autonomous networks, structured around three specialized agents. A supervisory *interpreter agent*, powered by language models, performs both lexical parsing of intents into executable optimization templates and cognitive refinement based on feedback, constraint feasibility, and evolving network conditions. An *optimizer agent* converts these templates into tractable optimization problems, analyzes trade-offs, and derives preferences across objectives. Lastly, a preference-driven *controller agent*, based on multi-objective reinforcement learning, leverages these preferences to operate near the Pareto frontier of network performance that best satisfies the original intent. Collectively, these agents enable networks to autonomously interpret, reason over, adapt to, and act upon diverse intents and network conditions in a scalable manner.

## 1 Introduction

Radio access networks (RANs) are large-scale, real-time distributed systems that must operate reliably in highly dynamic and uncertain radio environments, while serving a broad range of connectivity services and applications. Currently, these systems rely heavily on manual intervention for configuration optimization and functional fine-tuning. This dependence on human expertise limits scalability, slows adaptation to environmental changes, and increases operational costs.

The next generation of communication networks is expected to address these limitations by becoming increasingly autonomous. This evolution—already underway in 5G-Advanced through standardized intent management frameworks, e.g., 3GPP (2025c) and TMForum (2021)—envisions self-configuring, self-optimizing, and self-healing systems guided by high-level *network intents*. Intents specify performance objectives, requirements, and constraints for a connectivity service or management workflow 3GPP (2025c), allowing operators to express *what* the network should achieve rather than *how*. For example, an operator may specify a goal as *"maximize user coverage while minimizing energy consumption,"* leaving the network to autonomously determine the appropriate actions, such as antenna tilt adjustments to improve coverage or carrier deactivation to save energy. In this context, intents act as directives, while the network abstracts away the implementation details, much like a compiler translates high-level code into machine-executable instructions.

Converting intents into network actions is fundamentally a problem of planning and reasoning across multiple abstraction layers—from natural-language specifications to optimization formulations, and ultimately to control policies executed at the RAN. These requirements exceed the capabilities of current heuristic and rule-based approaches. Bridging this gap calls for a new class of artificial intelligence (AI) systems that move beyond perception and prediction, linking abstract objectives with dynamic decision-making through iterative reasoning and planning.

Agentic AI has recently emerged as a promising paradigm for building autonomous, goal-driven systems capable of interpreting objectives, planning multi-step actions, and adapting to dynamic envi-

ronments with minimal human oversight. Unlike traditional AI approaches based on fixed heuristics or monolithic models, Agentic AI structures intelligence into specialized agents that interact and co-operate through well-defined workflows Sapkota et al. (2025). Central to this paradigm are large-scale generative models—particularly large language models (LLMs)—which enable agents to understand and generate natural language, decompose goals, generalize across tasks, invoke specialized tools, and reason in open-ended contexts Liu et al. (2024). As such, Agentic AI offers a compelling architectural foundation for autonomous and intent-driven network management and optimization.

This paper takes a step toward realizing this vision by introducing an Agentic AI system comprising an **interpreter**, an **optimizer**, and a **controller**. Our contributions are:

1. **Cognitive intent processing.** The interpreter is a supervisory cognitive agent with two core functions: converting high-level intents into structured templates and recursively refining them on a slow timescale by reasoning over network observations and feedback on intent fulfillment. To meet RAN compute and memory constraints, we adopt a dual-SLM architecture that separates intent translation and in-context reasoning among two small language models (SLMs).

2. **Preference optimization.** The optimizer agent transforms optimization template models (OTMs) into constrained optimization problems over a preference space, performs preference planning via Bayesian optimization to dynamically adapt preferences to network conditions, and steers the controller policy to satisfy the service intents expressed by the OTM.

3. **Multi-objective control.** The controller leverages multi-objective reinforcement learning (MORL) to realize adaptive policies that operate near the Pareto front of network performance. A central technical contribution is distributed envelope Q-learning (D-EQL), a scalable distributed variant of envelope Q-learning (EQL) Yang et al. (2019) that: (i) decouples learner–actors with sharded prioritized replay for high-throughput training; (ii) distributes the exploration of the preference simplex across actors while learning a single preference-conditioned network; (iii) uses envelope updates with vector TD targets plus a cosine-stability loss; and (iv) refreshes priorities with hindsight preference relabeling. Together, these extensions improve scalability, accuracy and exploration over established MORL art Yang et al. (2019); Basaklar et al. (2023).

4. **Proof of concept.** We showcase the agentic system through an intent-aware radio resource management (RRM) use case combining interpreter and optimizer agents with a novel MORL-based link adaptation (LA), and adapt its policy on the fly to diverse connectivity service goals. Our approach outperforms traditional reinforcement learning (RL)—which cannot adapt a single policy across goals—and exceeds the state-of-the-art LA baseline of 5G/5G-A systems.

Results from high-fidelity system-level simulations of a 5G-compliant network suggest that Agentic AI can transform high-level human intents into self-optimizing control mechanisms for next-generation networks, thereby paving the way toward scalable network autonomy.

## 2 RELATED WORK

**Agentic AI:**  Agentic AI is an emerging paradigm that structures intelligence as a modular network of specialized agents collaborating to achieve complex, high-level goals (Hughes et al., 2025). Recent surveys highlight recurring design patterns and challenges related to reliability and evaluation (Guo et al., 2024; Li et al., 2024). A central mechanism is *goal decomposition*, whereby broad objectives are divided into subtasks handled by agents with distinct functions. Prior work has demonstrated that agents can integrate reasoning and action in recursive loops (Yao et al., 2023), improve performance through reflective memory (Shinn et al., 2023), and operate collectively via structured communication (Wu et al., 2024). To coordinate distributed intelligence, orchestration layers or meta-agents assign roles, manage life cycles and task dependencies, and resolve conflicts using centralized or decentralized mechanisms (Qian et al., 2024). Furthermore, persistent goals and memory enable adaptation over long time horizons (Wang et al., 2024; Agashe et al., 2025). Domain-specific systems, such as MAGIS (Tao et al., 2024), illustrate how these principles scale to collaborative workflows.

**Bayesian optimization:**  Zhan & Xing (2020) reviews the evolution of expected improvement (EI) as an acquisition function for surrogate-based optimization, detailing its extensions to parallel, multi-objective, constrained, noisy, multi-fidelity, and high-dimensional settings, analyzing their theoretical properties, and highlighting future research directions. Zhao et al. (2024) shows that the performance

of high-dimensional Bayesian optimization is strongly limited by poor random initialization of acquisition function maximizers and proposes AIBO, a simple framework that uses past evaluations and heuristic search to generate better starting points, significantly boosting optimization efficiency.

**Multi-objective reinforcement learning:** MORL addresses control problems in which optimality is defined by a Pareto front of policies, each capturing different trade-offs among multiple objectives.

Early approaches to multi-objective optimization (Kim & de Weck, 2005; Konak et al., 2006; Yoon et al., 2009) reduced the problem to scalar optimization—typically via utility functions with fixed weights across objectives—followed by standard RL. These methods are tied to a single preference setting and cannot adapt when goals or constraints change (Liu et al., 2015), thereby necessitating retraining. To improve generality, subsequent work sought to approximate the entire Pareto front by learning multiple optimal policies over the preference space (Natarajan & Tadepalli, 2005; Barrett & Narayanan, 2008; Mossalam et al., 2016). However, training a separate policy for each preference combination quickly becomes computationally infeasible in large domains.

A more scalable approach is to learn a single universal policy conditioned on preferences (Yang et al., 2019; Xu et al., 2020; Abdolmaleki et al., 2020), enabling adaptation across tasks without retraining. For instance, Yang et al. (2019) proposed envelope Q-learning, which generalizes the Bellman equation to optimize the convex envelope of multi-objective Q-values under linear preferences using deep networks. Extensions such as those in Basaklar et al. (2023) introduced parallelization to improve sample efficiency and Pareto approximation. Nonetheless, efficiently exploring the preference space and learning universal MORL policies remain open challenges (Hayes et al., 2022).

**Agentic AI in Communication Systems:** Intent-based management is already part of modern 5G-Advanced systems 3GPP (2025c), and its extension toward 6G is strongly supported in current standardization efforts 3GPP (2025g). Concurrently, academic and industrial interest in Agentic AI is rapidly growing, positioning it as a key enabler of next-generation autonomous networks, particularly for intent-driven operations Bimo et al. (2025); ZTE (2025); Intel & NEC (2025). Recent work on agent-based and LLM-guided control frameworks for network optimization and service management Qayyum et al. (2025); Jolicoeur-Martineau (2025); Bimo et al. (2025) highlights a shift toward systems capable of reasoning, adaptation, and collaboration. This trajectory is reflected across 3GPP, Open RAN, and TM Forum. For example, 3GPP TR 22.870 3GPP (2025a) identifies AI-agent–enabled service coordination, LLM-assisted interactions, and agent-supported UE–network cooperation as 6G use cases, while IETF (2025) defines protocols for AI-agent communication. Furthermore, the 3GPP SA5 workgroup has identified *intent-driven agentic autonomous management* as a priority areas for 6G 3GPP (2025i;h) while SA2 is examining agentic mechanisms for the 6G core network 3GPP (2025f). Together, these developments indicate that agentic and intent-based paradigms are increasingly viewed as foundational elements of future 6G architectures.

**Differentiation from Prior Agentic AI Work:** Existing Agentic AI systems have largely been applied to reasoning, planning, and tool use, where control loops operate over long timescales in relatively stable environments. By contrast, we integrate agentic AI into the fast control loops of RRM, where sub-millisecond decisions must adapt to fading channels, mobility, and heterogeneous service requirements. To our knowledge, this is among the first applications of Agentic AI in highly dynamic, stochastic environments, extending its reach to performance-critical autonomous networks.

We demonstrate the workflow with an end-to-end, cognitively guided intent-aware RRM design for supporting different connectivity services, where control policies adapted by reasoning over individual service goals and network observations are then executed in time-varying, frequency-selective environments to meet the goals. Our results show superior performance compared to traditional RL and the state-of-the-art LA algorithm adopted in 5G/5G-A systems.

## 3 AGENTIC AI SYSTEM FOR RAN CONTROL

At its core, the proposed *Agentic AI system* comprises three specialized agents—interpreter, optimizer, and controller—whose interactions form an *agentic workflow* consisting of two loops: an *intent management loop*, executed by the interpreter–optimizer pair, and an *intent fulfillment loop*, executed by the optimizer–controller pair. Each loop operates on a distinct timescale, forming a two-timescale control architecture analogous to Kahneman's dual-process theory (Kahneman, 2011), with a slower, deliberative outer System 2 and a faster, reactive inner System 1.

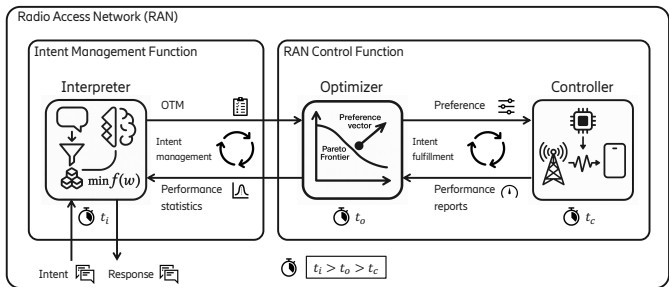

Figure 1: Agentic AI system for intent and resource management in autonomous networks.

The *interpreter* is a supervisory cognitive agent that converts high-level intents into structured templates and adaptively refines them on a slow timescale using network states and fulfillment feedback. The *optimizer* recursively plans and adjusts the downstream controller configurations to satisfy the intent, aggregating controller feedback into slower-timescale statistical summaries returned to the interpreter. The *controller* executes real-time decision-making, collects observations, and provides periodic performance reports to the optimizer.

This triadic workflow provides a blueprint for a broader Agentic AI system for autonomous management and optimization of communication networks. Its realization, however, requires a twofold extension. *Horizontally*, the interpreter may coordinate with multiple optimizer–controller pairs supporting different RAN functions within a single architectural layer. *Vertically*, the workflow can be embedded across different layers of the RAN protocol stack, whose operational timescales range from slow (for network management at higher layers) to very fast (for RRM at lower layers).

### 3.1 Timescales Separation

The workflow separates responsibilities across three timescales. The controller handles real-time decisions and thereby establishes the system's reference timescale $t_c$. Because this agent replaces an existing RAN control function, it inherits that function's native latency budget, which may range from sub-millisecond operation for RRM functions (e.g., link adaptation) to minutes or hours for network optimization tasks (e.g., cell shaping). The optimizer adjusts the controller's policy at a deliberately slower timescale $t_o$, spanning hundreds of milliseconds to seconds for fast RRM functions and up to hours for RAN management functions, ensuring that its decisions do not interfere with the primary control loop. The interpreter operates on the slowest supervisory cadence $t_i$, which spans seconds to minutes for RRM supervision and up to hours for RAN-wide management. At this timescale, the interpreter evaluates intent feasibility, reasons over observed key performance indicator (KPI) deviations, and generates refined intents without imposing timing constraints on downstream agents.

Decoupling long-term reasoning and intermediate adaptation from real-time control ensures that (a) the interpreter supervisory role is non-latency-critical; (b) latency-critical operations are confined to the controller—for any RAN control function involved; and (c) the fast control loop remains stable.

## 4 Language-Guided Intent Management

### 4.1 Interpreter Agent

The interpreter is a language-guided supervisory agent aligned with the scope of an intent management function (IMF) (TMForum, 2024). It performs two complementary functions: (a) *transforming intents* into structured OTMs, and (b) *cognitive reasoning* for recursive intent adaptation.

The interpreter agent must integrate domain awareness, intent stabilization, and adherence to the computational and memory constraints of the RAN system. Domain awareness includes understanding which control agents operate within each sub-domain, their capabilities, parameters, and timescales, as well as the KPIs they influence. This knowledge enables the interpreter to produce feasible OTM formulations for a given intent, route each intent to the appropriate RAN control agent, and ensure intent stabilization by reasoning over system observations, optimizer feedback, and network dynamics to perform safe, explainable OTM refinements when required.

| Model | Schema accuracy | OTM accuracy | | |
| --- | --- | --- | --- | --- |
| | | Objectives | Constraints | Overall |
| Qwen-2.5-7B-Instruct (Before fine-tuning) | 100.0% | 45.00% | 21.50% | 11.30% |
| Qwen-2.5-7B-Instruct (After fine-tuning) | 100.0% | **100.0%** | **98.00%** | **98.00%** |

Table 1: Schema and OTM accuracy for interpreters using the Qwen-2.5-7B-Instruct model.

Meeting these requirements within current 5G/5G-A RAN hardware necessitates a design that is both computationally efficient and functionally modular. Deploying a single large general-purpose LLM is impractical due to compute and memory constraints in current RAN platforms, and integrating dedicated accelerators is neither scalable nor cost-effective. To address this, we adopt a dual-SLM architecture that separates the interpreter's two core functions—intent translation and cognitive reasoning—across two lightweight, complementary SLMs, as detailed in Appendix A.

**Intent translation.** This module is the workflow entry point. It interprets the intent, decomposes it into sub-intents, selects the appropriate downstream control agent, and initiates the intent-fulfillment loop. A fine-tuned SLM renders the intent as a structured, schema-compliant OTM by disambiguating objectives, constraints, requirements, and metadata. This step extends beyond lexical parsing: the model must map high-level intents into optimization structures grounded in domain knowledge. Using a fine-tuned SLM ensures low-complexity generation of machine-readable OTMs that reflect RAN semantics and remain robust to linguistic variability. Appendix B discusses the generality of the OTM schema, while Appendix C outlines the fine-tuning of a Qwen-2.5-7B-Instruct model Qwen et al. (2025), which achieves the high schema validity and OTM accuracy shown in Table 1.

**Cognitive reasoning and adaptation.** Complementing the translator, a lightweight general-purpose SLM performs supervisory reasoning via in-context learning. It evaluates feasibility, diagnoses constraint violations, and refines OTMs when strict requirements cannot be met, proposing alternative trade-offs or adapting objectives to evolving network conditions. Intent stabilization is achieved through structured monitoring, advisory evaluation, and guarded execution (see Appendix A). This supervisory closed-loop reasoning extends beyond static templates or rule-based logic and is essential for autonomous, intent-driven, network management under real-world network dynamics.

This division of labor preserves contextual knowledge and ensures adaptability for intent handling, while remaining compatible with practical constraints of contemporary RAN deployments. The dual-SLM interpreter—built from small-scale models—and the infrequent, non-latency-critical nature of SLM inference within the agents' timescale separation allow the system to maintain low compute and energy overhead. As a result, the overall design is feasible on current 5G/5G-Advanced hardware.

### 4.2 OPTIMIZER AGENT

The optimizer agent performs three key tasks: (i) decoding the OTM received from the interpreter, (ii) recursively solving the associated optimization problem to align the controller's policy with the intent, and (iii) coordinating the two feedback loops within the workflow. Upon receiving an OTM, the optimizer formulates a constrained optimization problem aligned with the specified intent, such as

$$
\begin{aligned}
\underset{\boldsymbol{\omega} \in \Omega}{\text{minimize}} \quad & f(\boldsymbol{\omega}) \\
\text{subject to} \quad & g_i(\boldsymbol{\omega}) \leq b_i, \quad i = 1, \ldots, p,
\end{aligned}
\tag{1}
$$

where $f(\boldsymbol{\omega})$ quantifies the system performance (e.g., energy, latency, throughput), and the decision variable $\boldsymbol{\omega}$ belongs to a feasible set $\Omega \subseteq \mathbb{R}^m$. The inequality constraints $g_i(\boldsymbol{\omega}) \leq b_i$ capture operational limitations—e.g., bandwidth, latency, or power—or service requirements. Since both objective and constraints are often non-convex, the solution landscape may contain multiple local optima, making the identification of feasible or optimal solutions challenging.

The decision variables $\boldsymbol{\omega}$ link the optimizer to the controller by representing hyperparameters that tune the controller's policy. In our framework, the controller follows a MORL approach (Section 5), so $\boldsymbol{\omega}$ corresponds directly to the preference weights in its multi-dimensional reward function.

Since the explicit forms of $f$ and $g_i$ are unknown and their evaluations are computationally expensive, the optimizer employs Bayesian optimization (BO), leveraging surrogate models trained on RAN performance data (e.g., throughput, spectral efficiency, block error rate (BLER)) relevant to the intent.

These models guide the exploration of preference weights $\boldsymbol{\omega}$ (i.e., decision variables), which steer the controller's actions. Additional details of the BO design are provided in Appendix D.

### 4.2.1 PAX-BO: PREFERENCE-ALIGNED EXPLORATION BAYESIAN OPTIMIZATION

We next address the preference-based constrained BO problem (1) in the multi-service case, where $S$ connectivity services must be jointly optimized under $p$ constraints that capture requirements such as data rate, latency, and reliability. The optimization problem (1) becomes

$$
\begin{aligned}
\underset{\mathbf{W} \in \Omega^S}{\text{minimize}} \quad & f(\mathbf{W}) \\
\text{subject to} \quad & g_i(\mathbf{W}) \leq b_i, \quad i = 1, \ldots, p,
\end{aligned}
\tag{2}
$$

where $\mathbf{W} = [\boldsymbol{\omega}^{(1)}, \ldots, \boldsymbol{\omega}^{(S)}]$ collects the service-specific preference vectors $\boldsymbol{\omega}^{(s)} \in \Omega$ ($\Omega = \Delta^{m-1}$) on the probability simplex. The objective $f(\mathbf{W})$ quantifies system-wide performance, while the constraints $g_i(\mathbf{W}) \leq b_i$ enforce joint service requirements. Problem (2) reduces to (1) when $S = 1$.

**PAX-BO**, shown in Algorithm 2, solves Problem (2) by optimizing preference vectors on the simplex through BO in an unconstrained internal space. Let $U = [u^{(1)}, \ldots, u^{(S)}] \in \mathbb{R}^{m \times S}$ and $\bar{u} = \text{vec}(U)$. Each service $s$ has a projected simplex weight $\boldsymbol{\omega}^{(s)} = \Pi_\Delta(u^{(s)}) \in \Delta^{m-1}$, and $\mathbf{W}(U) = [\boldsymbol{\omega}^{(1)}, \ldots, \boldsymbol{\omega}^{(S)}] \in (\Delta^{m-1})^S$. At each iteration, we fit surrogate models that approximate the system objective and constraints as $\mathcal{F}(\bar{u}) \approx f(\mathbf{W}(U))$ and $\mathcal{G}_i(\bar{u}) \approx g_i(\mathbf{W}(U))$, and build a constraint-aware acquisition $\alpha(\bar{u})$ (e.g., Log-EI times a feasibility probability).

A *trust region (TR)*—an $\ell_\infty$ box with center $s_c$ and radius $L \in [L_{\min}, L_{\max}]$—constrains local exploration. At each iteration, the acquisition function is maximized within the TR, and the solution is projected back onto the simplex:

$$
\bar{u}_t = \arg \max_{\|\bar{v} - s_c\|_\infty \leq L} \alpha(\bar{v}), \quad U_t = \text{mat}(\bar{u}_t), \quad \mathbf{W}_t = \Pi_\Delta(U_t).
$$

After evaluating $o_t = f(\mathbf{W}_{t-1})$ and $c_t^{(i)} = g_i(\mathbf{W}_{t-1})$, we declare success if $c_t^{(i)} \leq 0$ for all $i$ and $o_t \geq f_{t-1}^\star + \epsilon$, with $\epsilon \ll 1$. On success, we set $f_t^\star \leftarrow o_t$, $s_c \leftarrow \bar{u}_{t-1}$, and expand $L$ after $s_{\text{th}}$ consecutive successes; otherwise, $L$ is shrunk after $f_{\text{th}}$ failures, clamped to $[L_{\min}, L_{\max}]$. If the TR stalls at $L_{\min}$ for $w$ rounds, a *reset* is triggered: $n$ candidates are sampled from $(\Delta^{m-1})^S$, scored by (acquisition)$\times$(feasibility)$\times$(novelty), and the best candidate reinitializes $s_c$ with $L \leftarrow L_0$.

Overall, PAX-BO is a lift-and-project BO method with TR safeguards and reset mechanisms, tailored to simplex-valued preferences that jointly influence a constrained system objective.

## 5 PREFERENCE-GUIDED INTENT FULFILLMENT

The optimizer and controller agents operate in a closed loop to achieve intent fulfillment. The optimizer recursively adapts the preference vector $\boldsymbol{\omega}$ based on performance feedback from the controller. The optimal (or near-optimal) vector $\boldsymbol{\omega}^\star$, obtained by solving (1), is then passed to the controller, which aligns network actions with the original intent.

### 5.1 CONTROLLER AGENT

The controller implements a policy trained via D-EQL, a distributed extension of EQL (Yang et al., 2019). D-EQL learns a single policy/value network conditioned on a linear preference vector $\boldsymbol{\omega} \in \Omega$ (the probability simplex) and scales exploration through a learner–actor architecture with prioritized replay (cf. Horgan et al. (2018)).

During training, actors are assigned to distinct strata of the simplex defined by a simplex-lattice partition. Each actor samples preferences uniformly within its stratum using barycentric sampling, executes an $\varepsilon$-greedy policy with the scalarization

$$
Q_{\boldsymbol{\omega}}(s, a; \boldsymbol{\theta}) = \boldsymbol{\omega}^\top Q(s, a, \boldsymbol{\omega}; \boldsymbol{\theta}),
$$

and initializes replay priorities by drawing an independent preference $\tilde{\boldsymbol{\omega}}$ to compute a scalar temporal-difference error. Transitions and priorities are batched locally and sent to sharded replay buffers.

| Algorithm | Partition | Replay memory | | | | Actor | | CFR1 | Hypervol. |
|---|---|---|---|---|---|---|---|---|---|
| | | Hindsight | Sampling | Update | Sharded | Distrib. | Comm. | Improv. | Improv. |
| Yang et al. (2019) | No | Yes | Prioritized | No | No | No | – | – | – |
| Basaklar et al. (2023) | Yes | Yes | Uniform | No | No | Yes | Synch. | 12.33% | 78.56% |
| D-EQL (ours) | Yes | Yes | Prioritized | Yes | Yes | Yes | Asynch. | **22.10%** | **89.37%** |

Table 2: Comparison of D-EQL with Yang et al. (2019) and Basaklar et al. (2023) in terms of design features and achieved CFR1 performance in Fruit Tree Navigation with depth 7.

The learner assigns strata of the simplex to actors for distributed exploration, retrieves prioritized minibatches from all shards, samples preferences from a Dirichlet distribution, and forms a Cartesian product so that each transition is evaluated under every sampled preference. The learner performs envelope backups by maximizing over actions and supporting preferences, updates parameters using a regression loss with an optional cosine-alignment term, refreshes priorities, and periodically synchronizes the target network. Updated weights are then broadcast to all actors.

The envelope backup is expressed as

$$\boldsymbol{y} = \boldsymbol{r} + \gamma(1-d)\,Q(s', a^\star, \tilde{\boldsymbol{\omega}}^\star; \boldsymbol{\theta}^-), \quad (a^\star, \tilde{\boldsymbol{\omega}}^\star) = \arg\max_{a', \boldsymbol{\omega}' \in \Omega} \boldsymbol{\omega}^\top Q(s', a', \boldsymbol{\omega}'; \boldsymbol{\theta}).$$

Compared with state-of-the-art MORL algorithms such as Yang et al. (2019) and Basaklar et al. (2023), D-EQL introduces (i) a hindsight replay memory with prioritized sampling and priority updates, (ii) partitioned exploration of the preference space across distributed asynchronous actors, and (iii) a sharded replay memory. This architecture improves scalability in environments with large state–action–preference spaces by enabling systematic simplex exploration, dense preference supervision, and high-throughput stable learning. As shown in Table 2, D-EQL achieves a 22.1% performance CFR1 improvement over Yang et al. (2019) and an additional 8% gain over Basaklar et al. (2023) in the Fruit Tree Navigation environment with depth 7, as well as 89.37% hypervolume improvement over Yang et al. (2019) and an extra 6.05% gain over Basaklar et al. (2023). Additional design details and extended comparisons are provided in Appendix F.

## 6 CASE STUDY: AGENTIC RADIO RESOURCE MANAGEMENT

RRM encompasses some of the most demanding and dynamic control functions in RANs, including user scheduling, resource allocation, link adaptation, power control, and beamforming. These mechanisms operate on sub-millisecond timescales and must continuously adapt to the stochastic nature of the wireless channel to maintain reliable and efficient over-the-air communications.

As proof of concept, we apply our Agentic AI system to support differentiated connectivity services using a MORL-based controller agent for LA—a key function that tunes modulation and coding scheme (MCS) parameters to the radio link capacity. The detailed description of the MORL LA controller agent is provided in Appendix G. Here, we note that the reward is a vector $\boldsymbol{r} = [r_1, r_2]^\top \in \mathbb{R}^2$ with two competing components: $r_1$ measures the number of information bits successfully delivered per packet, and $r_2$ captures the time–frequency resources consumed per packet transmission.

In our agentic system, the MORL LA controller agent defines the fastest operational timescale, running on a sub-millisecond cadence. This cadence sets the reference timescale for dimensioning the optimizer and interpreter. The optimizer updates the preference weights of the MORL controller once per second, based on performance reports and observed network conditions. This update rate is fast enough to steer the controller toward MCS selections aligned with the intent goals, yet slow enough not to interfere with the stability of the LA decision loop.

At the same cadence, the optimizer agent provides feedback to the interpreter agent for supervisory monitoring of intent fulfillment. However, the interpreter's cognitive loop is triggered only on an event-driven basis. Upon receiving an alert message from the optimizer, the interpreter leverages its general-purpose SLM to perform cognitive reasoning over KPIs deviations, intent-fulfillment, and evolving network conditions to determine whether the intent must be refined. In our case study, such intervention occurs when changing network conditions render the service requirements temporarily infeasible.

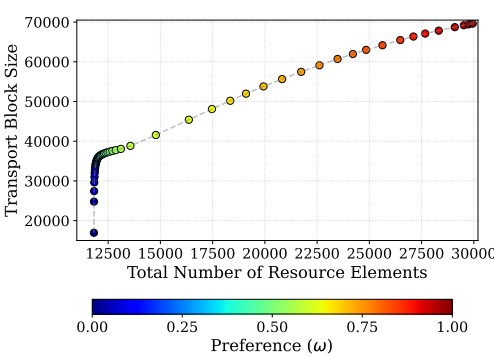

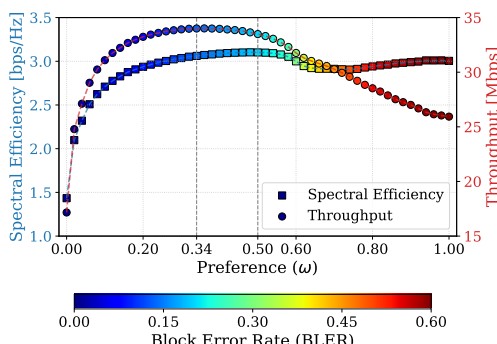

(a) Pareto frontier illustrating the trade-off between transport block size and resource utilization.

(b) Joint characterization of connectivity service KPIs dependence on preference weights $\omega$.

Figure 2: Characterization of preference-guided LA using MORL to satisfy service intents.

# 7 EXPERIMENT

This section evaluates the empirical performance of our Agentic AI system for intent-aware RRM using a 5G-compliant event-driven network simulator. We validate our approach in three steps using a multi-cell setup described in Appendix H: First we validate the MORL controller agent design; secondly, we evaluate the optimizer-controller loop; and lastly we benchmark the overall workflow.

## 7.1 MORL CONTROLLER AGENT FOR LINK ADAPTATION

Figure 2 illustrates how the preference-guided MORL controller for LA steers trade-offs among service KPIs, like spectral efficiency, throughput, and BLER, assuming long communication sessions (e.g., streaming services). Figure 2a shows the Pareto frontier for the two reward components, while Figure 2b maps each point on the frontier to link-level KPIs. When $\omega \approx 0$, the controller selects conservative MCS values, resulting in resource efficient and high-reliable transmissions (with near-zero BLER), but at the cost of low throughput (i.e., due to small transport block sizes (TBSs)) and spectral efficiency. At the other extreme, $\omega \approx 1$ drives aggressive MCS choices that exploit retransmissions to target a spectral efficiency beyond the channel capacity, inducing resource-hungry and unreliable transmissions (with BLER $\approx 60\%$). The best operating points emerge for intermediate preferences, with $\omega \approx 0.34$ maximizing throughput and $\omega \approx 0.5$ maximizing spectral efficiency. Appendix H extends the analysis to examples with multiple connectivity services.

## 7.2 INTENT-FULFILLMENT LOOP VALIDATION

Next, we evaluate *only* the optimizer–controller loop, assuming a single forward interaction with the interpreter to obtain an OTM. That is, when stochastic changes in the RAN environment render the OTM specifications infeasible, the interpreter's cognitive refinement loop is not triggered. While the optimizer–controller pair cannot resolve temporary infeasibility caused by evolving RAN conditions.

We illustrate this by considering an intent that combines two contrasting connectivity services:

> *Maximize cell throughput while serving mobile broadband users on a best-effort basis, and guaranteeing 99.99% reliability for a ultra-reliable traffic.*

This intent reflects quality of service (QoS) requirements for streaming and reliable services. In the agentic workflow, the interpreter constructs an OTM that (a) identifies the two services, (b) defines an overall objective based on their achieved throughput, and (c) formulates a reliability constraint for the reliable service. The optimizer then instantiates an optimization problem to adapt the two vectors, $\omega_{\mathrm{mbb}} = [\omega_{\mathrm{mbb}}, 1 - \omega_{\mathrm{mbb}}]^\top$ and $\omega_{\mathrm{rel}} = [\omega_{\mathrm{rel}}, 1 - \omega_{\mathrm{rel}}]^\top$, each aligned to a service, by maximizing the aggregate throughput $f(\omega_{\mathrm{mbb}}, \omega_{\mathrm{rel}}) = f_{\mathrm{mbb}}(\omega_{\mathrm{mbb}}, \omega_{\mathrm{rel}}) + f_{\mathrm{rel}}(\omega_{\mathrm{mbb}}, \omega_{\mathrm{rel}})$ subject to the reliability constraint $g_{\mathrm{rel}}(\omega_{\mathrm{mbb}}, \omega_{\mathrm{rel}}) \geq 0.9999$.

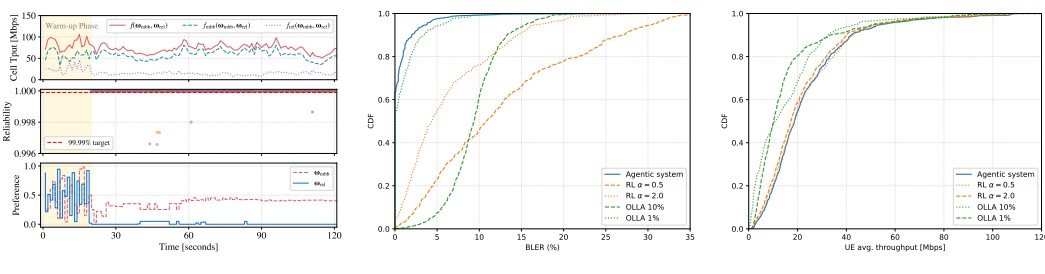

(a) Time series of services KPIs.     (b) BLER for reliable service.     (c) Streaming users throughput.

Figure 3: Validation of the intent fulfillment loop between optimizer-controller for two examples.

Figure 3a shows the optimizer–controller dynamics over a two-minute simulation. After an initial warm-up phase, the PAX-BO optimizer steers $\omega_{\text{mbb}}$ and $\omega_{\text{rel}}$ so that the D-EQL controller applies Pareto-optimal policies matched to each service's requirements under varying network conditions. For reliable services, the optimizer converges to $\omega_{\text{rel}} \approx 0$ (consistent with Figure 2b), driving the controller toward conservative MCS selections that deliver ultra-reliable performance throughout the simulation—exceeding 99.99% reliability in 94% of the run. Only a few packets are lost during isolated deep-fading episodes; under persistent fading, the interpreter could be invoked to relax the reliability target. For enhanced-streaming traffic, the optimizer converges to $\omega_{\text{mbb}} \approx 0.45$, prioritizing higher mean user throughput. Appendix H provides additional analysis and results.

Figure 3b and Figure 3c show that our agentic system outperforms both the state-of-the-art outer-loop link adaptation (OLLA) used in 5G systems and the traditional RL-based LA of Demirel et al. (2025). Unlike our approach—which adapts a single D-EQL model on-the-fly to different connectivity requirements and radio conditions—both OLLA and traditional RL require separate configurations optimized for each service type. For OLLA, we consider a standard target BLER of 10% for maximizing throughput in streaming services and 1% for highly reliable transmissions. Traditional RL similarly requires distinct models with reward functions tailored to each service; following Demirel et al. (2025), we use robustness parameters $\alpha = 0.5$ for throughput and $\alpha = 2$ for reliability. Figure 3b shows that our agentic system achieves substantially lower BLER for reliable services than both OLLA and the RL baseline with $\alpha = 2$, yielding more reliable transmissions. Figure 3c further shows that the same D-EQL model also attains throughput comparable to an RL model explicitly trained for throughput optimization. While D-EQL handles both services with a single model, using multiple RL models is impractical: inference must complete within a few hundred microseconds for all users, making rapid model switching across services infeasible.

### 7.3 TRIADIC AGENT WORKFLOW VALIDATION

We next evaluate the complete agentic AI system, with both intent management and intent fulfillment loops working in unison to provide a continuous solution to an intent formulation that combines a primary system objective (i.e., cell throughput) with QoS requirements of a connectivity service:

> *Maximize cell throughput and serve streaming users with a minimum average data rate of* 7 Mbps *whenever possible.*

The peculiarity of this problem stems from the highly likelihood of the QoS requirements to become infeasible for users with poor channel conditions (such as cell-edge and high mobility users). When such an event occurs, persisting with a rigid QoS requirement would induce the system to over-provision users with poor channel regardless of their inability to meet the QoS goal, at the expense of users with a better channel quality. In turns, this may induce users with better channel to achieve lower throughput (due to less resources) and therefore compromise the primary intent objective.

Figure 4 compares the agentic AI system with two settings: (a) a formulation with rigid QoS requirements; and (b) a formulation with flexible QoS requirements. In the latter case, when the optimizer agent alerts the interpreter agent of a consistent violation of the service constraint, the interpreter reasons over the cause of the problem and plans a solution to relax the QoS requirements.

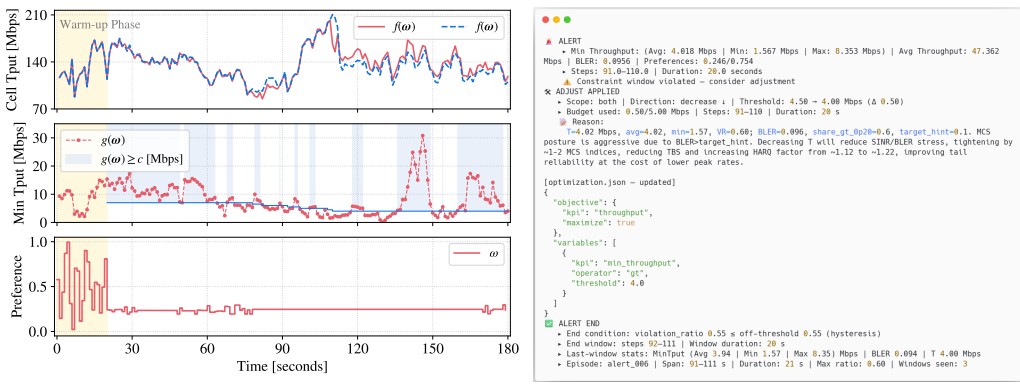

(a) Agentic system with/without OTM refinement.  (b) Intent management loop signaling.

Figure 4: Validation of the full agentic workflow, with intent management loop and intent fulfillment loop working in unison. We compare two formulations with rigid and flexible service requirements.

Figure 4b shows an instance of this intent management loop between the interpreter-optimizer agents, where the latter reacts to the constraint violation by relaxing the service threshold, in an attempt to improve the primary objective, and providing a revised OTM. This choice allows the optimizer agent to choose an $\omega$ setting that guides the MORL controller towards a less aggressive MCS selection policy for LA, making packets transmissions more reliable for users with poor channel conditions.

Despite the interpreter's recursive adaptation of QoS requirements, infeasibility may still persist. This occurs because (a) the adaptor module includes guardrails that prevent abrupt QoS changes during OTM refinement (cf. Appendix A); and (b) prolonged poor channel conditions—such as deep fading, high pathloss, or shadowing—may yield spectral efficiencies too low to satisfy the QoS constraints, regardless of how the interpreter adjusts them. Nonetheless, adapting the OTM still yields tangible system-level benefits. By relaxing QoS targets for users in persistently poor channel conditions, the system frees radio resources that can be reallocated to users with better channel quality, thus with higher spectral efficiency. This redistribution increases the primary intent objective (cell throughput), even if some individual QoS constraints remain infeasible. As illustrated in Figure 4a, once OTM adaptation begins in the second half of the simulation, the cell throughput improves by a 4.79%.

## 8 CONCLUSIONS

We presented an Agentic AI system for intent-driven control in autonomous networks, structured around three cooperating agents: interpreter, optimizer, and controller. Their coordinated interaction links high-level service intents to concrete network actions, enabling continuous reasoning, trade-off resolution, and real-time adaptation across multiple timescales of autonomous network control.

Our contributions span the full intent-to-control pipeline. The interpreter uses a lightweight dual-SLM architecture to convert natural-language intents into structured optimization templates, assess feasibility, diagnose constraint violations, and refine templates using optimizer feedback. The optimizer performs preference planning via BO, dynamically adjusting the downstream controller's policy to meet the service requirements encoded in the template. The controller builds on MORL to execute fast-timescale actions and adapt policies to evolving network conditions. To support this role, we introduce a distributed MORL algorithm that integrates envelope Q-learning with actor–learner decoupling, preference-space exploration, and prioritized hindsight replay, improving scalability, exploration coverage, and performance over state-of-the-art MORL approaches.

Proof-of-concept experiments in a high-fidelity, 5G-compliant RAN simulator demonstrate that the proposed system reconciles heterogeneous service requirements—including throughput and reliability—while operating near the Pareto front of network performance and adapting effectively to dynamic conditions, exceeding traditional RL and state-of-the-art functions of in 5G/5G-A systems.

Looking ahead, a key challenge is scaling this workflow across hierarchical layers of the RAN—from cell-level control to cluster-level coordination and end-to-end service orchestration—while ensuring intent consistency, agent interoperability, and robustness to uncertainty at each level.

# 9 LLM USAGE STATEMENT

In this paper, the authors used LLMs to check grammar, spelling, punctuation, and style compliance.

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

CONTENTS OF APPENDIX

## A  INTERPRETER AGENT: RESPONSIBILITIES, DESIGN, IMPLEMENTATION

### A.1  SCOPE AND RESPONSIBILITIES

The *interpreter agent* is the gateway from high-level intent to optimization-ready control. It fulfills two primary responsibilities: (1) translating intents expressed in natural language into an initial structured OTM; and (2) recursively reasoning over system observations and optimizer feedback to stabilize intent fulfillment by revising the OTM when required (e.g., when constraints become infeasible).

**Division of Labour (Dual-SLM).**  To address these responsibilities under tight computational budgets, we employ two complementary SLMs: (1) a **fine-tuned SLM** for intent-to-OTM translation; and (2) an **in-context learning (ICL) based SLM** for adaptive intent management, which reasons over structured prompts and windowed KPI statistics, refines intent requirements when needed, and provides an explicit textual rationale.

While alternative realizations of an interpreter agent are possible, our design enables the use of lightweight SLMs that adhere to the compute and memory constraints of 4G/5G RAN systems (see Appendix A.5).

### A.2  ARCHITECTURAL OVERVIEW

The interpreter agent architecture, showed in Figure 5, consists of four tightly coupled modules:

- **Translator (Appendix A.2.1)** uses a fine-tuned SLM to convert an incoming intent into a structured, machine-readable OTM that specifies objectives, constraints, aggregation units, and provenance for different connectivity services and operational goals.

- **Monitor (Appendix A.2.2)** subscribes to optimizer telemetry, aligns the telemetry stream to the OTM-defined intent-management timescale, extracts per-window summaries, and bridges short gaps.

- **ICL-based Advisor (Appendix A.2.3)** uses an ICL-based SLM to reason over window summaries and active policy thresholds, selects an advisory direction

$$a \in \{\texttt{increase}, \texttt{decrease}, \texttt{no\_change}\}$$

  and generates a compact rationale $\mathcal{R}$ grounded in RRM. It proposes only a direction, not a magnitude.

- **Adaptor (Appendix A.2.4)** converts the advisory action $a$ into a bounded threshold update $\Delta b$ under guardrails (e.g., caps, lifetime budget, floor/ceiling, cooldown), persists the updated threshold atomically into the OTM, and emits an audit record.

During the intent-management loop, the OTM is treated as a *living document* jointly maintained by the interpreter and optimizer agents. The optimizer continuously solves against the current OTM snapshot and reports telemetry (e.g., windowed KPI statistics) to the monitor. Guided by this feedback, the ICL-based advisor recommends adjustments when intent requirements become overly tight or infeasible under the current network state. The adaptor then applies bounded updates to the corresponding OTM constraints, yielding a refreshed OTM for the optimizer.

#### A.2.1  TRANSLATOR

The translator employs a fine-tuned SLM to convert intents into deterministic, schema-compliant OTM instances. Its role extends well beyond lexical parsing: it must interpret natural-language intents into meaningful optimization structures grounded in domain knowledge, and identify the appropriate downstream control agent to execute them. For example, a service intent requesting high reliability—such as the case in Section 7.2—may translate into a non-obvious constraint formulated in terms of BLER.

Further details on the translator design are provided in Appendix B and Appendix C, which discuss the OTM schema and the supervised fine-tuning and evaluation of the translator SLM, respectively.

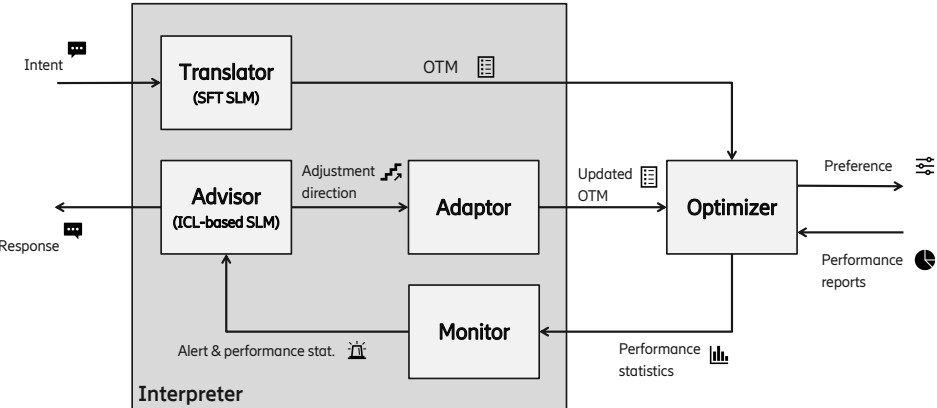

Figure 5: **Dual-SLM interpreter agent.** A (supervised) fine-tuned SLM generates the OTM; an intent monitor aligns telemetry; an ICL-based advisory module outputs discrete adjustment directions with rationale; and an adaptor applies bounded updates and persists them atomically. The optimizer then solves against the latest OTM snapshot, with telemetry closing the loop.

### A.2.2 SLIDING-WINDOW MONITOR

Consider a single constraint $k^A = \langle \texttt{kpi}, \odot, b, A, \texttt{unit} \rangle$ of an OTM, where $\odot \in \{\leq, \geq\}$ and $A$ is the per-step aggregation operator declared in the OTM (e.g., $\texttt{mean}$, $\texttt{min}$, $\texttt{max}$, $\texttt{p95}$). To simplify notation, we refer to the constraint function $k^A(\cdot)$ as $y(\cdot)$, and let $y_t$ denote the KPI value at step $t$ *after* applying $A$ over the telemetry bin of length $\Delta$ (e.g., $10\,\text{s}$). With window length $W$, the monitor maintains a ring buffer over $\{y_i\}_{i=t-W+1}^{t}$ and computes a *signed margin*:

$$m_i = s(y)\left(y_i - b\right) \qquad \text{where} \qquad s(y) = \begin{cases} +1, & \odot \in \{\geq\} \quad \text{(lower bound)} \\ -1, & \odot \in \{\leq\} \quad \text{(upper bound)}. \end{cases}$$

A step $i$ is a violation if and only if $m_i < 0$ (negative margin). The window statistics are then

$$\text{violation\_ratio}(t) = \frac{1}{W} \sum_{i=t-W+1}^{t} \mathbf{1}[m_i < 0],$$

$$\bar{y} = \frac{1}{W} \sum_{i=t-W+1}^{t} y_i, \qquad y_{\min} = \min_i y_i, \qquad y_{\max} = \max_i y_i,$$

and the average *shortfall/slack* (useful for controllers and prompts):

$$\text{shortfall\_avg} = \frac{1}{W} \sum_{i=t-W+1}^{t} \max\{0, -m_i\},$$

$$\text{slack\_avg} = \frac{1}{W} \sum_{i=t-W+1}^{t} \max\{0, m_i\}.$$

**Hysteresis and alerting.** Hysteresis prevents chattering: An ALERT_START event is declared when $\text{VR} > \rho_{\text{on}}$ and an ALERT_END event when $\text{VR} < \rho_{\text{off}}$ with $\rho_{\text{on}} > \rho_{\text{off}}$. At each window end (the decision point), if an alert is active, the monitor produces a compact, constraint-centric context:

{
    window : $\{W, t-W+1 \dots t, \text{VR}\}$,
    constraint_metric : $\{\bar{y}, y_{\min}, y_{\max}, b, \text{shortfall\_avg}, \text{slack\_avg}, \texttt{unit}\}$,
    constraint_id : $\texttt{id}$,
}

Optionally, the context may be augmented with domain-specific auxiliaries (e.g., $\texttt{aux\_kpis}$) if available.

Table 3: Key hyperparameters of the interpreter agent.

| Symbol | Name | Description |
|---|---|---|
| $W$ | Window size | Number of samples used to compute moving averages and the violation ratio. |
| $b$ | Threshold | Current target value for the monitored KPI. |
| $\rho_{\mathrm{on}}$ | Alert-on ratio | Violation ratio above which an alert episode is initiated. |
| $\rho_{\mathrm{off}}$ | Alert-off ratio | Violation ratio below which an alert episode is terminated. |
| $d$ | Step size | Base increment or decrement applied to threshold updates. |
| $g_\uparrow, g_\downarrow$ | Guardrail gains | Maximum upward or downward adjustment permitted per update. |
| $s_{\mathrm{max}}$ | Smoothing cap | Maximum smoothing applied across consecutive updates. |
| $B$ | Budget | Maximum number of updates allowed within a single alert episode. |
| $b_{\mathrm{min}}, b_{\mathrm{max}}$ | Bounds | Minimum and maximum permissible threshold values. |
| $C$ | Cooldown | Minimum number of steps that must elapse before another update can be applied. |

**Complexity.** The monitor executes in $O(1)$ time per step through the use of a fixed-size ring buffer and incremental summary updates, with no rescans required. Memory usage grows linearly with the window size, i.e., $O(W)$.

### A.2.3 ADVISOR (ADVISORY LAYER)

The advisory layer determines the *direction of adaptation* and supplies a textual justification $\mathcal{R}$. It does *not* specify the magnitude of change. Two modes are supported:

(a) **Rule-based.** Thresholds on summary statistics (e.g., violation ratio, mean deviation from the target, minimum deviation from the target, auxiliary posture indicators) determine an advisory action $a$.

(b) **ICL-based SLM** A structured prompt encodes (i) the set of allowed actions, (ii) the decision policy, (iii) domain-specific guardrails, and (iv) a strict JSON output schema. The SLM produces an advisory adjustment

```
{"action": "...", "justification": "..."}
```

conditioned on the parsed telemetry payload from the intent monitor.

**Guardrails in Prompting.** Schema fidelity and reproducibility are enforced through:

(a) JSON-only outputs;

(b) end-of-sentence token fences;

(c) banned tokens (e.g., URLs, markdown code fences); and

(d) near-deterministic decoding with low-variance sampling to avoid verbatim repetition while maintaining stability.

The justification must cite explicit numerical values extracted from the payload (e.g., target $b$, mean $\bar{y}$, minimum $y_{\mathrm{min}}$, violation ratio VR) and must classify posture relative to a domain-specific auxiliary metric (e.g., "aggressive" vs. "conservative").

**Prompt Contract (Abridged).** Allowed actions are $\{\texttt{increase}, \texttt{decrease}, \texttt{no\_change}\}$. The required output format is strictly JSON:

```
{"action":"...", "justification":"..."}.
```

The justification must reference the relevant statistics and the auxiliary posture label. Domain-specific instantiations (e.g., using BLER as the auxiliary metric) appear in examples in Section 7.

### A.2.4 ADAPTOR (MAGNITUDE, SAFETY, PERSISTENCE)

Given an advisory action $a \in \{\texttt{increase}, \texttt{decrease}, \texttt{no\_change}\}$, the adaptor computes a candidate step size $\Delta b$ using a deadband $d$ and asymmetric gains $(g_\uparrow, g_\downarrow)$:

$$\Delta b = \begin{cases} g_\downarrow \max\big(0, (b - \bar{x}) - d\big), & a = \texttt{decrease}, \\ g_\uparrow \max\big(0, (\bar{x} - b) - d\big), & a = \texttt{increase}, \\ 0, & a = \texttt{no\_change}. \end{cases}$$

Safety guardrails limit the actuation:

$$\Delta b \leftarrow \min\{\Delta b, \, s_{\max}, \, B_{\text{left}}, \, b - b_{\min}, \, b_{\max} - b\}, \qquad b \leftarrow \text{clip}(b \pm \Delta b, \, b_{\min}, b_{\max}).$$

Budgets and cooldown counters are updated after each actuation. Final thresholds are written atomically to the OTM, ensuring that the optimizer and monitor operate on consistent snapshots. Section 7 illustrates with concrete examples (e.g., throughput maximization with minimum guarantees per user, or bounds on BLER) how this generic mechanism applies across KPIs.

### A.3 ALGORITHMIC SUMMARY AND INTERFACES

The closed-loop operation of the interpreter agent—integrating monitoring, advisory, and adaptation—is summarized in Algorithm 1. The procedure shows how the agent detects constraint violations, issues advisory actions, and applies bounded adaptations under guardrails.

---

**Algorithm 1** Interpreter Agent (Monitor $\rightarrow$ Advisor $\rightarrow$ Adaptor)

---

1: **Input:** window size $W$; thresholds $(b, \rho_{\text{on}}, \rho_{\text{off}})$; guardrails $(d, g_\uparrow, g_\downarrow, s_{\max}, B, b_{\min}, b_{\max}, C)$
2: **for** each step $t$ **do**
3:      Push observation $y_t$ into ring buffer; update $(\bar{y}, y_{\min}, \text{VR})$
4:      **if** $\text{VR} > \rho_{\text{on}}$ and not in alert **then**
5:          Start episode; reset budget and cooldown
6:      **end if**
7:      **if** in alert **then**
8:          Build parsed telemetry payload; select action $a$ via rules or ICL SLM; log rationale $\mathcal{R}$
9:          **if** $a \neq \texttt{no\_change}$ and cooldown expired and $B_{\text{left}} > 0$ **then**
10:             Compute $\Delta b$; apply guardrails; update $b$; persist OTM; decrement budget; reset cooldown $C$
11:          **end if**
12:          **if** $\text{VR} < \rho_{\text{off}}$ **then**
13:             End episode; log summary
14:          **end if**
15:      **end if**
16: **end for**

---

### A.3.1 INTERFACES

**(i) From Monitor to Advisor.** Upon receiving telemetry from the optimizer, the intent monitor produces a compact summary aligned to the OTM timescale. This parsed payload becomes the sole input to the ICL-based advisory module. An example summary from our experiments is:

```
{
  "window":                 {
                              "start": 1020, "end": 1139, "W": 12, "violation_ratio": 0.60
                            },
  "constraint_metric": {
                          "name":"throughput",
                          "avg": 6.92,
                          "min": 3.08,
                          "monitor_threshold": 7.00,
                          "unit":"Mbps"
                        },
  "radio_kpis":          {"bler": {"avg":0.14, "target_hint":0.10}}
}
```

**(ii) From Advisor to Adaptor.** The advisory module returns only an *adjustment direction* along with a textual justification, both constrained by the *current OTM* used by the optimizer. It never proposes numeric magnitudes. Example output:

```
1  {
2     "action": "decrease",
3     "justification": "relax to reduce MCS pressure and HARQ overhead."
4  }
```

**(iii) From Adaptor to OTM (atomic).** The adaptor converts the advisory direction into a bounded step $\Delta b$, applies guardrails (e.g., clipping to $[b_{\min}, b_{\max}]$), persists the updated threshold atomically, and records the rationale:

```
1  {
2     "kpi":              "throughput",
3     "aggregation":      "min"
4     "old_threshold":    7.00,
5     "new_threshold":    6.92,
6     "delta":            -0.08,
7     "episode":          "alert_002",
8     "rationale":        "VR=0.60; BLER aggressive"
9  }
```

## A.4 MODELS

**Fine-tuned SLM (Intent-to-OTM).** A domain-specialized causal SLM is fine-tuned to generate OTM JSON directly from natural-language intents. Training uses instruction-style pairs of the form (intent, OTM) that adhere to domain schemas (objective, KPI, operator, threshold). The model is evaluated using exact-match accuracy and schema validity. This component is implemented using the Qwen-2.5-7B-Instruct model (Qwen et al., 2025) with supervised fine-tuning; additional details are provided in Appendix C.

**ICL-based SLM (Constraint Adaptation).** A general-purpose SLM—also based on Qwen-2.5-7B-Instruct (Qwen et al., 2025) but without task-specific weight updates—is prompted with: (i) the allowed actions and guardrails, (ii) policy rules governing the violation ratio (VR) and KPI slack/shortfall, (iii) BLER posture rules with target hints, and (iv) a strict JSON schema. Outputs are assessed for schema validity, internal consistency (e.g., adherence to policy rules), and justification quality.

## A.5 STABILITY, SAFETY, AND COMPLEXITY

Guardrails constrain actuation by ensuring that the target parameter $b$ remains within the safe interval $[b_{\min}, b_{\max}]$. A hysteresis mechanism further prevents rapid oscillations caused by frequent threshold updates. The computational overhead of the method is minimal: each control step requires constant time $O(1)$, and memory usage grows linearly with the window size $O(W)$. This design minimizes the impact on RAN compute and memory resources.

To evaluate the practical performance of the agentic AI system for intent management, we report the following metrics: (i) reduction in violation ratio relative to baseline operation; (ii) percentage of observation windows that request a change; (iii) percentage of updates clipped by guardrails; (iv) validity rate of JSON payloads against the schema; (v) observed episode lengths; and (vi) adaptation latency per update.

## A.6 FAILURE MODES AND MITIGATIONS

Despite these safeguards, the system remains susceptible to several failure modes. The corresponding mitigation strategies are:

- **Prompt sensitivity:** Malformed or ambiguous payloads may arise from language model outputs. This risk is mitigated through strict schema enforcement, exclusion of unsafe tokens, and regression testing on canonical telemetry payloads.

- **Distribution shift:** Variations in traffic or channel conditions can create discrepancies between training and deployment distributions. The system addresses this through window normalization and by providing BLER posture hints to the model. In extreme cases, the controller can revert to a rules-only mode to preserve stability.

- **Over-actuation:** Excessive threshold adjustments may cause oscillations or instability. To prevent this, the system enforces lifetime update budgets, per-step update caps, cooldown intervals, and explicit floor/ceiling bounds on $b$.

- **Explainability drift:** Generated rationales may deviate from the underlying numerical evidence. The advisory module $\mathcal{R}$ must cite explicit numerical values, and all rationale cards are logged and checked against policy expectations to ensure traceability and consistency.

This appendix outlines how the interpreter agent determines *when to act*, *how to act and why*, and *to what extent to act*. These behaviors are realized through dual SLMs, classical control guardrails, and auditable OTM persistence.

## B  OPTIMIZATION TEMPLATE MODEL

**Purpose.** The OTM defines the contract between the interpreter agent and the downstream optimizer. It (i) specifies the optimization *objective* and the associated *constraints*, including explicit units and aggregation semantics; (ii) records provenance for auditability (`origin`, `modified_by`); and (iii) serves as a *living document* that can be safely updated by the adaptor during execution.

**Formal view.** Let $\mathcal{X}$ denote the optimizer's decision space, and let $k(\cdot)$ be a network KPI evaluated under an aggregation operator $A$ (e.g., mean, min, $p95$). We define an OTM instance as

$$\max_{x \in \mathcal{X}} k_{\text{obj}}^{A_{\text{obj}}}(x) \quad \text{s.t.} \quad \forall i \in \{1, \cdots, m\} : \begin{cases} k_i^{A_i}(x) \leq b_i & \text{if operator} \in \{\text{lt}, \text{le}\} \\ k_i^{A_i}(x) \geq b_i & \text{if operator} \in \{\text{gt}, \text{ge}\} \end{cases} \quad (3)$$

where each constraint $i$ specifies `service`, `kpi`, `operator`, threshold $b_i$, aggregation $A_i$, units, and scope. In essence, this formulation revisits the optimization (1) by rewriting the objective $f(\cdot)$ and the constraints $g_i(\cdot)$ in terms of a more generic KPI construct $k(\cdot)$ used in the OTM schema.

### B.1  OTM SCHEMA AND DOMAIN SEMANTICS

The OTM schema is a minimal versioned JSON contract comprising four blocks, `objective`, `constraints`, and `metadata`, `version`, characterizing the OTM formalism in equation 3.

Listing 1: Generic OTM schema applicable to different RAN control problems.

```
{
  "objective": {
    "service": <service_name>,        // {"mbb", "urllc", "gaming", "streaming", slice, ...},
    "kpi": <kpi_name>,                 // {"throughput", "reliability", "latency", ...}
    "scope": <scope_name>,             // {"per_user","per_cell", "per_slice", ...}
    "aggregation": <aggr_name>,        // {"mean","min","max","p95","sum", ...}
    "unit": <unit_name>,               // {"Mbps","Gbps","ms","s","%", ...}
    "maximize": <value>,               // boolean value {true, false}
  },
  "constraints": [
    {  "id": <value>                   // string value (e.g., "C1", "C2")
       "service": <service_name>,      // {"mbb", "urllc", "gaming", "streaming", ...},
       "kpi": <kpi_name>,              // {"throughput", "reliability", "latency", ...}
       "scope": <scope_name>,          // {"per_user","per_cell", "per_slice", ...}
       "aggregation": <aggr_name>,     // {"mean","min","max","p95","sum", ...}
       "unit": <unit_name>,            // {"Mbps","Gbps","ms","s","%", ...}
       "operator": <operator_type>,    // {"lt","le","ge","gt"}
       "threshold": <value>,           // float expressed in "unit"
       "modified": <value>,            // boolean value {true, false}
    },
    {
       ...
    }
  ],
  "metadata": {
    "otm"{
       "id": <value>                   // string value (e.g., "O1", "O2")
       "created_by": <model_id>,       // string value {"SFT_LLM", ...}
       "timestamp": <value>,           // formatted as iso-8601
       "timescale": <value>            // string with window value (e.g., "10s_window")
       }
    "episode: {
       "id": <value>                   // string value (e.g., "E1", "E2")
       "episode_type: <type_name>      // {"alert", "alert_resolved", ...}
       "modified_by: <model_id>,       // string value {"ICL_LLM", ...}
       "timestamp: <value>             // formatted as iso-8601
       }
    "adaptation_log": []
  },
  "version": "1.0",
}
```

*Note:* Listings include `//` comments for readability; they are illustrative and not strict JSON.

This structure is simple, yet generic enough to accommodate a wide range of problems, from simple single-service policies and more complex multi-service optimization directives spanning typical

mobile traffic types—e.g., URLLC, mMTC, streaming, web, gaming, and voice—each associated with domain-appropriate KPIs (e.g., reliability and latency for URLLC, throughput and jitter for gaming and streaming). Specifically, the OTM blocks define:

- **objective**: This block specifies the primary goal to optimize for a plurality of services, including KPIs, their aggregation level (`mean/min/max/p95/sum`), scope, unit, and optimization sense (i.e., maximize or minimize).
- **constraints**: This block encodes optional service-specific KPI bounds, each expressed as an inequality $k^A \odot b$. To this end, it shares the same fields of the `objective` block, and additionally includes an `operator` (`lt/le/ge/gt`) the specifies the relation to a `threshold` $b$ expressed in the stated `unit`. Optionally, it includes fields indicating modification to a service constraint (`modified`, `modified_by`, `id`).
- **metadata**: This block records OTM static in formation, such as provenance, time of origination, etc. and dynamic information related to the last epsode event that triggered a modification of the OTM to the and the aggregation `timescale`, a `timestamp`, an `episode` identifier, and an append-only `adaptation_log`.
- **version**: specifies the OTM version.

**OTM fields semantics.** The OTM schema currently requires only 15 fields, some of which are common across the schema blocks:

- **service**: Specifies a service class (e.g., `mbb`, `urllc`, `gaming`, `streaming`, `slice`).
- **kpi**: Indicates canonical KPIs key resolvable by both the telemetry layer and the optimizer (e.g., `throughput`, `reliability`, `latency`, `bler`).
- **scope**: Indicates a spatial or logical domain related to a KPI scope (e.g., `per_user`, `per_user`, `per_slice`, `per_user_group`, `per_cell_group`, etc.)
- **aggregation**: Indicates an operator defining how raw samples are aggregated to optimize, evaluate or compare a KPI (e.g., `mean`, `min`, `max`, `sum`, $p95$, etc.).
- **unit**: Indicats the type of `unit` used for KPI or a threshold value (e.g., `Mbps`, `Gbps`, `ms`, `s`, `%`, etc.).
- **operator**: Defines relational semantics in a constraint like $\geq$, $\leq$, $=$ etc. (e.g., `le`, `ge`, `ge`, `gt`, `eq` etc).
- **threshold**: Defines the threshold value $b_i$ associated to a constraint stated `unit`; for modified constraints, the value is updated atomically by the adaptor.
- **maximize**: Defines the direction of an optimization (can be `true` or `false`)
- **id**: Indicate an identifies associated with OTM, a constraint, an event, etc.
- **episode_type**: Indicate the type of event that caused a revision of the OTM.
- **created_by**: Identifies the model or module that originated the OTM
- **modified/modified_by**: Indicate whether an OTM has been modified and by which model or module
- **timestamp**: Records events times, such as OTM creation and modification...
- **timescale**: Indicates monitoring window

The field **metadata.adaptation_log** is append-only and used to trace updates with ⟨old, new, $\Delta$, rationale, episode, time⟩. Table 4 exemplifies how typical lexical descriptions of service goals or requirements are mapped into the OTM schema fiels:

**Lifecycle and updates.** The fine-tuned LLM creates the initial OTM combining connectivity service intents and network operational intents. This includes verifying the OTM schema validity prior to hand-off to the optimizer through a set of rules: (i) All KPIs must declare units and aggregation; (ii) the `operator` must be consistent with KPI directionality; (iii) thresholds must lie within domain bounds. During the execution, the ICL-based advisor may propose an update direction with rationale based on telemetry data. The adaptor then computes thresholds adjustments $\Delta b$ under guardrails, updates the target threshold, and persists the new OTM snapshot atomically. Each episode produces a versioned OTM with a growing `adaptation_log`.

| Lexical description | OTM schema field values | | | | | | |
|---|---|---|---|---|---|---|---|
| | KPI | Unit | Aggregation | Scope | Maximize | Operator | Threshold |
| Maximize mean cell throughput | throughput | Mbps | mean | per_cell | true | – | – |
| Minimum user rate above 7Mbps | throughput | Mbps | min | per_user | – | ge | 7 |
| Mean users BLER smaller than 10% | bler | % | mean | per_user | – | le | 10 |
| 95%-tile users latency less than 10ms | latency | ms | p95 | per_user | – | le | 10 |

Table 4: Examples of OTM schema values for typical service definitions.

## B.2 EXAMPLE OF OTM ADAPTATION.

Listing 2 illustrates an OTM produced by the fine-tuned LLM. The `objective` is to maximise mean throughput (Mbps). Three constraints are active: **C1** enforces mean BLER $\leq 0.10$ over a per-cell window (unitless ratio); **C2** caps user-level latency at $20$ ms using the $p95$ aggregator; and **C3** requires a per-cell minimum user throughput of at least $b = 7.00$ Mbps. Provenance marks **C3** as `modified_by: ICL_LLM`, indicating that its threshold may be adjusted online. The `metadata` block specifies the aggregation timescale (`10s_window`) and records a snapshot timestamp/episode.

Listing 2: Illustrative OTM with multiple constraints, before adaptation.

```
1  {
2    "version": "1.0",
3    "objective": {
4      "service": "mbb",
5      "kpi": "throughput",
6      "aggregation": "mean",
7      "unit": "Mbps",
8      "maximize": true
9    },
10   "constraints": [
11     {
12       "id": "C1",
13       "service": "mbb",
14       "kpi": "bler",
15       "operator": "le",
16       "threshold": 0.10,
17       "aggregation": "mean",
18       "unit": "",
19       "scope": "per_cell_window",
20       "origin": "fine_tuned_LLM"
21     },
22     {
23       "id": "C2",
24       "service": "mbb",
25       "kpi": "latency_ms",
26       "operator": "le",
27       "threshold": 20,
28       "aggregation": "p95",
29       "unit": "ms",
30       "scope": "per_user_window",
31       "origin": "fine_tuned_LLM"
32     },
33     {
34       "id": "C3",
35       "service": "mbb",
36       "kpi": "tpt_min_mbps",
37       "operator": "ge",
38       "threshold": 7.00,
39       "aggregation": "min",
40       "unit": "Mbps",
41       "scope": "per_cell_window",
42       "origin": "fine_tuned_LLM",
43       "adapted_by": "ICL_LLM"
44     }
45   ],
46   "metadata": {
47     "timescale": "10s_window",
48     "timestamp": "2025-09-22T10:20:00Z",
49     "episode": "alert_001",
50     "adaptation_log": []
51   }
52 }
```

At runtime, the sliding-window monitor observes a violation ratio $\mathrm{VR} = 0.60$ for `tpt_min_mbps`, together with $\overline{\mathrm{BLER}} = 0.14$ (classified as *aggressive* relative to the $0.10$ target). The ICL-based advisory selects action `decrease`; the adaptor computes a clipped update $\Delta b = -0.08$ (subject to caps, budgets, and bounds) and persists the new threshold. Listing 3 shows the resulting *living* OTM: only **C3** changes ($b : 7.00 \to 6.92\,\mathrm{Mbps}$), while **C1** and **C2** remain unchanged. An `adaptation_log` entry documents the update with $\langle$`old_threshold`, `new_threshold`, $\Delta b$, `rationale`, `episode`, `timestamp`$\rangle$.

Listing 3: Same OTM after one adaptation of constraint C3.

```
1  {
2    "version": "1.0",
3    "objective": {
4      "service": "mbb",
5      "kpi": "throughput",
6      "aggregation": "mean",
7      "unit": "Mbps",
8      "maximize": true
9    },
10   "constraints": [
11     {
12       "id": "C1",
13       "service": "mbb",
14       "kpi": "bler",
15       "operator": "le",
16       "threshold": 0.10,
17       "aggregation": "mean",
18       "unit": "",
19       "scope": "per_cell_window",
20       "origin": "fine_tuned_LLM"
21     },
22     {
23       "id": "C2",
24       "service": "mbb",
25       "kpi": "latency_ms",
26       "operator": "le",
27       "threshold": 20,
28       "aggregation": "p95",
29       "unit": "ms",
30       "scope": "per_user_window",
31       "origin": "fine_tuned_LLM"
32     },
33     {
34       "id": "C3",
35       "service": "mbb",
36       "kpi": "tpt_min_mbps",
37       "operator": "ge",
38       "threshold": 6.92,
39       "aggregation": "min",
40       "unit": "Mbps",
41       "scope": "per_cell_window",
42       "origin": "fine_tuned_LLM",
43       "adapted_by": "ICL_LLM"
44     }
45   ],
46   "metadata": {
47     "timescale": "10s_window",
48     "timestamp": "2025-09-22T10:28:00Z",
49     "episode": "alert_002",
50     "adaptation_log": [
51       {
52         "id": "C3",
53         "old_threshold": 7.00,
54         "new_threshold": 6.92,
55         "delta": -0.08,
56         "rationale": "VR=0.60; avg=6.92<b=7.00; BLER posture aggressive; relax b to stabilize
             HARQ."
57       }
58     ]
59   }
60 }
```

**Design rationale.** The OTM is deliberately minimal (objective, constraints, metadata) yet extensible (aggregation, scope, provenance). This ensures interface stability across RAN domains while enabling adaptive operation and full auditability of constraint updates.

## C    TRANSLATOR SLM FINE-TUNING

### C.1    DATASET CURATION

This section describes the methodology used to construct the supervised corpus for training the Intent-to-OTM translator, together with a statistical characterization of the resulting dataset. The objective of the curation process is to create a corpus that captures the semantic breadth of natural-language QoS intents encountered in operational networks while ensuring strict adherence to the OTM schema required for structured policy generation. The design integrates domain knowledge from 5G/6G communication systems, QoS-engineering practice, service semantics, and the linguistic variability typical of operator-to-system interactions. The final dataset comprises 90,000 samples derived from 30,000 distinct OTM structures, each paired with three three paraphrased intent utterances.

The construction process is guided by four principles: *schema consistency*, *domain realism*, *linguistic diversity*, and *multi-service generality*. Every instance conforms to the prescribed OTM JSON structure to eliminate structural ambiguity. Services, KPIs, thresholds, aggregation functions, and operator semantics are selected to reflect realistic RAN-engineering practice rather than arbitrary sampling. Multiple paraphrases express the same underlying intent using different linguistic styles, while both single-service and multi-service formulations are included to reflect realistic optimization scenarios such as cross-slice coordination or heterogeneous multi-tenant workloads. Together, these principles ensure that the model learns not only syntactically correct outputs but also semantically grounded mappings aligned with operational decision-making.

Seven KPIs central to QoS and quality of experience (QoE) optimization are represented: latency, packet delay budget, jitter, packet error rate, block error rate, throughput, and spectral efficiency. Each KPI is characterized by its physical unit, optimization orientation (minimize or maximize), and a plausible operational range. Service-specific threshold distributions are used to maintain realism. URLLC thresholds, for example, are drawn from tight low-delay intervals consistent with ultra-reliable low-latency requirements; gaming jitter values are sampled from moderate-sensitivity ranges; and streaming throughput thresholds reflect bandwidth levels typical of video services. These calibrated ranges ensure that the model encounters thresholds reflective of actual RAN-optimization tasks rather than arbitrary numeric values.

Real-world QoS requirements frequently rely on percentile-based performance metrics, and the dataset reflects this by including mean, minimum, maximum, and percentile aggregations from p25 to p99. Sampling is intentionally biased toward domain-appropriate usage: reliability-sensitive KPIs such as latency, jitter, and error rates predominantly use high percentiles (p95 or p99), whereas throughput-oriented KPIs typically rely on mean values. This probabilistic, domain-aware selection encourages the model to internalize the relationship between service reliability expectations and suitable aggregation choices. Constraint operators are chosen in accordance with KPI orientation, with minimization KPIs paired with "$\leq$" constraints and maximization KPIs paired with "$\geq$". Semantically invalid combinations, such as lower bounds on error rates, are excluded to prevent the model from learning physically implausible relations.

Each OTM instance is paired with three natural-language paraphrases produced from four stylistic registers: operator-style technical phrasing, 3GPP-inspired formal language, casual expressions, and terse imperative commands. These stylistic variants emulate the diverse ways in which human operators, analysts, and automated systems articulate QoS intents. The paraphrases incorporate synonyms for KPIs and services, linguistic variations in percentile expressions, and syntactic diversity ranging from multi-sentence descriptions to compact directives. This controlled diversity promotes robustness to real-world phrasing while preserving semantic consistency across paraphrases.

To reflect realistic optimization scenarios in multi-slice and multi-tenant RAN deployments, a controlled fraction of OTMs include multi-service dependencies in which the optimization objective applies to one service while constraints reference another. These cases emulate common operational patterns such as managing cross-service interference or guaranteeing simultaneous user-experience requirements across heterogeneous traffic types. Their presence strengthens the model's ability to process complex interdependencies and to generate coherent, jointly feasible policies.

All generated samples include metadata fields such as an ISO-8601 timestamp and an episode identifier. The episode field is fixed to "unspecified" to avoid introducing unintended temporal

semantics while maintaining compatibility with future policy-orchestration workflows requiring contextual metadata.

The statistical structure of the corpus reflects these design choices. Each OTM specifies one optimization objective and between zero and three constraints consistent with QoS-engineering practices in 5G and 6G networks. Because each template is associated with three paraphrases, the full corpus contains 90,000 samples. Constraint cardinality follows a non-uniform distribution chosen to represent operator practice: approximately 10% of OTMs contain no constraints, 45% contain one, 30% contain two, and 15% contain three. Consequently, roughly 90% of the corpus includes at least one constraint, and nearly half include multiple constraints. This distribution exposes the model to a broad range of multi-constraint optimization scenarios rather than biasing it toward oversimplified workflows.

Service representation spans eight canonical categories—gaming, streaming, web, messaging, URLLC, mMTC, VoLTE, and VoIP. Sampling is intentionally skewed toward services with stringent QoS requirements. URLLC and gaming each account for approximately 20–25% of OTMs, streaming contributes about 15%, voice (VoLTE and VoIP combined) contributes roughly another 15%, and the remainder corresponds to web, messaging, and mMTC use cases. This distribution ensures adequate coverage across both throughput-oriented and latency-critical traffic classes.

KPI coverage is similarly broad: all seven KPIs appear throughout the dataset following the service-specific domain profiles described above. Over 95% of reliability-related constraints use high-percentile aggregations (p90–p99), preserving realism in statistical QoS modeling. Approximately 12–18% of OTM instances include multi-service dependencies, providing inductive signals for joint optimization patterns common in next-generation RAN automation. Finally, linguistic variation reflects the stylistic sampling weights: operator style (40%), 3GPP-inspired formal expressions (30%), casual phrasing (20%), and terse directives (10%). This variation enhances generalization to heterogeneous real-world intent expressions while maintaining semantic consistency across samples.

Collectively, the dataset provides extensive coverage of service semantics, KPI behavior, constraint types, and linguistic variation. Its strict structural consistency, calibrated numerical modeling, and broad paraphrastic diversity make it well suited for supervised fine-tuning of models tasked with translating diverse natural-language intents into precise, schema-compliant OTM structures aligned with RAN-optimization practice.

## C.2 TRAINING METHODOLOGY

The Intent-to-OTM translator is trained using supervised fine-tuning on the curated corpus described in Section C. The task is formulated as a conditional sequence-generation problem: given a natural-language intent and a fixed system prompt, the model must produce a complete and structurally valid OTM in JSON format. Since the mapping between intents and OTMs is deterministic and schema-constrained, the training objective emphasizes exact reproduction of field names, values, ordering, and hierarchical structure.

A transformer-based, instruction-tuned model (Qwen-2.5-7B-Instruct Qwen et al. (2025)) serves as the underlying architecture. Fine-tuning proceeds in a left-to-right autoregressive manner in which each token is generated conditioned on both the input intent and the previously generated output. This preserves the strengths of the pre-trained model while enabling specialization toward domain-specific reasoning over services, KPIs, and QoS constraints.

The optimization pipeline follows established practices for adapting large language models. Stability and parameter-efficiency mechanisms are incorporated, and the specific optimizer settings, learning-rate schedule, and adaptation configuration are reported in Table 13 of Appendix I. These components ensure reliable convergence when generating long, nested JSON structures that are sensitive to single-token variations.

Fine-tuning is performed for a small number of epochs, as the deterministic target format and internal consistency of the dataset enable rapid convergence without significant risk of overfitting. A held-out validation set is used to monitor generalization performance and to detect potential memorization of stylistic artifacts present in the synthetic paraphrases.

Model evaluation combines syntactic and semantic criteria. Token-level accuracy measures fidelity to the target JSON sequence, while a schema-validity check verifies exact compliance with the required OTM specification. In addition, semantic alignment metrics assess whether the model correctly identifies the optimization objective, reproduces the appropriate constraints, and matches the complete ground-truth OTM. Together, these metrics provide a comprehensive assessment of structured intent translation accuracy.

Table 5: Comparison of Evaluation Metrics Between the Fine-Tuned and Baseline Models

| Metric | Fine-Tuned Model | Baseline Model |
|---|---|---|
| Total evaluation examples | 1000 | 1000 |
| JSON valid rate | 1.000 | 1.000 |
| Objective match rate | **1.000** | 0.450 |
| Constraints match rate | **0.98** | 0.215 |
| Full OTM match rate | **0.98** | 0.113 |
| Number of constraint mismatches | 20 | 785 |
| Number of objective mismatches | 0 | 550 |
| Full-match examples | 980 | 113 |

## C.3 COMPARATIVE EVALUATION

This section presents a comparative evaluation of two models for translating natural-language QoS intents into structured OTM representations. The *fine-tuned model* is a supervised LoRA-adapted Qwen-2.5-7B-Instruct trained on a curated corpus of 90,000 intent–OTM pairs (see Section C.2), whereas the *baseline model* is the unmodified Qwen-2.5-7B-Instruct. Both models were evaluated on a held-out test set of 1,000 examples under a strict schema-constrained matching protocol.

As summarized in Table 5, the fine-tuned model demonstrates near-perfect structural and semantic adherence to the OTM schema, achieving 100% JSON validity, 100% objective correctness, and a 98% full OTM match rate. In contrast, the baseline model—although also producing syntactically valid JSON in every case—achieves only 45% objective match and 11.3% full OTM match. These results indicate that prompt-only use of a generic instruction-tuned model is insufficient for reliable schema-grounded semantic parsing.

The most pronounced disparity appears in the reconstruction of constraint sets. The fine-tuned model correctly predicts the complete constraint set—including KPI type, operator, aggregation level, threshold, and service scope—in 98% of cases. The baseline model succeeds in only 21.5% of examples, frequently selecting incorrect KPIs, operators, or units despite emitting valid JSON. As shown in Table 6, the baseline model exhibits severe error inflation, particularly in the "missing constraint" (724 occurrences) and "extra constraint" (737 occurrences) categories. It is worth noting that a single erroneous sample may contribute to multiple error categories simultaneously. In contrast, the fine-tuned model's residual errors remain modest and are concentrated primarily in scope mismatches (50% of erroneous samples), with all other categories occurring infrequently.

Two complementary mechanisms explain the fine-tuned model's performance advantage. First, supervised adaptation aligns the model's internal representation with the deterministic structure of the OTM schema. Although the baseline model possesses broad linguistic and domain knowledge, it lacks incentives to prioritize schema-specific conventions such as canonical KPI naming, service–KPI associations, valid threshold ranges, and operator semantics (e.g., $\leq$ for reliability-oriented KPIs). Fine-tuning effectively anchors the model's output distribution to the OTM schema.

Second, the curated dataset encodes domain priors that are internalized during training. For example, URLLC latency values cluster in the 1–10 ms range; gaming intents commonly include jitter constraints with p95 aggregation; streaming intents typically optimize for throughput; and error-rate KPIs almost always appear with $\leq$ operators. Lacking these priors, the baseline model frequently produces semantically plausible but non-canonical KPI selections, percentile aggregations, or threshold values.

Table 6: Comparison of Error Categories Between the Fine-Tuned and Baseline Models

| Error Category | Fine-Tuned Model | | Baseline Model | |
|---|---|---|---|---|
| | Count | Percentage | Count | Percentage |
| Scope mismatch | 10 | 50% | 76 | 9.7% |
| Missing constraint | 4 | 25% | 724 | 92.2% |
| Extra constraint | 5 | 20% | 737 | 93.9% |
| Threshold mismatch | 2 | 10% | 60 | 7.6% |
| Operator mismatch | 2 | 10% | 68 | 8.7% |
| Aggregation mismatch | 2 | 10% | 20 | 2.5% |
| Count mismatch | 1 | 5% | 72 | 9.2% |

Importantly, the fine-tuned model produces no objective-field errors (0 occurrences in Table 6), indicating complete mastery of service–objective alignment. Its remaining errors occur almost exclusively within the constraint block and are typically minor: substituting p90 for p95, small numerical deviations around thresholds, or occasional variations in service scope. These contrast sharply with the baseline model's structurally inconsistent outputs, which reflect a lack of grounding in the semantics encoded by the OTM schema.

Taken together, Tables 5–6 show that supervised, domain-specific fine-tuning is essential for this task. Although both models generate syntactically valid JSON, only the fine-tuned model functions as a high-precision compiler from natural-language intents to machine-interpretable OTM structures. For practical deployment in autonomous RAN-optimization pipelines, relying solely on prompting a pretrained instruction-tuned model is insufficient; structured domain adaptation is required to ensure correctness, robustness, and operational safety.

# D  OPTIMIZER AGENT DESIGN

## D.1  BAYESIAN OPTIMIZATION

We consider the problem of minimizing an unknown objective function $f : \mathcal{X} \to \mathbb{R}$, over a domain $\mathcal{X} \subseteq \mathbb{R}^d$. The goal is to identify

$$x^\star = \arg\min_{x \in \mathcal{X}} f(x).$$

In classical numerical optimization, one distinguishes between *global optimization*, where the absolute minimum of $f$ is sought, and *local optimization*, where the search is restricted to neighborhoods of an initial point. If $f$ is convex and $\mathcal{X}$ is a convex set, the problem reduces to convex optimization, for which efficient algorithms exist.

However, in many modern applications, $f$ is a *black-box function*, meaning that its closed-form expression is unavailable and evaluations are costly (e.g., expensive simulations or training machine learning models). In such cases, *Bayesian optimization (BO)* provides an efficient framework for global optimization by maintaining a probabilistic model of $f$ and selecting query points via a surrogate criterion known as an *acquisition function*.

### D.1.1  GAUSSIAN PROCESS PRIORS

Bayesian optimization typically employs a *Gaussian process (GP)* prior to model the unknown function $f$. A GP is defined by a mean function $\mu(x)$ and covariance function $K(x, x')$:

$$p(f) = \mathcal{GP}(f; \mu, K).$$

Given a set of noiseless observations

$$\mathcal{D} = \{(x_i, f(x_i))\}_{i=1}^n,$$

the posterior distribution over $f$ is again a GP with updated mean and covariance functions $\mu_{f|\mathcal{D}}(x)$ and $K_{f|\mathcal{D}}(x, x')$. This posterior provides both a predictive mean (exploitation) and predictive uncertainty (exploration), which form the basis of acquisition functions.

### D.1.2  ACQUISITION FUNCTIONS

An *acquisition function* $a(x)$ encodes the utility of evaluating $f$ at a candidate point $x$. Since acquisition functions are cheap to evaluate, the optimization problem is reduced to

$$x_{t+1} = \arg\max_{x \in \mathcal{X}} a(x),$$

or its minimization equivalent. Common acquisition functions include:

**Probability of improvement (PI):**  Selects points with the highest probability of improving upon the best observed value $f'$:

$$a_{\text{PI}}(x) = \Phi\left(\frac{f' - \mu(x)}{\sigma(x)}\right),$$

where $\Phi$ is the Gaussian cumulative distribution function (CDF).

**Expected improvement (EI):**  Accounts for the *magnitude* of improvement:

$$a_{\text{EI}}(x) = (f' - \mu(x))\,\Phi\left(\frac{f' - \mu(x)}{\sigma(x)}\right) + \sigma(x)\,\phi\left(\frac{f' - \mu(x)}{\sigma(x)}\right),$$

where $\phi$ is the Gaussian probability density function (PDF). This criterion naturally balances exploration ($\sigma(x)$) and exploitation ($\mu(x)$).

**Entropy search (ES)**  Reduces uncertainty about the optimizer's location by minimizing the entropy of the distribution $p(x^*|\mathcal{D})$. While analytically intractable, approximations make this approach feasible.

**Upper confidence bound (UCB):** Promotes exploration via an optimism-in-the-face-of-uncertainty principle:

$$a_{\mathrm{UCB}}(x; \beta) = \mu(x) - \beta\sigma(x),$$

where $\beta > 0$ is a tunable trade-off parameter. Despite lacking an expected-utility interpretation, UCB has strong theoretical guarantees for asymptotic convergence to the global optimum.

This Bayesian decision-theoretic framework provides a principled way to trade off exploration and exploitation, making BO a powerful tool for solving expensive black-box optimization problems.

### D.2 PAX-BO: PREFERENCE-ALIGNED EXPLORATION BAYESIAN OPTIMIZATION

We have

$$\Delta^{d-1} = \{\omega \in \mathbb{R}^d_{\geq 0} : \mathbf{1}^\top \omega = 1\}$$

**Decision space and projection.** We have $S$ services $\mathcal{S} = \{1, \ldots, S\}$ and preference dimension $d$. The internal (unconstrained) variable is

$$U = \big[u^{(1)}, \ldots, u^{(S)}\big] \in \mathbb{R}^{d \times S}, \qquad \bar{u} = \mathrm{vec}(U) \in \mathbb{R}^{dS}.$$

Each service $s$ uses a probability-simplex preference

$$\mathbf{w}^{(s)} \in \Delta^{d-1} \quad \text{with} \quad \Delta^{d-1} = \{\mathbf{w} \in \mathbb{R}^d_{\geq 0} : \mathbf{1}^\top \mathbf{w} = 1\},$$

obtained by the columnwise Euclidean simplex projection

$$\mathbf{w}^{(s)} = \Pi_\Delta\big(u^{(s)}\big), \qquad \mathbf{W}(U) = \big[\mathbf{w}^{(1)}, \ldots, \mathbf{w}^{(S)}\big] \in (\Delta^{d-1})^S.$$

Closed form for $\Pi_\Delta$: sort $u$ in descending order, find the threshold $\theta$, and set $\Pi_\Delta(u) = \max(u - \theta\mathbf{1}, 0)$ with $\mathbf{1}^\top \Pi_\Delta(u) = 1$.

**Objective, constraints, and data.** We optimize a single objective $f : (\Delta^{d-1})^S \to \mathbb{R}$ subject to scalar constraints $\{g_i : (\Delta^{d-1})^S \to \mathbb{R}\}_{i=1}^p$. A configuration $U$ is feasible iff

$$g_i\big(\mathbf{W}(U)\big) \leq 0, \qquad i = 1, \ldots, p.$$

At iteration $t$, we evaluate at $U_{t-1}$ and observe

$$o_t = f\big(\mathbf{W}(U_{t-1})\big), \qquad c_t^{(i)} = g_i\big(\mathbf{W}(U_{t-1})\big) \ (i = 1, \ldots, p),$$

forming the dataset

$$\mathcal{D}_t = \Big\{(\bar{u}_k, \ o_k, \ c_k^{(1)}, \ldots, c_k^{(p)})\Big\}_{k=1}^t.$$

**Surrogates and acquisition in $U$-space.** Fit surrogates for the compositions

$$\mathcal{F}(\bar{u}) \approx f\big(\mathbf{W}(U)\big), \qquad \mathcal{G}_i(\bar{u}) \approx g_i\big(\mathbf{W}(U)\big) \ (i = 1, \ldots, p).$$

Let the incumbent best feasible value be

$$f_{t-1}^\star = \max\{ o_k : \ c_k^{(i)} \leq 0 \ \forall i, \ k \leq t-1 \} \quad (\text{use } -\infty \text{ if none}).$$

Define the acquisition on $\bar{u}$ (e.g., constrained Log-EI) by

$$\alpha(\bar{u}) = \mathrm{ACQ}\big(\mathcal{F}, \{\mathcal{G}_i\}; \ \bar{u}, \ f_{t-1}^\star\big) \approx \mathrm{LogEI}\big(\mu_\mathcal{F}(\bar{u}), \sigma_\mathcal{F}(\bar{u}); f_{t-1}^\star\big) \times \prod_{i=1}^p \Phi\Big(-\mu_{\mathcal{G}_i}(\bar{u})/\sigma_{\mathcal{G}_i}(\bar{u})\Big).$$

**Single trust region (TR) in $U$-space.** Maintain center $s_c \in \mathbb{R}^{dS}$ and radius $L > 0$ (half side-length in $\ell_\infty$), giving the box

$$\mathcal{B}_t = \big\{\bar{v} \in \mathbb{R}^{dS} : \ \|\bar{v} - s_c\|_\infty \leq L\big\}.$$

Let $\kappa_s, \kappa_f, \kappa_\ell$ be the success, failure, and "stuck-at-floor" counters. Parameters: $L_0$ (initial), $L_{\min}$ (floor), $L_{\max}$ (cap); thresholds $s_{\mathrm{th}}$ (expand), $f_{\mathrm{th}}$ (shrink); tolerance $\epsilon > 0$.

**Local proposal (inner step).** Choose the next internal point by maximizing $\alpha$ within the TR:

$$\bar{u}_t \in \arg\max_{\bar{v} \in \mathcal{B}_t} \alpha(\bar{v}), \qquad U_t = \mathrm{mat}(\bar{u}_t), \qquad \mathbf{W}_t = \mathbf{W}(U_t) = \Pi_\Delta(U_t).$$

**Success test and TR adaptation.** After executing $\bar{u}_{t-1}$, define the success flag

$$\mathrm{SF}_t = \Big(\forall i : c_t^{(i)} \leq 0\Big) \,\wedge\, \Big(o_t \geq f_{t-1}^\star + \epsilon\Big).$$

If $\mathrm{SF}_t$:

$$f_t^\star \leftarrow o_t, \quad s_c \leftarrow \bar{u}_{t-1}, \quad \kappa_s \leftarrow \kappa_s + 1, \; \kappa_f \leftarrow 0, \; \kappa_\ell \leftarrow 0,$$

and if $\kappa_s \geq s_{\mathrm{th}}$ then

$$L \leftarrow \min(2L, L_{\max}), \qquad \kappa_s \leftarrow 0, \; \kappa_f \leftarrow 0.$$

Else (failure):

$$f_t^\star \leftarrow f_{t-1}^\star, \quad \kappa_f \leftarrow \kappa_f + 1, \; \kappa_s \leftarrow 0,$$

and if $\kappa_f \geq f_{\mathrm{th}}$ then

$$L \leftarrow \max(L/2, L_{\min}), \qquad \kappa_s \leftarrow 0, \; \kappa_f \leftarrow 0, \qquad \kappa_\ell \leftarrow \kappa_\ell + \mathbf{1}\{L = L_{\min}\}.$$

**Smart reset (escape when stuck).** If $L = L_{\min}$ and $\kappa_\ell \geq w$, draw $n$ candidates on the product simplex: $\{\mathbf{W}^{(j)} \in (\Delta^{d-1})^S\}_{j=1}^n$ (e.g., Dirichlet/QMC per column), and set $\bar{u}^{(j)} = \mathrm{vec}(\mathbf{W}^{(j)})$. For each $j$ compute

$$\alpha^{(j)} = \alpha(\bar{u}^{(j)}), \quad P_{\mathrm{feas}}^{(j)} = \prod_{i=1}^p \Phi\Big(-\mu_{\mathcal{G}_i}(\bar{u}^{(j)})/\sigma_{\mathcal{G}_i}(\bar{u}^{(j)})\Big), \quad d^{(j)} = \min_{k \leq t} \big\|\mathbf{W}^{(j)} - \mathbf{W}(U_k)\big\|_F.$$

Normalize $z \in \{\alpha, P_{\mathrm{feas}}, d\}$ to $\tilde{z}^{(j)} \in [0,1]$ and score

$$\mathrm{score}^{(j)} = \tilde{\alpha}^{(j)} \, \tilde{P}_{\mathrm{feas}}^{(j)} \, \big(\tilde{d}^{(j)}\big)^\beta.$$

Choose $j^\star = \arg\max_j \mathrm{score}^{(j)}$ and reset

$$s_c \leftarrow \bar{u}^{(j^\star)}, \qquad L \leftarrow L_0, \qquad \kappa_s, \kappa_f, \kappa_\ell \leftarrow 0.$$

**Action selection (vector-valued $Q$).** Given state $s_t$ and requested service $\sigma_t$, act by preference-aligned scalarization:

$$a_t \in \arg\max_{a \in \mathcal{A}} \langle Q(s_t, a), \mathbf{w}_t^{(\sigma_t)}\rangle, \qquad \mathbf{w}_t^{(\sigma_t)} = \Pi_\Delta\big(u_t^{(\sigma_t)}\big).$$

**Remarks.** (i) All modeling and optimization happens in the unconstrained $\bar{u}$-space; feasibility is enforced by the projection $\mathbf{W}(U) = \Pi_\Delta(U)$. (ii) The single TR stabilizes steps; expand/shrink is governed by $(s_{\mathrm{th}}, f_{\mathrm{th}})$ and improvement tolerance $\epsilon$. (iii) The smart reset proposes diverse, high-acquisition, high-feasibility candidates directly on $(\Delta^{d-1})^S$.

---

**Algorithm 2** PAX-BO — Preference-Aligned eXploration Bayesian Optimization

---

**Objects.**

- Services $\mathcal{S} \triangleq \{1, \ldots, S\}$, preference dimension $d$.

- Internal (unconstrained) variables $U = [u^{(1)}, \ldots, u^{(S)}] \in \mathbb{R}^{d \times S}$; vectorization $\bar{u} = \text{vec}(U) \in \mathbb{R}^{dS}$.

- Columnwise simplex projection: $p^{(s)} = \Pi_\Delta(u^{(s)}) \in \Delta^{d-1}$; stack $P(U) = [p^{(1)}, \ldots, p^{(S)}] \in (\Delta^{d-1})^S$.

- Single objective $f : (\Delta^{d-1})^S \to \mathbb{R}$; constraints are *scalar* functions $\{g_i : (\Delta^{d-1})^S \to \mathbb{R}\}_{i=1}^p$; feasible iff $g_i(P) \leq 0 \; \forall i$.

- Surrogates (e.g., GPs) model compositions $\mathcal{F}(\bar{u}) \approx f(P(U))$ and $\mathcal{G}_i(\bar{u}) \approx g_i(P(U))$ for $i = 1, \ldots, p$.

- Best feasible value $f_t^\star = \max\{o_j : g_i(P_j) \leq 0 \; \forall i\}$ (use $-\infty$ if none).

- Acquisition on $U$: $\alpha(\bar{u}) = \text{ACQ}(\mathcal{F}, \{\mathcal{G}_i\}_{i=1}^p; \bar{u}, f_t^\star)$ (e.g., constrained Log-EI).

**Projection.** For $u \in \mathbb{R}^d$: $\Pi_\Delta(u) = \arg\min_{p \in \Delta^{d-1}} \|p - u\|_2$, closed form: sort/threshold; $p = \max(u - \theta \mathbf{1}, 0)$ with $\mathbf{1}^\top p = 1$.

**Single trust region in $u$-space.** Center $s_c \in \mathbb{R}^{dS}$, radius $L > 0$, box $\mathcal{B} \triangleq \{\bar{v} : \|\bar{v} - s_c\|_\infty \leq L\}$. Counters: successes $\kappa_s$, failures $\kappa_f$, stuck-at-$L_{\min}$ $\kappa_\ell$.

1: **Given:** window $W$; trust–region radii $L_0$ (initial), $L_{\min}$ (shrink floor), $L_{\max}$ (expansion cap); thresholds $s_{\text{th}}$ (successes to expand), $f_{\text{th}}$ (failures to shrink); tolerance $\epsilon$; reset window $w$; candidate count $n$; diversity exponent $\beta$

2: **Init:** choose $U_0$; $P_0 = \Pi_\Delta(U_0)$; $s_c \leftarrow \text{vec}(U_0)$; $L \leftarrow L_0$; $\kappa_s \leftarrow 0$; $\kappa_f \leftarrow 0$; $\kappa_\ell \leftarrow 0$; $\mathcal{D}_0 \leftarrow \emptyset$

3: **for** $t = 1, 2, \ldots$ **do**

4:     **Evaluate** at $\bar{u}_{t-1}$: $P_{t-1} = \Pi_\Delta(U_{t-1})$; observe $o_t = f(P_{t-1})$ and $c_t^{(i)} = g_i(P_{t-1})$ for $i = 1, \ldots, p$

5:     **Update data** $\mathcal{D}_t = \mathcal{D}_{t-1} \cup \{(\bar{u}_{t-1}, o_t, (c_t^{(i)})_{i=1}^p)\}$; keep last $W$; refit $\mathcal{F}, \{\mathcal{G}_i\}$ on $(\bar{u}, o, (c^{(i)}))$; compute $f_{t-1}^\star$

6:     **Suggest next $U$ (local TR maximization)**

$$\bar{u}_t \in \arg\max_{\bar{v} \in \mathcal{B}} \alpha(\bar{v}), \qquad U_t = \text{mat}(\bar{u}_t), \qquad P_t = \Pi_\Delta(U_t).$$

7:     **TR update** with success flag $\text{SF} \leftarrow \left(\forall i : c_t^{(i)} \leq 0\right) \wedge \left(o_t \geq f_{t-1}^\star + \epsilon\right)$

8:     **if** SF **then**

9:         $f_t^\star \leftarrow o_t$; $s_c \leftarrow \bar{u}_{t-1}$; $\kappa_s \leftarrow \kappa_s + 1$; $\kappa_f \leftarrow 0$; $\kappa_\ell \leftarrow 0$

10:         **if** $\kappa_s \geq s_{\text{th}}$ **then** $L \leftarrow \min(2L, L_{\max})$; $\kappa_s \leftarrow 0$; $\kappa_f \leftarrow 0$

11:         **end if**

12:     **else**

13:         $\kappa_f \leftarrow \kappa_f + 1$; $\kappa_s \leftarrow 0$

14:         **if** $\kappa_f \geq f_{\text{th}}$ **then**

15:             $L \leftarrow \max\left(L/2, L_{\min}\right)$; $\kappa_s \leftarrow 0$; $\kappa_f \leftarrow 0$;

16:             **if** $L = L_{\min}$ **then** $\kappa_\ell \leftarrow \kappa_\ell + 1$

17:         **end if**

18:         **end if**

19:     **end if**

20:     **Smart reset (if stuck).** If $L = L_{\min}$ and $\kappa_\ell \geq w$:

21:         Sample $n$ candidates $P^{(j)} \in (\Delta^{d-1})^S$ (Dirichlet/QMC per service); set $\bar{u}^{(j)} = \text{vec}(P^{(j)})$

22:         For each $j$: compute $\alpha(\bar{u}^{(j)})$; $P_{\text{feas}}^{(j)} = \prod_{i=1}^p \Phi\left(-\mu_{\mathcal{G}_i}(\bar{u}^{(j)})/\sigma_{\mathcal{G}_i}(\bar{u}^{(j)})\right)$; $d^{(j)} = \min_{(\bar{u}_k, \cdot, \cdot) \in \mathcal{D}_t} \|P^{(j)} - P_k\|_F$

23:         Normalize $\tilde{z} = \dfrac{z - \min z}{\max z - \min z + \varepsilon}$ for $z \in \{\alpha, P_{\text{feas}}, d\}$; $\text{score}^{(j)} = \tilde{\alpha}^{(j)} \tilde{P}_{\text{feas}}^{(j)} (\tilde{d}^{(j)})^\beta$

24:         Set $s_c \leftarrow \bar{u}^{(j^\star)}$ where $j^\star = \arg\max_j \text{score}^{(j)}$; $L \leftarrow L_0$; $\kappa_s \leftarrow 0$; $\kappa_f \leftarrow 0$; $\kappa_\ell \leftarrow 0$

25:     **Action (vector-valued $Q$).** For state $s_t$ and requested service $\sigma_t$:

$$a_t \in \arg\max_{a \in \mathcal{A}} \langle Q(s_t, a, p_t^{(\sigma_t)}), p_t^{(\sigma_t)} \rangle \quad \text{with} \quad p_t^{(\sigma_t)} = \Pi_\Delta(u_t^{(\sigma_t)}).$$

26: **end for**

---

# E MULTI-OBJECTIVE REINFORCEMENT LEARNING

## E.1 MULTI-OBJECTIVE MARKOV DECISION PROCESS

A multi-objective Markov decision process (MOMDP) extends the traditional Markov decision process (MDP) framework (Puterman, 1994; Sutton & Barto, 2018) by considering not just one, but multiple objectives, which may conflict with each other (Roijers et al., 2013; Yang et al., 2019). Within this framework, an agent seeks to optimize several reward functions simultaneously, each corresponding to a different objective. These objectives can either conflict or complement each other; thus, improvements in one may adversely affect another or contribute positively to shared goals. The primary goal of an MOMDP is to derive a policy that achieves an optimal balance among multiple objectives. This trade-off is typically represented by a Pareto front comprising a set of optimal policies such that no policy can outperform another across all objectives simultaneously, thereby making them non-dominated (Roijers et al., 2013; Moffaert & Nowé, 2014).

Formally, an MOMDP is defined by the tuple $\langle \mathcal{S}, \mathcal{A}, \mathcal{P}, \boldsymbol{r}, \Omega, f_\omega, \gamma \rangle$, where $\mathcal{S}$ denotes the state space, $\mathcal{A}$ the action space, $\mathcal{P}(s' \mid s, a)$ the transition probabilities, and $\gamma \in [0, 1)$ a discount factor. The vector $\boldsymbol{r} = [r_1, r_2, \cdots, r_m]^\top$ represents the $m$-dimensional reward vector, and we assume the preference space $\Omega$ to be the standard $(m-1)$-dimensional simplex (probability simplex), defined as

$$\Delta^{m-1} = \left\{ \boldsymbol{\omega} \in \mathbb{R}^m : \sum_{i=1}^m \omega_i = 1 \text{ and } \omega_i \geq 0 \text{ for } i = 1, \cdots, m \right\}. \tag{4}$$

Here, each preference vector $\boldsymbol{\omega} \in \Omega$ assigns a normalized non-negative weight to each objective, reflecting its relative importance. We focus on a class of MOMDPs with a *linear preference function*, in which a scalarization function $f_\omega(\boldsymbol{r}) = \boldsymbol{\omega}^\top \boldsymbol{r}$ converts the reward vector into a scalar return using the preference vector $\boldsymbol{\omega}$ (Roijers et al., 2013; Hayes et al., 2022). The cumulative expected return under a policy $\pi$ is then given by

$$\hat{\boldsymbol{r}}^\pi = \mathbb{E}\left[ \sum_{t=0}^\infty \gamma^t \boldsymbol{r}(s_t, a_t) \,\middle|\, \pi \right].$$

**Observation 1** *When the preference vector $\boldsymbol{\omega} \in \Omega$ is fixed, an MOMDP reduces to a standard MDP with scalar rewards.*

**Remark 1 (Interpretation of Linear Scalarization)** *The scalarization function $f_\omega(\boldsymbol{r}) = \boldsymbol{\omega}^\top \boldsymbol{r}$ can be interpreted as an expectation. If $\boldsymbol{r} = [r_1, r_2, \ldots, r_m]^\top$ is the reward vector and $\boldsymbol{\omega} = [w_1, w_2, \ldots, w_m]^\top$ is a weight vector with $\sum_{i=1}^m w_i = 1$ and $w_i \geq 0$, then*

$$r_s = \boldsymbol{\omega}^\top \boldsymbol{r} = \sum_{i=1}^m w_i r_i = \mathbb{E}_{i \sim \boldsymbol{\omega}}[r_i].$$

*Thus, $\boldsymbol{\omega}$ can be interpreted as a probability distribution over objectives, and the function $f_\omega$ corresponds to the expected reward under this distribution (Roijers et al., 2013).*

**Example 1 (Multi-Objective Q-Learning)** *In traditional single-objective Q-learning, the agent estimates a scalar Q-value for each state–action pair,*

$$Q(s, a) = \mathbb{E}\left[ \sum_{t=0}^\infty \gamma^t r(s_t, a_t) \,\middle|\, s_0 = s, a_0 = a \right].$$

*In contrast, in multi-objective reinforcement learning, each action may yield a vector of rewards corresponding to different objectives (Roijers et al., 2013; Nguyen et al., 2018).*

*For example, in a self-driving car scenario, the agent may consider:*

- *$r_{speed}$: how fast the car goes,*

- *$r_{fuel}$: fuel efficiency,*

- *$r_{safety}$: safety score.*

*Thus, the Q-value function becomes vector-valued:*

$$Q(s,a) = \begin{bmatrix} Q_{speed}(s,a) \\ Q_{fuel}(s,a) \\ Q_{safety}(s,a) \end{bmatrix} \in \mathbb{R}^3.$$

*Each decision-maker may have different trade-offs between these objectives, expressed as a* preference *vector $\boldsymbol{\omega} \in \Omega = \Delta^2$. For instance, $\boldsymbol{\omega} = [0.7, 0.2, 0.1]^\top$ indicates $70\%$ priority on speed, $20\%$ on fuel efficiency, and $10\%$ on safety. The corresponding scalarized utility is given by $\boldsymbol{\omega}^\top Q(s,a)$.*

## E.2 CONVEX COVERAGE SET

In multi-objective optimization, the *Pareto front* $\mathcal{F}^\star$ contains all Pareto optimal solutions (Roijers et al., 2013; Moffaert & Nowé, 2014), defined as

$$\mathcal{F}^\star \triangleq \left\{ \boldsymbol{r} \in \mathbb{R}^m : \nexists \boldsymbol{r}' \in \mathbb{R}^m \text{ such that } \boldsymbol{r}'_i \geq \boldsymbol{r}_i \text{ for all } i \text{ and } \boldsymbol{r}'_j > \boldsymbol{r}_j \text{ for at least one } j \right\}.$$

However, not all Pareto-optimal points are relevant when preferences are restricted to linear scalarizations. In this case, only a subset of the Pareto front—the *convex coverage set (CCS)*—is sufficient (Roijers et al., 2015).

The CCS is a subset of $\mathcal{F}^\star$ consisting of non-dominated solutions that are optimal under some linear preference vector. Mathematically, it is defined as

$$\mathcal{C} \triangleq \left\{ \hat{\boldsymbol{r}} \in \mathcal{F}^\star : \exists \boldsymbol{\omega} \in \Delta^{m-1} \text{ such that } \boldsymbol{\omega}^\top \hat{\boldsymbol{r}} \geq \boldsymbol{\omega}^\top \hat{\boldsymbol{r}}' \text{ for all } \hat{\boldsymbol{r}}' \in \mathcal{F}^\star \right\}.$$

Thus, the CCS comprises those points on the outer convex boundary of $\mathcal{F}^\star$ that maximize utility for at least one preference vector $\boldsymbol{\omega}$ (with $\Omega = \Delta^{m-1}$ as defined above) (Roijers et al., 2015).

**Example 2** *Consider a bi-objective optimization problem with the objectives of maximizing accuracy and maximizing interpretability. The CCS would include those points on the Pareto front that maximize a weighted sum of the two objectives for some given trade-off between them; these points lie on the convex outer boundary of the feasible set in the objective space.*

Therefore, the CCS represents the minimal subset of the Pareto front that guarantees optimality under some linear preference. It is particularly valuable in decision-making scenarios where preferences may vary, since it identifies exactly those solutions that are relevant for all possible linear trade-offs (Roijers et al., 2013).

## E.3 ENVELOPE Q-LEARNING

**Scope.** Learn a *single* preference-conditioned action-value function $Q : \mathcal{S} \times \mathcal{A} \times \Delta^{m-1} \to \mathbb{R}^m$ such that, for any linear preference $\boldsymbol{\omega} \in \Delta^{m-1}$, the scalar projection $\boldsymbol{\omega}^\top Q(s,a,\boldsymbol{\omega})$ equals the optimal scalarized value for acting under $\boldsymbol{\omega}$ (Yang et al., 2019). The induced policy is

$$\pi_\omega(s) = \arg\max_{a \in \mathcal{A}} \boldsymbol{\omega}^\top Q(s,a,\boldsymbol{\omega};\boldsymbol{\theta}). \tag{5}$$

**Envelope maximizer selection (double DQN (DDQN) style).** Given a transition $(s,a,\boldsymbol{r},s')$ with reward vector $\boldsymbol{r}(s,a) \in \mathbb{R}^m$, EQL (Yang et al., 2019) bootstraps from the *envelope* at the next state by selecting the action–preference pair that maximizes the $\omega$-projection using the *online* network (parameters $\boldsymbol{\theta}$):

$$(a^\star, \omega^\star) = \arg\max_{a' \in \mathcal{A},\, \omega' \in \Delta^{m-1}} \omega^\top Q(s', a', \boldsymbol{\omega}';\theta). \tag{6}$$

This couples preferences because $\boldsymbol{\omega}^\star$ is not necessarily equal to the current $\boldsymbol{\omega}$.

**Vector temporal difference (TD) target (envelope bootstrap).** Evaluate the selected pair with the *target* network $\theta^-$ to obtain a *vector* target

$$y = r(s,a) + \gamma Q\big(s', a^\star, \omega^\star; \theta^-\big) \in \mathbb{R}^m, \tag{7}$$

where the expectation over $s' \sim \mathcal{P}(\cdot \mid s,a)$ is approximated by sampling $s'$ from the replay buffer.

---

**Algorithm 3** Envelope Q-learning (EQL) with Preference-Guided Replay

---

**Require:** Discount factor $\gamma$, prioritized buffer $\mathcal{B}$, target period $C$, Dirichlet prior $\alpha$, preferences-per-sample $K$, minibatch size $B$

1: Initialize $\theta$; set $\theta^- \leftarrow \theta$

                                                    $\triangleright$ Interaction (Figure 6, steps 1–5)

2: **for** each environment step **do**

3:      observe $s$          $\triangleright$ (1)

4:      sample $\omega \sim \text{Dir}(\alpha)$          $\triangleright$ (2)

5:      choose $a$ by $\varepsilon$-greedy on $\omega^\top Q(s, a, \omega; \theta)$          $\triangleright$ (3)

6:      execute $a$; observe $r, s'$          $\triangleright$ (4)

7:      push $(s, a, r, s')$ into $\mathcal{B}$ with initial priority $p_{\max}$          $\triangleright$ (5)

8: **end for**

                                          $\triangleright$ Learning (Figure 6, steps 6–11)

9: **for** each gradient step **do**

10:      sample $\{(s_i, a_i, r_i, s_i')\}_{i=1}^B \sim \mathcal{B}$ by priority          $\triangleright$ (7)

11:      **for** $i = 1$ to $B$ **do**

12:          sample $\mathcal{W}_i = \{\omega_{ij}'\}_{j=1}^K$          $\triangleright$ (6), (8)

13:          $(a_i^\star, \omega_i^\star) \leftarrow \arg\max_{a', \omega' \in \mathcal{W}_i} \omega_i^\top Q(s_i', a', \omega'; \theta)$

14:          **Vector target:** $y_i \leftarrow r_i + \gamma Q(s_i', a_i^\star, \omega_i^\star; \theta^-)$          $\triangleright$ Equation (7)

15:          $\delta_i^A \leftarrow y_i - Q(s_i, a_i, \omega_i; \theta)$          $\triangleright$ vector TD

16:          $\delta_i^B \leftarrow \omega_i^\top y_i - \omega_i^\top Q(s_i, a_i, \omega_i; \theta)$          $\triangleright$ scalarized TD

17:      **end for**

18:      minimize $\mathcal{L}(\theta) = \frac{1}{B} \sum_i \left[ (1-\lambda)|\delta_i^B| + \lambda \|\delta_i^A\|_2^2 \right]$          $\triangleright$ Equations (8) to (10)

19:      update priorities in $\mathcal{B}$ using $\|\delta_i^A\|_1$ or $|\delta_i^B|$          $\triangleright$ Equation (12)

20:      **if** step mod $C = 0$ **then**

21:          $\theta^- \leftarrow \theta$

22:      **end if**

23: **end for**

---

**Losses.** The primary objective regresses the full vector target (cf. Eq. (6) in (Yang et al., 2019)):

$$\mathcal{L}^A(\theta) = \mathbb{E}_{(s,a,r,s'),\,\omega} \left[ \left\| y - Q(s, a, \omega; \theta) \right\|_2^2 \right]. \tag{8}$$

Because the optimal frontier contains many discrete extreme points (a nonsmooth landscape), an auxiliary *scalarized* loss improves optimization stability (cf. Eq. (7) in (Yang et al., 2019)):

$$\mathcal{L}^B(\theta) = \mathbb{E}_{(s,a,r,s'),\,\omega} \left[ \left\| \omega^\top y - \omega^\top Q(s, a, \omega; \theta) \right\|_2 \right]. \tag{9}$$

The neural network is trained by employing a simple homotopy:

$$\mathcal{L}(\theta) = (1-\lambda)\,\mathcal{L}^B(\theta) + \lambda\,\mathcal{L}^A(\theta), \qquad \lambda \in [0,1],\ \lambda \uparrow 1. \tag{10}$$

**Approximating the inner maximization.** The maximization over $\omega' \in \Delta^{m-1}$ in Equation (6) is approximated by sampling a small candidate set $\mathcal{W} = \{\omega_j'\}_{j=1}^K \subset \Delta^{m-1}$ (e.g., from a Dirichlet distribution) and computing

$$(a^\star, \omega^\star) \approx \arg \max_{a' \in \mathcal{A},\, \omega' \in \mathcal{W}} \omega^\top Q(s', a', \omega'; \theta). \tag{11}$$

Each transition is *relabeled* with multiple sampled preferences (hindsight preference relabeling), which couples learning across the preference space and greatly improves sample efficiency. The complete training loop—with preference sampling, hindsight relabeling, prioritized replay, and the envelope bootstrap—is summarized in Figure 6 and Algorithm 3.

**Replay priority.** Priorities can be derived from vector or scalarized TD errors, e.g.,

$$p \propto \left\| y - Q(s, a, \omega; \theta) \right\|_1 \quad \text{or} \quad p \propto \left| \omega^\top y - \omega^\top Q(s, a, \omega; \theta) \right|. \tag{12}$$

**Theory and intuition.** The *envelope Bellman operator* (induced by Equation (7)) has a unique fixed point and is a $\gamma$-contraction under a suitable metric; hence EQL converges in tabular settings (Yang et al., 2019).

**Geometric interpretation.** The selection of the envelope maximizer, seen in Equation (6), *backs up from* the upper convex hull of the next state returns. Define

$$\mathcal{V}(s') \triangleq \big\{ Q(s', a', \omega') : a' \in \mathcal{A}, \ \omega' \in \Omega \big\} \subset \mathbb{R}^m.$$

Since $\omega^\top(\cdot)$ is linear, maximizing it over a set equals maximizing it over that set's convex hull (support-function invariance):

$$\max_{a', \omega'} \omega^\top Q(s', a', \omega') \ = \ \max_{v \in \text{conv } \mathcal{V}(s')} \omega^\top v. \tag{13}$$

Consequently, Equation (6) selects a supporting extreme point of $\text{conv } \mathcal{V}(s')$, and the target vector in Equation (7) *bootstraps* from this convex envelope of the solution frontier; hence dominated trade-offs are not reinforced and EQL effectively targets the convex coverage set (CCS).[1]

**Adaptation.** At runtime, the trained policy $\pi_\omega$ can be executed with any desired preference vector $\omega$ without retraining.

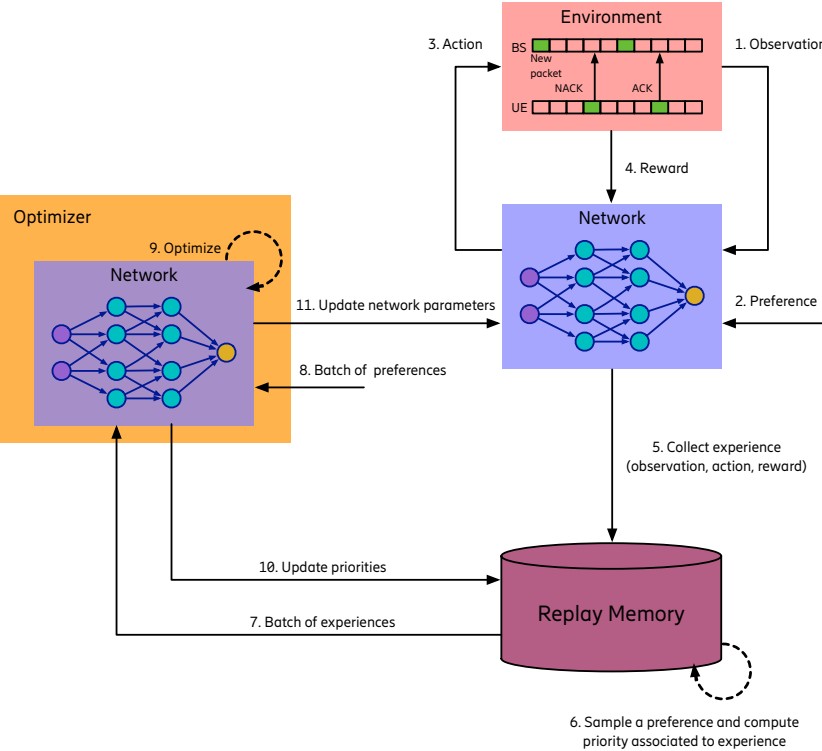

Figure 6: **Multi-objective RL with preference-guided optimization (EQL training loop). (1)** Observe state; **(2)** sample preference $\omega$; **(3)** act; **(4)** receive vector reward $r$; **(5)** store $(s, a, r, s')$; **(6)** memory samples auxiliary preferences and computes envelope-based priorities; **(7)** return prioritized batch; **(8)** return batch of preferences; **(9)** optimize using the vector target in Equation (7) and losses Equations (8) to (10); **(10)** update priorities; **(11)** update network parameters.

**EQL shortcomings:** Two design aspects limit the scalability of EQL Yang et al. (2019) in large state–action spaces. First, EQL relies on a singleton architecture Algorithm 3, where a single actor must explore the full joint state–action–preference space, leading to poor coverage and inefficient learning. Second, sample priorities are assigned only once at generation and never updated during training (unlike, e.g., (Horgan et al., 2018)), which slows convergence. Since RAN control problems involve vast state–action spaces, we propose a distributed EQL variant where multiple actors share the exploration load that leads to improved coverage and performance appendix F.5.

---

[1]With finite $\mathcal{A}$ (and discretized $\Omega$ in practice), $\mathcal{V}(s')$ is finite, so $\text{conv } \mathcal{V}(s')$ is a polytope and the maximum in equation 13 is attained at a vertex.

## F  DISTRIBUTED ENVELOPE Q-LEARNING

We propose D-EQL, a distributed MORL algorithm that extends EQL (Yang et al., 2019) with an APE-X–style distributed architecture (Horgan et al., 2018) for faster and more efficient exploration of the preference space. As in vanilla EQL, our method optimizes a single policy/value network over preferences for multiple competing objectives. Unlike the original setting, we employ learner–actor decoupling (Horgan et al., 2018) and distribute exploration over the preference space across multiple parallel actors. Specifically, we partition the preference simplex into subspaces and allocate different actors to explore different subspaces in parallel. While the partitioning of the preference space is inspired by (Xu et al., 2020), distributing exploration across actors improves coverage and exploration efficiency. Furthermore, we employ distributed prioritized experience replay with hindsight to improve sample efficiency: prioritized replay selects the most informative experiences at each training step, while hindsight relabeling increases reuse by updating priorities under multiple preferences $\omega$.

### F.1  D-EQL ARCHITECTURE

Our framework follows a scalable distributed (multi-objective) reinforcement learning architecture that decouples data collection, storage, and learning; see Figure 7. A set of CPU-based actors, each running a replica of the policy network, interact in parallel with multiple simulation environments to generate trajectories of state, action, reward, and next state tuples. Actors generate experiences by exploring *only* a partition the preference simplex. These experiences are first stored locally and then pushed to a sharded replay memory, where data is distributed either via load-balancing or fixed actor-to-shard mappings. Each shard operates as an independent replay buffer, enabling parallel writes and prioritized sampling. A GPU-based learner initially allocates a subspaces of the preference simplex to each actor for distributed exploration. The learner then periodically samples mini-batches from all shards, performs gradient updates on the policy network, and returns updated priorities to maintain efficient replay. Updated network weights are then broadcast to all actors, ensuring consistent synchronization across distributed processes. This design allows the system to scale efficiently with the number of actors and environments, achieving high-throughput of experience collection while preserving stability in training.

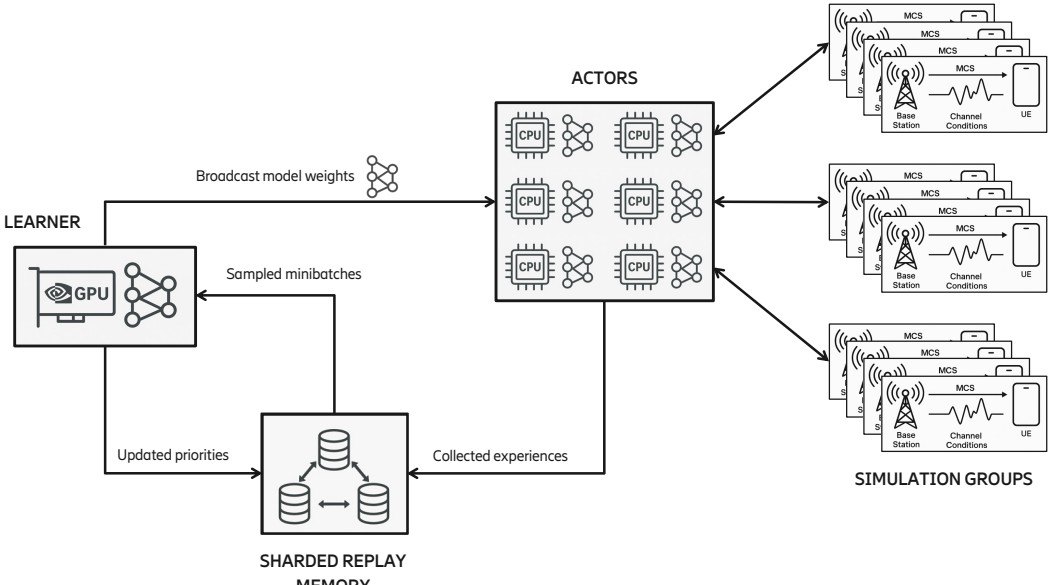

Figure 7: **Overview of the distributed (multi-objective) reinforcement learning architecture.** CPU-based actors interact in parallel with multiple simulation groups to generate trajectories, which are stored in a sharded replay memory, while exploring different subspaces of the preference simplex. A GPU-based learner samples mini-batches from the shards, updates the policy network, and broadcasts the updated weights back to the actors while sending updated priorities to the replay memory. This design enables scalable experience collection and stable policy learning.

## F.2 DISTRIBUTED ACTORS

We consider a MOMDP with state space $\mathcal{S}$, action set $\mathcal{A}$, transition kernel $\mathcal{P}(s' \mid s, a)$, discount factor $\gamma \in [0, 1)$, and vectorial reward $\boldsymbol{r}(s, a) \in \mathbb{R}^m$. Preferences lie on the probability simplex $\Omega = \Delta^{m-1}$. The learner maintains vectorial action-value functions $Q(s, a, \boldsymbol{\omega}; \boldsymbol{\theta}) \in \mathbb{R}^m$ (online) and $Q(s, a, \boldsymbol{\omega}; \boldsymbol{\theta}^-) \in \mathbb{R}^m$ (target) and periodically shares them with all actors. For any $\boldsymbol{\omega} \in \Omega$, we define the scalarization as

$$Q_{\boldsymbol{\omega}}(s, a; \boldsymbol{\theta}) \triangleq \boldsymbol{\omega}^\top Q(s, a, \boldsymbol{\omega}; \boldsymbol{\theta}) \in \mathbb{R}. \tag{14}$$

Each actor (i) generates experiences under an $\varepsilon$-greedy policy conditioned on a preference $\boldsymbol{\omega}$, (ii) computes a *scalarized* DDQN TD error to initialize prioritized experience replay priorities, and (iii) sends batched transitions and priorities to its assigned replay memory shard. Each actor is assigned to a stratum $\Omega_L^{(u)}$ from a simplex lattice (see Algorithm 6) and samples $\boldsymbol{\omega}$ uniformly in that simplex via barycentric weights (see Algorithm 7).

### BEHAVIOR POLICY AND DATA GENERATION

At the beginning of episode $e$, the actor draws a preference with support on the actor's stratum $\Omega_L^{(u)}$ (see Appendix F.4) and keeps it fixed:

$$\boldsymbol{\omega}^{(e)} \sim p_u(\cdot), \qquad \boldsymbol{\omega}_t \equiv \boldsymbol{\omega}^{(e)} \text{ for all } t \text{ in episode } e.$$

At the environment step $t \in \mathbb{N}$, the exploration rate is linearly annealed from $\varepsilon_{\max}$ to $\varepsilon_{\min}$ over $T_{\text{decay}}$ steps:

$$\varepsilon_t = \max\left\{\varepsilon_{\min}, \ \varepsilon_{\max} - \frac{\varepsilon_{\max} - \varepsilon_{\min}}{T_{\text{decay}}} t\right\}. \tag{15}$$

Here, $\varepsilon_{\max} \in (0, 1]$ is the initial exploration rate, $\varepsilon_{\min} \in [0, \varepsilon_{\max})$ is the floor, and $T_{\text{decay}} \in \mathbb{N}$ is the annealing horizon (after which $\varepsilon_t$ is clamped).

We define the greedy action under the fixed preference (using Equation 14) as

$$a_{\boldsymbol{\omega}}^\star(s) = \arg\max_{a \in \mathcal{A}} Q_{\boldsymbol{\omega}}(s, a; \boldsymbol{\theta}). \tag{16}$$

Let $\xi_t \sim \text{Unif}(0, 1)$ and let $\text{UnifAct}(\mathcal{A})$ denote a single uniform draw from $\mathcal{A}$ used only when exploring. As a result, the executed action becomes:

$$a_t = \begin{cases} \text{UnifAct}(\mathcal{A}), & \text{if } \xi_t < \varepsilon_t, \\ a_{\boldsymbol{\omega}}^\star(s_t), & \text{otherwise}. \end{cases} \tag{17}$$

Applying $a_t$ yields a transition $(s_t, a_t, \boldsymbol{r}_t, s_{t+1}, d_t)$ with a terminal flag $d_t \in \{0, 1\}$.

### INITIAL PRIORITY COMPUTATION

To initialize prioritized replay, the actor computes a *scalar* DDQN TD error using a *fresh* preference $\tilde{\boldsymbol{\omega}} \sim p_u(\cdot)$ (independent of the behavior preference) to diversify the priorities:

$$a_{\tilde{\boldsymbol{\omega}}}^\star(s_{t+1}) = \arg\max_{a' \in \mathcal{A}} \tilde{\boldsymbol{\omega}}^\top Q(s_{t+1}, a', \tilde{\boldsymbol{\omega}}; \boldsymbol{\theta}), \tag{18}$$

$$\delta_{\text{act}} = \underbrace{\tilde{\boldsymbol{\omega}}^\top \boldsymbol{r}_t}_{r_{\tilde{\omega}}} + \gamma(1 - d_t)\,\tilde{\boldsymbol{\omega}}^\top Q(s_{t+1}, a_{\tilde{\boldsymbol{\omega}}}^\star(s_{t+1}), \tilde{\boldsymbol{\omega}}; \boldsymbol{\theta}^-) - \tilde{\boldsymbol{\omega}}^\top Q(s_t, a_t, \tilde{\boldsymbol{\omega}}; \boldsymbol{\theta}), \tag{19}$$

$$p_{\text{init}} = |\delta_{\text{act}}| + \epsilon_0, \qquad \epsilon_0 \ll 1. \tag{20}$$

The pair $\big((s_t, a_t, \boldsymbol{r}_t, s_{t+1}, d_t), \ p_{\text{init}}\big)$ is buffered locally and flushed to the assigned replay shard.

### LOCAL BATCHING AND BATCHED COMMUNICATION

Let $u \in \{0, \dots, U - 1\}$ denote the actor id and $K$ be the number of replay shards. The actor accumulates transitions in a local circular buffer of capacity $C$ and, when full, sends the batch to a designated shard via a single remote procedure call (RPC). Shard selection uses the deterministic mapping

$$k(u) = u \bmod K. \tag{21}$$

The actor $u$ then transmits the batch $\{(s_n, a_n, \boldsymbol{r}_n, s_n', d_n), p_{\text{init},n}\}_{n=1}^C$ to shard $k(u)$.

---

**Algorithm 4** D-EQL Actor

---

**Require:** actor id $u$, shards $\{\text{SHARD}_k\}_{k=0}^{K-1}$, local buffer capacity $C$, discount factor $\gamma$, schedule
params $(\varepsilon_{\max}, \varepsilon_{\min}, T_{\text{decay}})$, stratum $\Omega_L^{(u)}$ with sampler $p_u(\cdot)$, online $Q(\cdot; \boldsymbol{\theta})$, target $Q(\cdot; \boldsymbol{\theta}^-)$
1: $k \leftarrow u \bmod K$           ▷ assigned shard, cf. Equation 21
2: $\mathcal{B} \leftarrow \varnothing, \mathcal{P} \leftarrow \varnothing, t \leftarrow 0$
3: **for** episode $e = 1, 2, \ldots$ **do**
4:    **Sample episodic preference:** $\boldsymbol{\omega}^{(e)} \sim p_u(\cdot)$; set $\boldsymbol{\omega}_t \equiv \boldsymbol{\omega}^{(e)}$ for this episode
5:    **while** episode not terminated **do**
6:      observe $s_t$
7:      compute $\varepsilon_t$ via the linear schedule Equation 15
8:      draw $\xi_t \sim \text{Unif}(0,1)$
9:      **if** $\xi_t < \varepsilon_t$ **then**
10:        $a_t \leftarrow \text{UnifAct}(\mathcal{A})$
11:      **else**
12:        $a_t \leftarrow \arg\max_{a \in \mathcal{A}} Q_{\boldsymbol{\omega}_t}(s_t, a; \boldsymbol{\theta})$      ▷ greedy map Equation 16
13:      **end if**
14:      execute $a_t$; observe $(\boldsymbol{r}_t, s_{t+1}, d_t)$
15:      **Sample priority preference:** $\tilde{\boldsymbol{\omega}} \sim p_u(\cdot)$
16:      $a^\star \leftarrow \arg\max_{a' \in \mathcal{A}} \tilde{\boldsymbol{\omega}}^\top Q(s_{t+1}, a', \tilde{\boldsymbol{\omega}}; \boldsymbol{\theta})$      ▷ selection Equation 18
17:      $\delta_{\text{act}} \leftarrow \tilde{\boldsymbol{\omega}}^\top \boldsymbol{r}_t + \gamma(1 - d_t)\tilde{\boldsymbol{\omega}}^\top Q(s_{t+1}, a^\star, \tilde{\boldsymbol{\omega}}; \boldsymbol{\theta}^-) - \tilde{\boldsymbol{\omega}}^\top Q(s_t, a_t, \tilde{\boldsymbol{\omega}}; \boldsymbol{\theta})$    ▷ TD error
  Equation 19
18:      $p_{\text{init}} \leftarrow |\delta_{\text{act}}| + \epsilon_0$          ▷ priority Equation 20
19:      append $\big((s_t, a_t, \boldsymbol{r}_t, s_{t+1}, d_t), p_{\text{init}}\big)$ to $(\mathcal{B}, \mathcal{P})$
20:      **if** $|\mathcal{B}| = C$ **then**
21:        $\text{ADDEXPERIENCES}\big(\text{SHARD}_k, \mathcal{B}, \mathcal{P}\big)$; reset $\mathcal{B}, \mathcal{P} \leftarrow \varnothing$
22:      **end if**
23:      $\text{PERIODICALLY}(\boldsymbol{\theta} \leftarrow \text{LEARNER.PARAMETERS}())$
24:      $t \leftarrow t + 1$
25:    **end while**
26: **end for**

---

### F.3 CENTRALIZED LEARNER

The learner (i) assembles prioritized mini-batches from $K$ replay shards, (ii) samples a mini-batch of preferences from a Dirichlet and forms a *Cartesian product* with the transitions, (iii) for each (transition, preference) pair computes an *envelope* DDQN target by maximizing over both actions and a finite set of supporting preferences, (iv) updates the online parameters $\boldsymbol{\theta}$ using a vector regression objective plus a cosine similarity term, (v) refreshes per-transition priorities on the shards, and (vi) periodically synchronizes the target network $\boldsymbol{\theta}^-$ and publishes the latest online parameters to actors.

MINI-BATCH ASSEMBLY FROM REPLAY SHARDS

Let assume shard $k \in \{0, \ldots, K-1\}$ stores $N_k$ items with priorities $\{p_{k,i}\}$, with a total running value $Z_k \triangleq \sum_{i=1}^{N_k} p_{k,i}^\alpha$, where $\alpha \in [0,1]$. A pair $(k, i)$ is sampled with probability

$$\Pr\big((k,i)\big) = \frac{Z_k}{\sum_{\ell=0}^{K-1} Z_\ell} \cdot \frac{p_{k,i}^\alpha}{Z_k} = \frac{p_{k,i}^\alpha}{\sum_\ell \sum_j p_{\ell,j}^\alpha}. \tag{22}$$

Given $N \triangleq \sum_k N_k$, experience replay importance weights $w_{k,i}$ (with exponent $\beta \in [0,1]$) are

$$w_{k,i} = \left(\frac{1}{N} \cdot \frac{1}{\Pr((k,i))}\right)^\beta \Big/ \max_{k',i'} \left(\frac{1}{N} \cdot \frac{1}{\Pr((k',i'))}\right)^\beta. \tag{23}$$

The learner queries the shards values $\{Z_k\}_{k=1}^K$, allocates per-shard batch sizes proportionally, fetches tuples (transition, index, $w_{k,i}$, shard id), and aggregates them into a transition batch of size $B$.

PREFERENCE SAMPLING

For each training step, we draw an i.i.d. mini-batch of preferences from a Dirichlet distribution:

$$\{\boldsymbol{\omega}_j\}_{j=1}^P \subset \Omega = \Delta^{m-1}, \qquad \boldsymbol{\omega}_j \overset{\text{i.i.d.}}{\sim} \text{Dir}(\boldsymbol{\alpha}), \qquad \boldsymbol{\alpha} \in (0,\infty)^m. \tag{24}$$

The choice $\boldsymbol{\alpha} = \mathbf{1}_m$ yields the uniform distribution on the simplex; $\boldsymbol{\alpha} < \mathbf{1}$ emphasizes corners, while $\boldsymbol{\alpha} > \mathbf{1}$ emphasizes the interior. To couple transitions and preferences, we form the *Cartesian product* index set

$$\mathcal{I} = \{1, \ldots, B\} \times \{1, \ldots, P\},$$

so every transition is paired with every preference. For $(i,j) \in \mathcal{I}$, we write $(s_i, a_i, \boldsymbol{r}_i, s_i', d_i, \boldsymbol{\omega}_j)$ and define the scalarization $Q_{\boldsymbol{\omega}_j}(s, a; \boldsymbol{\theta}) = \boldsymbol{\omega}_j^\top Q(s, a, \boldsymbol{\omega}_j; \boldsymbol{\theta})$.

ENVELOPE DDQN SELECTION AND VECTOR TARGET

The EQL's envelope backup requires maximizing over both actions and supporting preferences. Directly optimizing over $\Omega$ is intractable, so we *approximate* the inner maximization by searching over a sampled set $\mathcal{W} = \{\boldsymbol{\omega}_j\}_{j=1}^P$. For each pair $(i,j) \in \mathcal{I}$, we get:

$$(a_{i,j}^\star, \tilde{\boldsymbol{\omega}}_{i,j}^\star) = \underset{a' \in \mathcal{A}, \, \boldsymbol{\omega}' \in \mathcal{W}}{\arg\max} \ \boldsymbol{\omega}_j^\top Q(s_i', a', \boldsymbol{\omega}'; \boldsymbol{\theta}), \tag{25}$$

$$\boldsymbol{y}_{i,j} = \boldsymbol{r}_i + \gamma(1 - d_i) Q(s_i', a_{i,j}^\star, \tilde{\boldsymbol{\omega}}_{i,j}^\star; \boldsymbol{\theta}^-) \in \mathbb{R}^m. \tag{26}$$

Thus each *query preference* $\boldsymbol{\omega}_j$ selects a *supporting* preference $\tilde{\boldsymbol{\omega}}_{i,j}^\star \in \mathcal{W}$ and action $a_{i,j}^\star$ that together realize the envelope along direction $\boldsymbol{\omega}_j$. The online prediction is $Q_{\text{pred},i,j} = Q(s_i, a_i, \boldsymbol{\omega}_j; \boldsymbol{\theta}) \in \mathbb{R}^m$.

TRAINING LOSS

For each $(i,j) \in \mathcal{I}$, we define:

$$\mathcal{L}_{\text{mmse}}(i,j; \boldsymbol{\theta}) = \big\| \boldsymbol{y}_{i,j} - Q(s_i, a_i, \boldsymbol{\omega}_j; \boldsymbol{\theta}) \big\|_2^2, \tag{27}$$

$$\mathcal{L}_{\text{cos}}(i,j; \boldsymbol{\theta}) = 1 - \frac{\boldsymbol{\omega}_j^\top Q(s_i, a_i, \boldsymbol{\omega}_j; \boldsymbol{\theta})}{\|\boldsymbol{\omega}_j\|_2 \, \|Q(s_i, a_i, \boldsymbol{\omega}_j; \boldsymbol{\theta})\|_2}. \tag{28}$$

With tradeoff $\lambda \geq 0$ and PER weights $w_{k,i}$ tied to the *transition* (replicated over its $P$ preference copies), the learner minimizes

$$\mathcal{L}(\boldsymbol{\theta}) = \frac{1}{BP} \sum_{i=1}^B \sum_{j=1}^P w_{k(i),\, idx(i)} \Big( \mathcal{L}_{\text{mmse}}(i,j; \boldsymbol{\theta}) + \lambda \, \mathcal{L}_{\text{cos}}(i,j; \boldsymbol{\theta}) \Big), \tag{29}$$

where $k(i)$ and $idx(i)$ are the shard id and local index of transition $i$.

PRIORITY REFRESH (LEARNER SIDE)

To refresh priority weights, we first compute scalarized residuals per pair $(i,j)$,

$$\delta_{i,j} = \boldsymbol{\omega}_j^\top \big( \boldsymbol{y}_{i,j} - Q(s_i, a_i, \boldsymbol{\omega}_j; \boldsymbol{\theta}) \big), \tag{30}$$

then aggregate to a single priority per *original transition* $i$,

$$p_{\text{new}}(i) = \max_{1 \leq j \leq P} |\delta_{i,j}| + \epsilon_0, \qquad \epsilon_0 > 0, \tag{31}$$

and return $\big(\text{indices}(i), p_{\text{new}}(i)\big)$ to the corresponding shards to update PER totals.

TARGET UPDATES AND PARAMETER BROADCAST

Every $C_{\text{tgt}}$ steps, the learner updates the target network either by a hard copy

$$\boldsymbol{\theta}^- \leftarrow \boldsymbol{\theta} \tag{32}$$

or a soft update with factor $\tau \in (0, 1]$:

$$\boldsymbol{\theta}^- \leftarrow (1 - \tau) \boldsymbol{\theta}^- + \tau \boldsymbol{\theta}. \tag{33}$$

Every $C_{\text{push}}$ steps, the latest online parameters are published to the shared model used by all actors.

---

**Algorithm 5** D-EQL Learner

---

**Require:** shards $k \in \{0, \ldots, K-1\}$, discount $\gamma$, PER exponents $(\alpha, \beta)$, target period $C_{\text{tgt}}$, push period $C_{\text{push}}$, preference batch size $P$, tradeoff $\lambda$, Dirichlet parameter $\boldsymbol{\alpha}$
1: Initialize online $\boldsymbol{\theta}$; set target $\boldsymbol{\theta}^- \leftarrow \boldsymbol{\theta}$
2: **while** training **do**
3:     Query $\{Z_k\}$; allocate per-shard batch sizes; fetch prioritized transitions with $(\text{indices}(i),\ w_{k(i),\ idx(i)})$          ▷ Equation 22–Equation 23
4:     Sample preferences $\{\boldsymbol{\omega}_j\}_{j=1}^P \overset{\text{i.i.d.}}{\sim} \text{Dir}(\boldsymbol{\alpha})$          ▷ Equation 24
5:     Form Cartesian product $\mathcal{I} = \{1..B\} \times \{1..P\}$ (replicate transitions across all $\boldsymbol{\omega}_j$)
6:     **for all** $(i, j) \in \mathcal{I}$ **do**
7:         $(a_{i,j}^\star, \tilde{\boldsymbol{\omega}}_{i,j}^\star) \leftarrow \arg\max_{a' \in \mathcal{A},\ \tilde{\boldsymbol{\omega}} \in \{\boldsymbol{\omega}_1, \ldots, \boldsymbol{\omega}_P\}}\ \boldsymbol{\omega}_j^\top Q(s_i', a', \tilde{\boldsymbol{\omega}}; \boldsymbol{\theta})$          ▷ Equation 25
8:         $\boldsymbol{y}_{i,j} \leftarrow \boldsymbol{r}_i + \gamma(1 - d_i)\, Q(s_i', a_{i,j}^\star, \tilde{\boldsymbol{\omega}}_{i,j}^\star; \boldsymbol{\theta}^-)$          ▷ Equation 26
9:     **end for**
10:    Compute $\mathcal{L}(\boldsymbol{\theta})$ via Equation 29; take a gradient step on $\boldsymbol{\theta}$
11:    For each $i \in \{1..B\}$, compute $p_{\text{new}}(i)$ via Equation 30–Equation 31; send updates to shards
12:    **if** step mod $C_{\text{tgt}} = 0$ **then**
13:        Update $\boldsymbol{\theta}^-$ via Equation 32 or Equation 33
14:    **end if**
15:    **if** step mod $C_{\text{push}} = 0$ **then**
16:        Publish $\boldsymbol{\theta}$ to actors
17:    **end if**
18: **end while**

---

**Notes.** *(i)* The envelope selection (Equation 25) is the finite-set approximation of the bi-level inner maximization over preferences; letting $P \uparrow \infty$ densifies the approximation. *(ii)* The Cartesian product ensures that *every* transition is trained under *every* sampled preference each step (dense supervision). *(iii)* Priorities are defined per transition by aggregating scalarized TD residuals over $P$ preference replicas.

**Communication summary.** The learner *pulls* batches proportional to $Z_k$, *pushes* refreshed priorities $p_{\text{new}}$ (updating shard totals), and periodically *broadcasts* the latest online parameters. Preference expansion couples updates across $\Omega$, while DDQN selection/evaluation preserves stability.

### F.4 STRATIFIED SAMPLING ON THE PROBABILITY SIMPLEX

We want to sample preferences $\boldsymbol{\omega} \in \Omega := \Delta^{m-1}$ so that *all regions* of the simplex are adequately covered during data generation, thereby reducing variance and avoiding mode collapse toward a few scalarizations.

**Baseline (no stratification).** When no partition is imposed, we draw independent and identically distributed (i.i.d.) preferences from a Dirichlet distribution,

$$\boldsymbol{\omega} \sim \text{Dir}(\boldsymbol{\alpha}), \qquad \boldsymbol{\alpha} \in (0, \infty)^m. \tag{34}$$

Choosing $\boldsymbol{\alpha} = \mathbf{1}_m$ yields the uniform distribution on $\Omega$.

#### F.4.1 DETERMINISTIC EQUAL-VOLUME STRATA VIA A SIMPLEX LATTICE

We partition $\Omega$ into congruent $(m-1)$-simplices by using a barycentric lattice with resolution $L \in \mathbb{N}$. We define the lattice vertices as:

$$\mathcal{V}_L := \left\{ \frac{\boldsymbol{k}}{L} \in \Omega\ :\ \boldsymbol{k} = (k_1, \ldots, k_m) \in \mathbb{N}^m,\ \sum_{i=1}^m k_i = L \right\}, \tag{35}$$

and we consider a fixed permutation $\pi$ of $\{1, \ldots, m\}$. For each *base* point $\boldsymbol{k} \in \mathbb{N}^m$ with $\sum_i k_i = L - 1$, we form the $m$ lattice points as:

$$\boldsymbol{v}_0 = \frac{\boldsymbol{k}}{L}, \quad \boldsymbol{v}_r = \frac{\boldsymbol{k} + \boldsymbol{e}_{\pi(1)} + \cdots + \boldsymbol{e}_{\pi(r)}}{L}, \quad r = 1, \ldots, m-1, \tag{36}$$

---

**Algorithm 6** Simplex–Lattice Stratification (build strata at resolution $L$)

---

**Require:** dimension $m$, resolution $L$, a fixed permutation $\pi$ of $\{1, \ldots, m\}$
1: $\mathcal{S} \leftarrow \varnothing$                 ▷ list of strata (each as $m$ vertices in $\mathbb{R}^m$)
2: **for all** $\boldsymbol{k} \in \mathbb{N}^m$ with $\sum_{i=1}^m k_i = L - 1$ **do**
3:      $\boldsymbol{v}_0 \leftarrow (\boldsymbol{k})/L$
4:      **for** $r = 1$ to $m - 1$ **do**
5:          $\boldsymbol{v}_r \leftarrow (\boldsymbol{k} + \boldsymbol{e}_{\pi(1)} + \cdots + \boldsymbol{e}_{\pi(r)})/L$
6:      **end for**
7:      append $\mathrm{conv}\{\boldsymbol{v}_0, \ldots, \boldsymbol{v}_{m-1}\}$ to $\mathcal{S}$
8: **end for**
9: **return** $\mathcal{S}$               ▷ $|\mathcal{S}| = L^{m-1}$ equal-volume strata

---

**Algorithm 7** Sample uniformly from a stratum $\Omega_L(\boldsymbol{k})$

---

**Require:** vertices $\{\boldsymbol{v}_0, \ldots, \boldsymbol{v}_{m-1}\}$ of $\Omega_L(\boldsymbol{k})$
1: draw $\mathbf{z} \sim \mathrm{Dir}(\mathbf{1}_m)$
2: **return** $\boldsymbol{\omega} = \sum_{r=0}^{m-1} z_r \boldsymbol{v}_r$              ▷ uniform in the stratum

---

and define the micro-simplex (stratum):

$$\Omega_L(\boldsymbol{k}) := \mathrm{conv}\{\boldsymbol{v}_0, \boldsymbol{v}_1, \ldots, \boldsymbol{v}_{m-1}\} \subset \Omega. \tag{37}$$

The collection $\{\Omega_L(\boldsymbol{k}) : \boldsymbol{k} \in \mathbb{N}^m, \sum_i k_i = L - 1\}$ tiles $\Omega$ into $L^{m-1}$ equal-volume strata.

#### F.4.2 UNIFORM SAMPLING WITHIN A STRATUM

Let $\Omega_L(\boldsymbol{k}) = \mathrm{conv}\{\boldsymbol{v}_0, \ldots, \boldsymbol{v}_{m-1}\}$ be any stratum. We draw barycentric weights $\mathbf{z} \sim \mathrm{Dir}(\mathbf{1}_m)$ and map affinely:

$$\boldsymbol{\omega} = \sum_{r=0}^{m-1} z_r \boldsymbol{v}_r \in \Omega_L(\boldsymbol{k}). \tag{38}$$

This yields a sample *uniform* in the stratum.

#### F.4.3 ASSIGNING STRATA TO ACTORS

Index the $L^{m-1}$ strata in a fixed order as $\{\Omega_L^{(u)}\}_{u=0}^{U-1}$ with $U = L^{m-1}$ (or group them when $U$ exceeds the number of actors). Actor $u$ repeatedly samples $\omega \in \Omega_L^{(u)}$ via Algorithm 7, ensuring non-overlapping coverage across actors.

#### F.4.4 DISCUSSION AND ALTERNATIVES

**Coverage and variance.** Compared to i.i.d. Dirichlet sampling shown in Equation 34, the lattice partition yields systematic coverage of the entire simplex and reduces estimator variance by ensuring that each subregion is represented.

**Resolution.** Larger $L$ gives finer strata ($L^{m-1}$ pieces) and smoother coverage at the cost of more partitions to manage.

**Clustering alternative.** When equal-volume strata are unnecessary, a simple alternative is to draw a large pilot set $\{\omega^{(n)}\}_{n=1}^{N_0} \sim \mathrm{Dir}(\boldsymbol{\alpha})$ and run $k$-means on the $(m-1)$-dimensional simplex (with cosine or Euclidean distance); the Voronoi cells of the cluster centers define strata. Sampling within a cell can be done by re-running Dirichlet draws and accepting points whose nearest center matches the cell (approximately uniform within each cell).

### F.5 EXPERIMENTS

#### F.5.1 ENVIRONMENTS AND SETUP

We evaluate our approach using two well-established MORL benchmarks: deep sea treasure (DST) and fruit tree navigation (FTN). Both environments are widely used in the literature (Yang et al.,

2019; Basaklar et al., 2023), providing standardized testbeds for assessing Pareto front coverage and preference generalization.

The DST environment is a grid-world wherein an agent controls a submarine that must navigate from the surface to collect one of several treasures placed at different depths. Each treasure yields a two-dimensional reward: a positive value and a negative time penalty. The task is inherently multi-objective, requiring the agent to balance collecting high-value treasures against minimizing travel time. The Pareto front is well understood and serves as a reliable benchmark for coverage and accuracy.

The FTN environment generalizes this idea to a tree-structured setting. Starting from the root, the agent makes sequential decisions until it reaches a leaf node, where it receives a multi-dimensional reward corresponding to the chosen fruit. The tree depth controls task complexity. At depth five, the agent makes five sequential decisions; at depth seven, the number of possible outcomes grows exponentially, producing a much larger and more diverse Pareto front. This makes FTN with higher depth a significantly more challenging benchmark, particularly for algorithms that must adapt to unseen preferences or maintain wide coverage.

In all experiments, algorithms are trained across a range of randomly sampled linear preference vectors, following the setup in Yang et al. (2019). At test time, additional preference vectors are sampled to assess generalization. All results are averaged over multiple random seeds to account for variance.

### F.5.2 METRICS AND RESULTS

We evaluate performance using three widely adopted metrics in MORL:

1. **Coverage Ratio F1 (CRF1):** A harmonic mean of precision and recall that captures both accuracy and coverage of the Pareto front.

2. **Hypervolume:** The volume dominated by the obtained solutions with respect to a reference point, reflecting both the quality and diversity of the Pareto front.

3. **Sparsity:** The average distance between neighboring solutions, indicating how uniformly the Pareto front is covered.

Table 7: We compare the performance of the proposed distributed Q-learning algorithm with the original Q-learning approach Yang et al. (2019) and Basaklar et al. (2023) in terms of CFR1, hypervolume and sparsity, showing superior performance in more complex scenarios.

|  | Deep Sea Treasure | | | Fruit Tree Nav. (d=5) | | Fruit Tree Nav. (d=7) | |
| --- | --- | --- | --- | --- | --- | --- | --- |
|  | CRF1 | Hyperv. | Sparsity | CRF1 | Hyperv. | CRF1 | Hyperv. |
| Yang et al. (2019) | 0.994 | 227.39 | 2.62 | 1.0 | 6920.58 | 0.819 | 6395.27 |
| Basaklar et al. (2023) | 1.0 | 241.73 | 1.14 | 1.0 | 6920.58 | 0.920 | 11 419.58 |
| D-EQL (ours) | **1.0** | **241.73** | **1.14** | **1.0** | **6920.58** | **1.0** | **12110.74** |

Table 7 compares D-EQL with prior approaches. On the simpler DST domain, all methods achieve near-perfect coverage. Our method, on pair with Basaklar et al. (2023), attains CRF1 = 1.0 and simultaneously achieves the highest hypervolume and lowest sparsity, indicating comprehensive and well-distributed solutions along the Pareto front.

On FTN with depth five, performance saturates across all methods with perfect coverage and identical hypervolume, reflecting the relative simplicity of this setting.

The advantage of D-EQL becomes evident in the more complex FTN with depth seven. Prior methods show a notable drop in coverage (CRF1 = 0.819 for Yang et al. (2019) and 0.92 for Basaklar et al. (2023)), whereas D-EQL maintains perfect coverage (CRF1 = 1.0). Moreover, D-EQL achieves the highest hypervolume (12110.74), showing 22.1% and 8.69% improvements over Yang et al. (2019) and Basaklar et al. (2023), respectively, and demonstrating broader Pareto front coverage and superior solution diversity. These results show that our distributed training scheme preserves accuracy while scaling effectively to environments with exponentially growing outcome spaces.

Overall, D-EQL achieves state-of-the-art performance: it matches existing methods on simpler tasks and clearly outperforms them in complex scenarios, highlighting the benefits of distributed training for multi-objective reinforcement learning. This makes D-EQL a more suitable for handling the vast dimensions of state-action spaces in RAN control functions.

### F.5.3 HYPERPARAMETERS

The hyperparameters used in D-EQL training for DST and FTN environment are listed in Table 8.

Table 8: Hyperparameters used for D-EQL on deep sea treasure and fruit tree navigation.

| Hyperparameter | DST | FTN |
|---|---|---|
| **Model** | | |
| Hidden feature (units per layer) | 256 | 512 |
| Activation | SiLU | SiLU Elfwing et al. (2018) |
| Number of layers | 3 | 3 |
| **Actor** | | |
| Number of actors | 10 | 10 |
| $\epsilon$-greedy (linear, start $\rightarrow$ final) | $0.8 \rightarrow 0.1$ | $0.8 \rightarrow 0.1$ |
| Anneal timesteps | $1 \times 10^6$ | $1 \times 10^6$ |
| Local buffer capacity | 125 | 125 |
| Max environment step | $1 \times 10^6$ | $1 \times 10^6$ |
| **Learner** | | |
| Learning rate | $3.5 \times 10^{-4}$ | $3.5 \times 10^{-4}$ |
| Target update period (gradient step) | 1 | 1 |
| Model sync period (gradient step) | 250 | 250 |
| Discount factor $\gamma$ | 0.99 | 0.99 |
| Prefetched batches | 16 | 16 |
| Transition batch size | 128 | 128 |
| Preference batch size | 128 | 128 |
| **Replay Memory (PER)** | | |
| Number of shards | 1 | 1 |
| Capacity | $5 \times 10^5$ | $5 \times 10^5$ |
| Priority exponent $\alpha$ | 0.7 | 0.7 |
| Importance sampling $\beta$ (linear, start $\rightarrow$ final) | $0.4 \rightarrow 1.0$ | $0.4 \rightarrow 1.0$ |
| $\beta$ anneal timesteps | $2 \times 10^4$ | $2 \times 10^4$ |

## G  CASE STUDY

We design a controller agent for LA, a crucial functionality of modern wireless communication systems that employs adaptive coding and modulation to optimize the spectral efficiency of the radio link between transmitter and receiver. By adopting a MORL approach, the LA controller agent can adjust transmission parameters to meet connectivity service intents expressed in terms of data rate, reliability, and latency requirements for individual users.

### G.1  LINK ADAPTATION

LA adapts the modulation order and code rate of individual packet transmissions to match the capacity of the radio link capacity, given the radio link state. The LA parameters are encoded into a unique value, referred to as MCS index in 3GPP (2025e), that is provided to the receiver for packet decoding.

The 3GPP 5$^{th}$ Generation (5G) New Radio (NR) system, rely on an OLLA approach inspired to Pedersen et al. (2007) to maximize the link spectral efficiency while adhering to a predefined BLER target using receiver-side channel state information (CSI), such as channel quality indicator (CQI) 3GPP (2025e), and hybrid automatic repeat request (HARQ) feedback– a 1-bit information indicating whether a prior packet transmission was successful or not. While this approach suits best-effort traffic, its reliance on long communication sessions to converge makes is suboptimal to address connectivity service intents under more general conditions, such as short bursty traffic, fast channel aging, medium-high user mobility, etc.

A MORL approach instead enables to dynamic LA toward selection transmission parameters that best align with different service intents. For example, selecting MCS conservatively—e.g., lower modulation orders such as Quadri-Phase Shift Keying (QPSK) or reduced code rates—favor robustness by lowering the probability of decoding errors. This enables to achieve highly reliable transmissions at the cost of throughput, since more time-frequency resource element (RE) are required per information bit. Conversely, an aggressive MCS selection can push spectral efficiency closer to or even beyond the instantaneous link capacity, exploiting retransmissions to increase data rate and reduce latency for best-effort traffic. However, overly aggressive choices may lead to excessive retransmissions and throughput degradation. By explicitly balancing these conflicting objectives, MORL allows LA to adapt beyond fixed BLER-driven policies, supporting a wider range of connectivity intents.

### G.2  MOMDP DESIGN FOR LINK ADAPTATION

Our goal is to train a single pareto efficient uniform model (PEUMO) for LA to learn the Pareto frontier outlining the optimal trade-off between the utilization of radio resources and the amount of information bits delivered by a packet transmission.

As LA and HARQ operate on a per-user equipment (UE) and per-packet transmission basis, we formulate this problem as an episodic MOMDP $\mathcal{M} = \langle \mathcal{S}, \mathcal{A}, p, \boldsymbol{r}, \Omega, \gamma, \rho_0 \rangle$, where $\mathcal{S}$ denotes the state space, $\mathcal{A}$ the action space, $p(s' \mid s, a)$ the transition dynamics, $\boldsymbol{r} : \mathcal{S} \times \mathcal{A} \to \mathbb{R}^K$ a multi-dimensional reward vector, $\Omega \subseteq \mathbb{R}^K$ the reward preference space, $\gamma \in [0, 1)$ a discount factor, and $\rho_0$ the initial state distribution.

An episode models the lifespan of a UE packet in the HARQ process—from its first transmission to either a successful reception or the packet being dropped upon $N$ transmission attempts, as illustrated in Figure 8. This enables us to train a single RL policy from the collective experience generated by any UEs across the network. A transition in the episode represents the duration of a packet transmission in the HARQ process, from the selection of LA parameters (i.e., the action) to the reception of the associated HARQ feedback, i.e., an positive acknowledgment (ACK) or negative acknowledgment (NACK) for successful or failed transmission, respectively. For instance, the 3GPP 5G NR system, used in our evaluations, supports at most four packet retransmissions. Hence, the episode length $N$ may range from one to five steps. Each step is characterized by a state, an action, and an associated reward and preference vectors, as presented next.

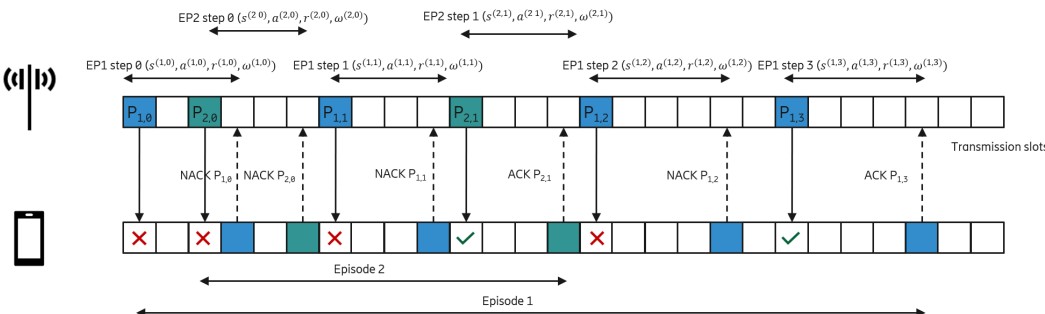

Figure 8: Example of MOMDP episodes modelling the downlink LA and HARQ process. Here, the notation $P_{i,n}$ denotes the $n$-th transmission attempt of the $i$-th data packet.

### G.3 ACTION SPACE

The action space consists of the set of MCS index values supported by a communication standard, i.e., $\mathcal{A} = \{a_m \mid a_m = m, m = 0, \dots, M - 1\}$. Therefore, an action $a_m \in \mathcal{A}$ implicitly provides a combination of modulation order, code rate, and spectral efficiency to be used to transmit a packet. The 5G NR system used in our evaluations in Section 7 supports $M = 28$ MCS index values as specified in Table 5.1.3.1-1 or Table 5.1.3.1-2 in 3GPP (2025e), corresponding to modulation orders up to 64QAM and 256QAM, respectively.

For a new packet transmission, the selection of an MCS index, combined with the time-frequency resources allocated by a scheduler, determines the amount of information bits, i.e., the TBS, to be transmitted. Packet re-transmissions, however, reuse the TBS value of the original transmission, as no new information is transmitted. A packet re-transmission, however, may occur with a different MCS index therefore resulting in possibly a different amount of radio resources.

### G.4 REWARD VECTOR AND PREFERENCE SPACE

We design a two-dimensional reward function $\boldsymbol{r} = [r_1, r_2]^\top \in \mathcal{S} \times \mathcal{A} \in \mathbb{R}^2$ with two competing components: $r_1$ representing the amount of information bits successfully carried by a packet; and $r_2$ denoting the cost, in terms of time-frequency resource, incurred in each individual transmission of the packet. Specifically, for each transmission attempt $n$ of a packet, we define the reward function as

$$
\boldsymbol{r}^{(n)}(s, a) = \begin{cases}
\begin{bmatrix} 0 \\ -\frac{N_{RE}^{(n)}}{N_{RE}^{\max}} \end{bmatrix} & \text{if transmission fails at } n-\text{th attempt,} \\[3ex]
\begin{bmatrix} \frac{TBS}{N_{RE}^{\max}} \\ -\frac{N_{RE}^{(n)}}{N_{RE}^{\max}} \end{bmatrix} & \text{if transmision succeeds at } n-\text{th attempt,}
\end{cases}
\tag{39}
$$

where the TBS and $N_{RE}^{(n)}$ denote the number of information bits carried by the packet and the number of RE used for the $n$-th transmission attempt, respectively, and $N_{RE}^{\max}$ is the maximum number of REs available, given the system bandwidth. Scaling the reward components by $N_{RE}^{\max}$ has a twofold purpose: Firstly, it keeps each component within similar range of values, while preserving the functional relation between the MCS index selected to transmit TBS information bits and the required number of time-frequency RE. This relation is specified by communication standards, as in the 3$^{\text{rd}}$ Generation Partnership Project (3GPP) technical specification (TS) 38.211 3GPP (2025d). Secondly, it makes the reward design agnostic to the system bandwidth, with $\frac{TBS}{N_{RE}^{\max}}$ representing the spectral efficiency for transmitting TBS bits using the entire system bandwidth. This allows us to employ domain randomization in training (cf. appendix H.1) to improve model generalization over the RAN environment.

Therefore, for each packet transmission attempt $n$, the first reward component takes value $r_1^{(n)} = \frac{TBS}{N_{RE}^{\max}}$ is the transmission is successful or $r_1^{(n)} = 0$ otherwise. The second reward component, on the

other hand, always indicates the resource cost incurred at the current transmission attempt $n$, i.e., $r_2^{(n)} = -\frac{N_{RE}^{(n)}}{N_{RE}^{\max}}$, regardless of whether the transmission attempt succeeds or fails.

### G.5 STATE DESIGN

A key goal of our design is to achieve model generalization across diverse RAN environments, enabling a single MORL model to operate reliably under different deployments and radio conditions. To this end, we construct a rich state space $\mathcal{S} \subseteq \mathbb{R}^K$ and apply domain randomization in training Igl et al. (2019). To model the state space $\mathcal{S}$ for link adaptation, we follow Demirel et al. (2025) which considers a deep Q-network (DQN) approach for LA with a single, fixed reward design based on the link spectral efficiency. In particular, we model $\mathcal{S}$ using two types of features: (a) semi-static information characterizing the network deployment surrounding the UE; (b) and information describing observable link dynamics relevant to infer LA parameters.

Semi-static information characterizing the network deployment may include, for instance, deployment type (e.g., rural, urban, dense urban, etc.), location, orientation, relationships among network sites or radio cells, as well as technology configurations, such as whether the system operated in time-duplex or full duplex mode, carrier frequency, system bandwidth, transmit power, antenna array type, etc. On the other hand, information characterizing the dynamics of LA consist of real-time observation (measured in a milliseconds timescale), such as channel state information, HARQ feedback, measurement of path loss, data buffer state, historical actions, and more. We refer to Demirel et al. (2025) for a complete description of the state features.

## H    EXTENDED EXPERIMENTAL EVALUATION

This Appendix extends the discussion and empirical evaluation presented in Section 7 with additional results. We organized the material as follows: Appendix H.1 and H.2 describe the network simulator environment and training setup for the MORL controller agent; Appendix H.3 extend the analysis of the controller agent presented in Section 7.1, including an additional scenario with two communication services. Finally, Appendix H.4 extends our analysis of the intent fulfillment loop.

### H.1    NETWORK SIMULATOR ENVIRONMENT

We train and evaluate the MORL controller agent using a high-fidelity, event-driven system-level simulator compliant with the 3GPP 5G NR specifications. Each rollout simulation models a heterogeneous multi-cell RAN operating in time division duplexing (TDD) mode with single-user multiple input multiple output (SU-MIMO) transmission. The carrier frequency is set to 3.5 GHz, and the physical layer follows the orthogonal frequency division multiplexing (OFDM) numerology $\mu = 0$ specified in 3GPP TS 38.211 (cf. Table 4.2-1 3GPP (2025d)).

To improve model generalization across diverse RANdeployments and radio enviroments, we apply domain randomization across multiple network characteristics, summarized in Table 9. Each simulation consists of three tri-sector radio sites, randomly configured as either conventional multiple input multiple output (MIMO) or massive multiple input multiple output (mMIMO), with antenna attributes defined in Table 9. Site-level parameters such as location, cell radius, system bandwidth, and downlink transmit power are also randomized by sampling values from the same parameter set.

The training scenario is further diversified by randomizing cell load, traffic type, UEs, and receiver configuration. UEs are generated with a mixture of full buffer (FB) and enhanced mobile broadband (eMBB) traffic, randomly placed in the simulated area according to one of the indoor/outdoor probability distributions in Table 9. Each eMBB UE generates traffic with variable packet size and inter-arrival times, modeled using empirical distributions derived from field measurement campaigns.

Finally, individual UEs are randomized in terms of antenna configuration, mobility (speed), and receiver implementation. The latter accounts for manufacturer-specific differences in hardware (e.g., antenna arrays and chipsets) and internal algorithms (e.g., CSI estimation), which influence perceived radio conditions. This randomized environment ensures that the MORL controller agent is trained under various realistic network conditions, thus improving its ability to generalize to unseen scenarios.

### H.2    TRAINING SETUP

We train the MORL LA controller agent using our D-EQL algorithm, described in Appendix F, with a single GPU and 560 CPU cores. The learner uses Adam optimizer (Kingma & Ba, 2017) with a learning rate of $5 \times 10^{-5}$, weight decay of $0.02/512$, and default momentum terms $(\beta_1, \beta_2) = (0.9, 0.999)$, and mean squared error (MSE) loss. He initialization is used for all network parameters. A soft target update policy is applied with an update factor of 0.001 and a period of one timestep. The model synchronization period is 200 gradient iterations, and training begins after 50,000 timesteps. To reduce communication overhead between the learner and replay memory, 16 batches are prefetched per cycle. Experience and preference batches contain 512 and 128 samples, respectively.

The actor subsystem consists of 40 CPU-based rollout workers, each interacting with 14 parallel simulations (one CPU core per simulation), resulting in efficient experience generation. Each actor collects about 112 samples per second, for a total of roughly 279 million over the training horizon. The learner processes about 27,500 samples per second for gradient updates. Each actor maintains a local buffer of 2,500 samples and follows a linear epsilon-greedy strategy, decaying $\epsilon$ from 0.8 to 0.05 over 5.5 million timesteps. The agent operates with a discount factor of 1.0. Training throughput is about 53.8 batches per second, with each batch containing 65,536 samples.

Replay memory is organized as a single module with four independent shards, each capable of storing four million samples. Each shard has a fixed communication path to a designated learner shard, minimizing cross-shard delays. Prioritized experience replay is employed with parameters $\alpha = 0.6$ and $\beta = 0.4$ to improve sample efficiency. In total, the system runs 11,200 simulations under different random seeds to ensure reproducibility across diverse network conditions. Communication details of the distributed system are further discussed in Appendix I.

Table 9: RAN environment simulation parameters for domain randomization during training.

| Parameter | Value range | Description |
|---|---|---|
| Duplexing type | TDD | Fixed |
| Carrier frequency | 3.5 GHz | Fixed |
| Deployment type | 3-site 9-sector | |
| Site type | {MIMO, mMIMO} | Randomized |
| Antenna array | 1x2x2 MIMO (4) | Fixed |
| | 8x4x2 mMIMO (64) | Fixed |
| Cell radius | {166, 300, 600, 900, 1200} m | Randomized |
| Bandwidth | {20, 40, 50, 80, 100} MHz | Randomized |
| Number of sub-bands | {20, 106, 133, 217, 273} | Randomized |
| DL TX power | {20, 40, 50, 80, 100} W | Randomized |
| UE antennas | {2, 4} | Randomized |
| Maximum TX rank | {2, 4} | As per UE ant. |
| Maximum DL TX | 5 | Fixed |
| UE traffic type | {FB, eMBB} | Randomized |
| Number FB UEs | {1, 5, 10} | Randomized |
| Number eMBB UEs | {0, 10, 25, 50, 100, 200, 300} | Randomized |
| Speed UE FB | {0.67, 10, 15, 30} m/s | Randomized |
| Speed UE eMBB | {0.67, 1.5, 3} m/s | Randomized |
| UE receiver types | {type0, type1, type2, type3} | Randomized |
| Indoor probability | {0.2, 0.4, 0.8} | Randomized |

Furthermore, we explore a preference space $\Omega = \Delta^1 \triangleq \{\boldsymbol{\omega} \mid \boldsymbol{\omega} = [\omega, 1-\omega]^\top, \omega \in [0,1]\}$ defined for the two-dimensional reward in (39). The preference space is partitioned into strata, and each actor is assigned to explore a different stratum. Preferences are then sampled from the corresponding strata following the procedure in Algorithm 6 (stratum construction) and Algorithm 7 (stratum-based sampling). Further details on all hyperparameters used in training are summarized in Appendix I.

### H.3 TESTING THE MORL LA CONTROLLER AGENT

#### H.3.1 SINGLE CONNECTIVITY SERVICE

We extend the analysis presented in Section 7.1 by further evaluating the MORL-based LA controller for a single connectivity service: video streaming users. Focusing on a single user class simplifies the analysis of the Pareto front achievable by the MORL controller agent.

Figure 9 shows the Pareto front defined by the two-dimensional reward function in Equation (39) for a 3-cell deployment with 10 streaming users. Each point on the frontier is obtained from 480 independent simulations, where each radio cell employs the MORL-based LA controller with a fixed preference value $\omega \in [0,1]$. The parameter $\omega$ determines the trade-off between minimizing radio resources required for packet transmission and maximizing the transmitted payload size. Moving along the frontier results in different performance trade-offs in network KPIs, as detailed in Figure 10.

At the top-right corner of Figure 9 and 10, large values of $\omega$ prioritize payload maximization (i.e., high TBS), but at the cost of excessive radio resource consumption. In this case, the controller agent selects overly aggressive MCS values relative to the channel state (see Figure 11a), targeting spectral efficiencies beyond channel capacity. This leads to frequent transmission failures (BLER $\approx 60\%$, see Figure 11b) and numerous retransmissions, yielding suboptimal throughput and spectral efficiency.

In contrast, when $\omega \approx 0$ (bottom-left corner of Figure 9 and 10), the controller favors conservative MCS choices (see Figure 11a), targeting spectral efficiencies well below channel capacity. Although this results in low resource utilization and highly reliable transmissions (BLER $\approx 0\%$, see Figure 11b), it under-utilizes favorable channel conditions by employing low modulation orders and code rates. Thus, the system fails to deliver higher payloads, limiting throughput and spectral efficiency.

Overall, throughput and spectral efficiency peak at $\omega \approx 0.34$ and $\omega \approx 0.5$, respectively. Beyond these values, throughput declines more rapidly than spectral efficiency due to the rising BLER, which

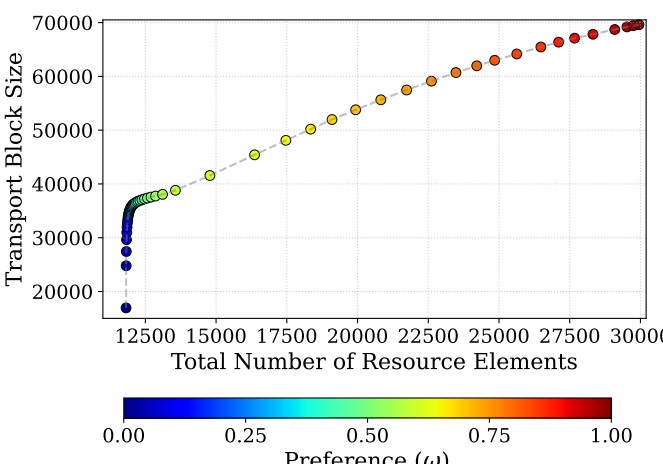

Figure 9: **Pareto front illustrating the trade-off between transport block size and resource utilization.** The Pareto front captures the relationship between the transport block size (vertical axis) and the total number of resource elements (horizontal axis) across a range of system configurations. Each point represents an outcome from $480$ independent simulations, computed using distinct preference vectors $\omega \in [0, 1]$, and is color-coded by the corresponding preference weight.

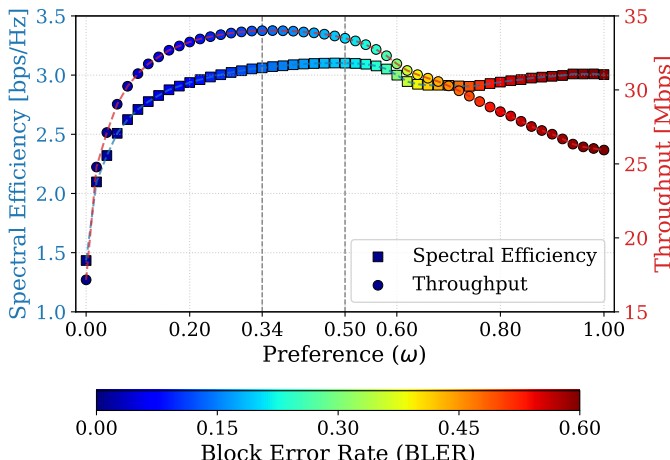

Figure 10: **Joint characterization of spectral efficiency and throughput under varying preference weights $\omega$, with BLER-encoded performance.** The figure presents the trade-off between spectral efficiency (squares, left axis) and throughput (circles, right axis) as a function of the preference weight $\omega$, which governs the optimization objective. Data points are color-coded based on BLER, with cooler hues indicating lower error rates. Dashed vertical lines denote peaks in the performance trends.

reduces transmission reliability. These results highlight the intrinsic tension between maximizing data rate and maintaining reliability in link adaptation, highlighting how a MORL controller agent can be deployed to provide differentiated connectivity services.

Figure 11 further illustrates how the controller policy changes by selecting different preference values. In particular, Figure 11a illustrate the action (i.e., MCS index) distribution induced by different preference values $\omega$. For example, it clearly highlights how small values of $\omega$ induce a link adaptation policy that selects overly conservative MCS index values relative to the channel state, thus aiming for transmissions with low spectral efficiency (i.e, characterized by low modulation order and code rate). Although this makes the transmission very robust, as demonstrated by the corresponding BLER distribution in Figure 11b, such policy leads to low data throughput.

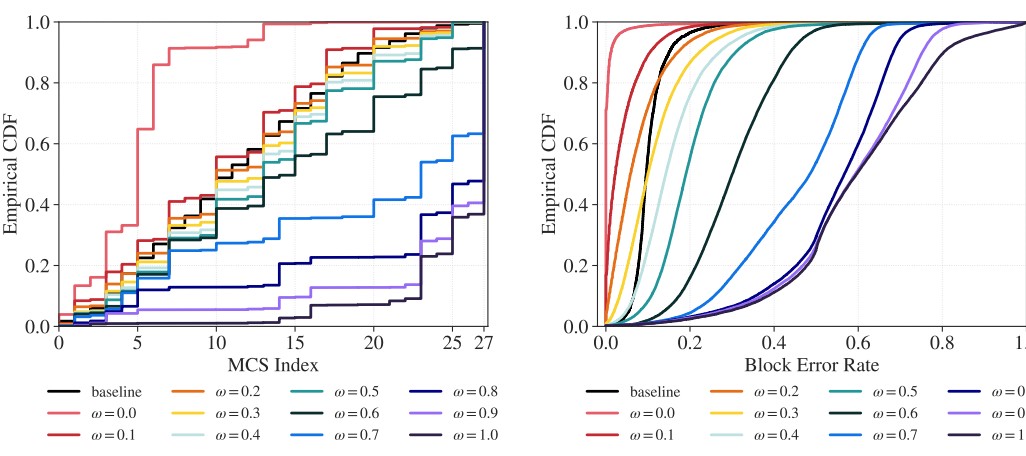

(a) Action (MCS index) distributions for various $\omega$.

(b) Block error rate distributions for various $\omega$.

Figure 11: Controller agent behavior for different preference values $\omega$.

Conversely, Figure 11a also shows how large preference values $\omega \approx 1$ induce a link adaptation policy that selects overly aggressive MCS index values relative to the channel state, thus aiming for transmissions with too high spectral efficiency (i.e, characterized by high modulation order and code rates). This makes the transmissions over-the-air unreliable, as demonstrated by the corresponding BLER distribution in Figure 11b.

### H.3.2 MULTI CONNECTIVITY SERVICES WITH QOS DIFFERENTIATION

We consider a practical scenario with two service applications with distinct QoS profiles concurrently sharing the resources of a radio cell: *real-time gaming* and *web browsing* users. Table 10 characterizes their QoS profile in terms of purpose, service type, differentiated services code point (DSCP) value, 5G QoS identifier (5QI) value and QoS features.

Table 10: QoS Profile Comparison: Real-time gaming vs Web Browsing

| Aspect | Real-time gaming | Web browsing |
|---|---|---|
| **Purpose** | Real-time, delay-sensitive traffic | Delay-tolerant, no bandwidth guarantees |
| **Service type** | Expedited forwarding (EF) | Best effort (BE) |
| **DSCP value** | EF (46) | BE (0) |
| **5QI** | 3 | 9 |
| **QoS features** | Guaranteed bit rate (GBR) Ultra-low latency Low jitter Packet delay budget (PDB) $\approx 50$ms packet error rate (PER) $\approx 1 \times 10^{-3}$ | Non-guaranteed bit rate (non-GBR) No strict latency No strict jitter PDB $\approx 300$ms |

**Real-time gaming** traffic consists of continuous, high-frequency bidirectional streams, often transmitted over user datagram protocol (UDP) based protocols to support real-time video rendering and user input feedback. This type of traffic demands substantially higher bitrates (ranging from 5 to 25 Mbps), ultra-low latency, and minimal jitter to maintain responsive and seamless game-play. As such, real-time gaming is classified as GBR traffic and expedited forwarding service, necessitating stringent QoS settings, including 5QI = 3 and DSCP values like EF (46), corresponding to a PDB of $\approx 50$ ms and PER $\approx 10^{-3}$ (cf. Table 5.7.4-1, 3GPP (2025b)).

**Web browsing** traffic is instead elastic, delay-tolerant, and bursty, following a request-response model (like the HyperText Transfer Protocol (HTTP)) over reliable transmission control protocol (TCP) connections. It generally demands low to moderate bitrates (typically below 1 Mbps) and is relatively

Table 11: RAN environment simulation parameters.

| Load scenario | Number of gaming users | | Web users arrival rate | | Performance KPIs |
|---|---|---|---|---|---|
| | Indoor | Outdoor | Indoor | Outdoor | |
| **Low** | 12 | 6 | 3.15 | 1.35 | Figure 12 |
| **Medium** | 24 | 12 | 6.3 | 2.7 | Figure 13 |
| **High** | 48 | 24 | 12.6 | 5.4 | Figure 14 |
| **Very high** | 72 | 36 | 18.9 | 8.1 | Figure 15 |

insensitive to latency and jitter, making it tolerant of network delays. Web browsing is typically classified as non-GBR traffic, associated with 5QI = 9 and DSCP values such as BE (0), corresponding to a PDB of $\approx 300$ ms and PER $\approx 10^{-6}$.

**The evaluation scenario** consists of a dense-urban deployment comprising three 3-sector sites operating at 3.5 GHz with a 100 MHz bandwidth with inter-site distance of 167 meters to ensure full uplink coverage across the simulation area. Traffic is predominantly downlink-oriented, with minimal uplink activity. Real-time gaming users remain active throughout the simulation duration, whereas web browsing users follow a Poisson arrival process with a distribution modeled to fit realistic field data patterns and depart the simulation upon completing their downloads (e.g., webpage, email, etc.).

Unlike the single-service application considered in Appendix H.3.1, the controller agent here applies a different preference vector to each service application: $\boldsymbol{\omega}_{\mathrm{g}} = [\omega_{\mathrm{g}}, 1 - \omega_{\mathrm{g}}]^T$ and $\boldsymbol{\omega}_{\mathrm{w}} = [\omega_{\mathrm{w}}, 1 - \omega_{\mathrm{w}}]^T$. Like before, we analyze how shifting the MORL controller policy along Pareto front defined by the two reward components in Equation (39), by tuning $\boldsymbol{\omega}_{\mathrm{g}}$ or $\boldsymbol{\omega}_{\mathrm{w}}$, produces different trade-offs in various performance KPIs. Under these settings, the values achievable for a performance KPI $g(\cdot)$ of each service application depends on both preference vectors, i.e., $g_g = g_g(\boldsymbol{\omega}_{\mathrm{g}}, \boldsymbol{\omega}_{\mathrm{w}})$ $g_w = g_w(\boldsymbol{\omega}_{\mathrm{g}}, \boldsymbol{\omega}_{\mathrm{w}})$.

Figure 12 to Figure 15 present the achievable user experience for real-time gaming and web browsing services in terms of three KPIs that closely relate to their QoS profile: user throughput, latency, and BLER. Each figure refers to one of the four traffic load scenarios, with a mixture of indoor and outdoor users, summarized in Table 11. Each figure also depicts the average MCS value selected by the MORL controller, showing how the controller agent applies a different policy to each service application for different combinations of preference vectors $\boldsymbol{\omega}_{\mathrm{g}}$ or $\boldsymbol{\omega}_{\mathrm{w}}$ and network load conditions.

For example, let us analyze the throughput distributions for real-time gaming uses (Figure 12a to Figure 15a) and web browsing user (Figure 12b to Figure 15b) for the various scenarios. For low and medium low load conditions, cf. Figure 12a-12b and Figure 13a-13b, respectively the mean throughput distribution of the two services shows similarities due to the abundance of radio resources compared to traffic load. The difference in mean throughput magnitude between the two type of services (i.e., Mbps vs Kbps) can be explained by the difference in traffic: continuous video streaming vs sporadic downloads of small packets.

At high and very high load conditions, cf. Figure 14a-14b and Figure 15a-15b, respectively, the two distributions of throughput start showing significant differences, clearly revealing how each service achieves the best mean throughput with different combinations of preference values $(\boldsymbol{\omega}_{\mathrm{g}}, \boldsymbol{\omega}_{\mathrm{w}})$. As the traffic load becomes very high, the region of preference values $(\boldsymbol{\omega}_{\mathrm{g}}, \boldsymbol{\omega}_{\mathrm{w}})$ that optimizes the throughput of each service shrinks into a smaller and well defined area. Furthermore, since in these scenarios more users share the same amount of radio resources, both services achieve lower throughput.

Similar trends can be observed for latency (expressed as round-trip time (RTT) for real-time gaming users and as webpage load time for web browsing users, respectively), and block error rate. The conditions observed in different service KPI in Figure 12 to Figure 15 can be related to the constraints $g_i(\boldsymbol{\omega}) \leq b_i$ that can be required to be fulfilled by a service intent in the optimization problem (1) solved by the optimizer agent to dynamically adapt the preference vectors for each service applications. For instance, in Section 7.3 we presented an example with a video streaming service requiring a minimum of 7 Mbps per active user (i.e., $g_{i,thr}(\boldsymbol{\omega}) \geq 7$). This threshold is rated as good for most real-time gaming applications at 720p and 1080p resolutions, and excellent for video streaming, given that typical requirements range from 5 Mbps for HD to 15 Mbps for 4K content.

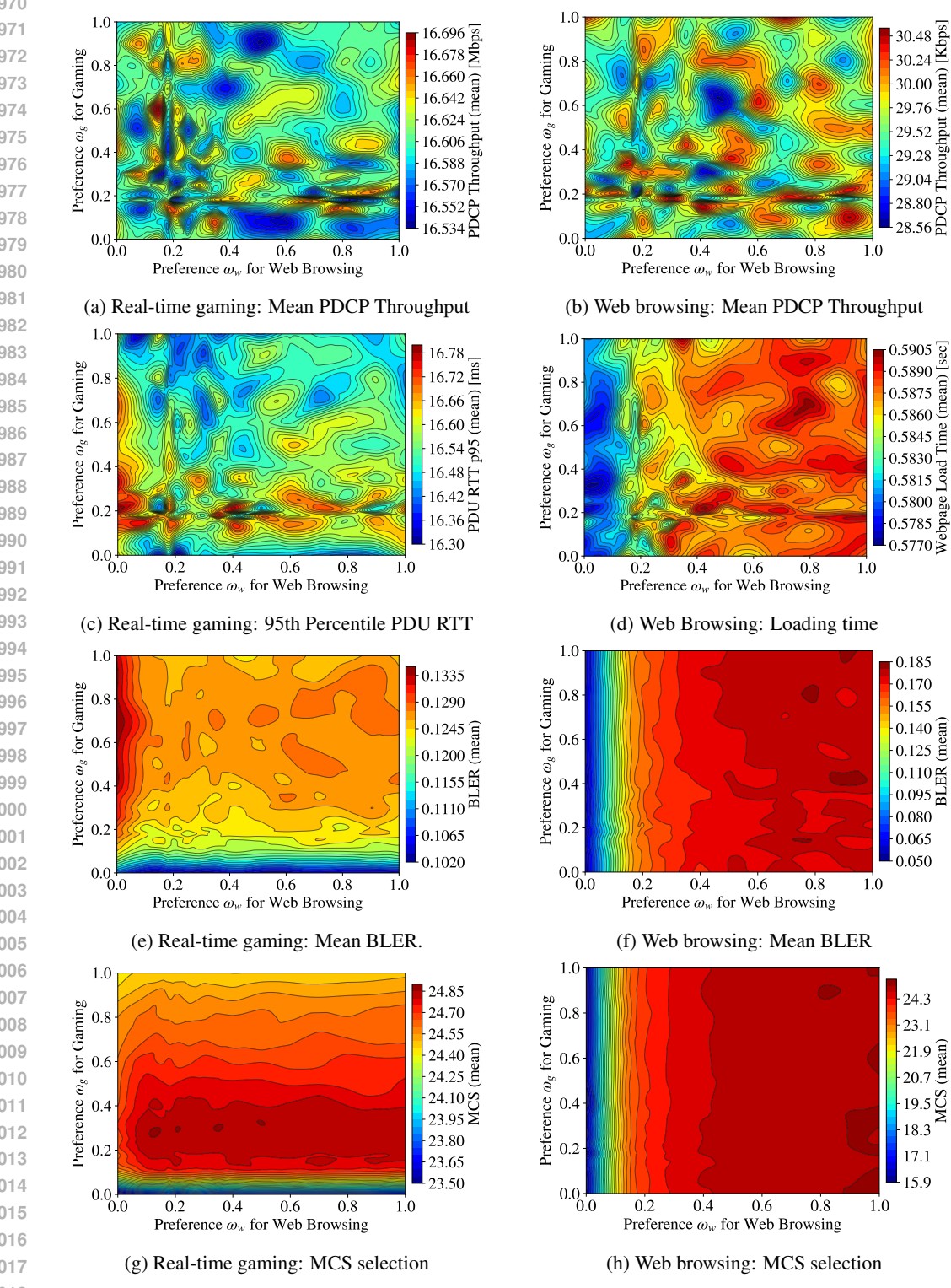

(a) Real-time gaming: Mean PDCP Throughput

(b) Web browsing: Mean PDCP Throughput

(c) Real-time gaming: 95th Percentile PDU RTT

(d) Web Browsing: Loading time

(e) Real-time gaming: Mean BLER.

(f) Web browsing: Mean BLER

(g) Real-time gaming: MCS selection

(h) Web browsing: MCS selection

Figure 12: **Impact of user preference weights on the performance of real-time gaming and browsing users in low network load conditions.** Each subplot shows a distinct QoS metric for real-time gaming users (left column) and for web browsing users (right column) under varying preference weights ($\omega_g$, $\omega_w$) reflecting resource allocation priorities for the two connectivity services. Metrics include: (a)-(b) mean user throughput, (c)-(d) mean latency (defined according to the service), (e)-(f) mean BLER. Furthermore, (g)-(h) show the action (mean MCS) distribution under ($\omega_g$, $\omega_w$).

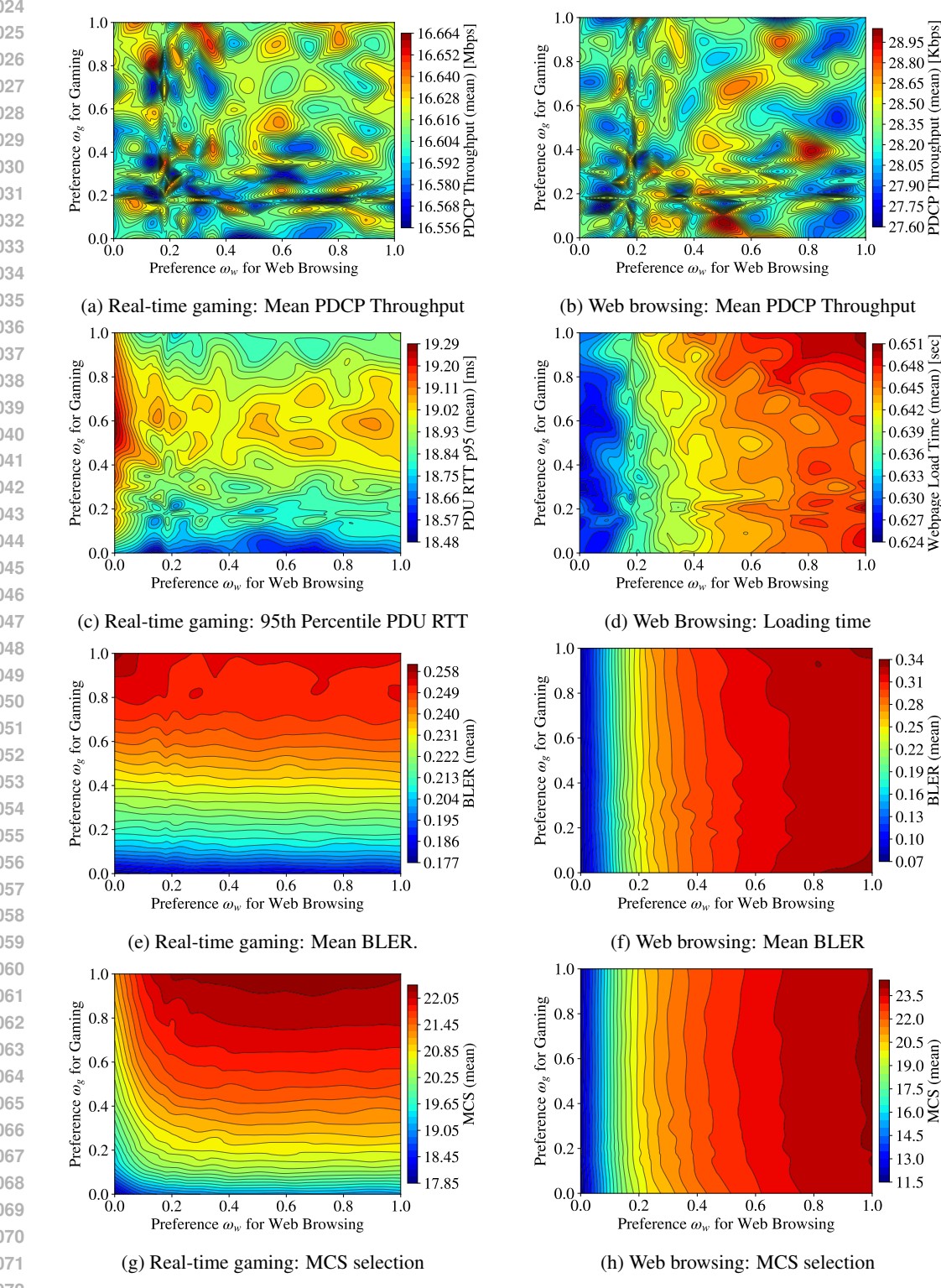

(a) Real-time gaming: Mean PDCP Throughput

(b) Web browsing: Mean PDCP Throughput

(c) Real-time gaming: 95th Percentile PDU RTT

(d) Web Browsing: Loading time

(e) Real-time gaming: Mean BLER.

(f) Web browsing: Mean BLER

(g) Real-time gaming: MCS selection

(h) Web browsing: MCS selection

Figure 13: **Impact of user preference weights on the performance of real-time gaming and browsing users in medium network load conditions.** Each subplot shows a distinct QoS metric for real-time gaming users (left column) and for web browsing users (right column) under varying preference weights $(\omega_g, \omega_w)$ reflecting resource allocation priorities for the two connectivity services. Metrics include: (a)-(b) mean user throughput, (c)-(d) mean latency (defined according to the service), (e)-(f) mean BLER. Furthermore, (g)-(h) show the action (mean MCS) distribution under $(\omega_g, \omega_w)$.

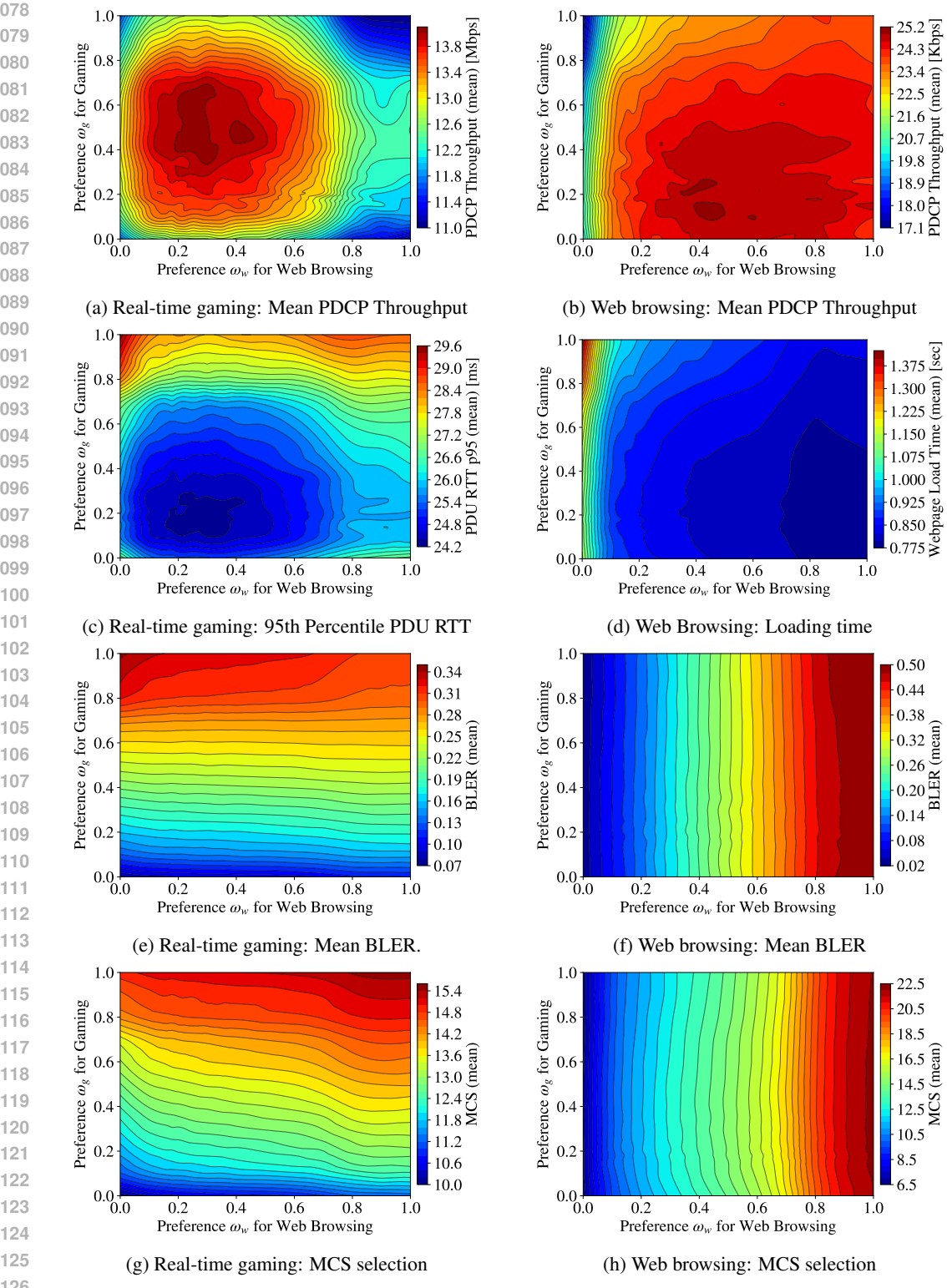

(a) Real-time gaming: Mean PDCP Throughput

(b) Web browsing: Mean PDCP Throughput

(c) Real-time gaming: 95th Percentile PDU RTT

(d) Web Browsing: Loading time

(e) Real-time gaming: Mean BLER.

(f) Web browsing: Mean BLER

(g) Real-time gaming: MCS selection

(h) Web browsing: MCS selection

Figure 14: **Impact of user preference weights on the performance of real-time gaming and browsing users in high network load conditions.** Each subplot shows a distinct QoS metric for real-time gaming users (left column) and for web browsing users (right column) under varying preference weights ($\omega_g$, $\omega_w$) reflecting resource allocation priorities for the two connectivity services. Metrics include: (a)-(b) mean user throughput, (c)-(d) mean latency (defined according to the service), (e)-(f) mean BLER. Furthermore, (g)-(h) show the action (mean MCS) distribution under ($\omega_g$, $\omega_w$).

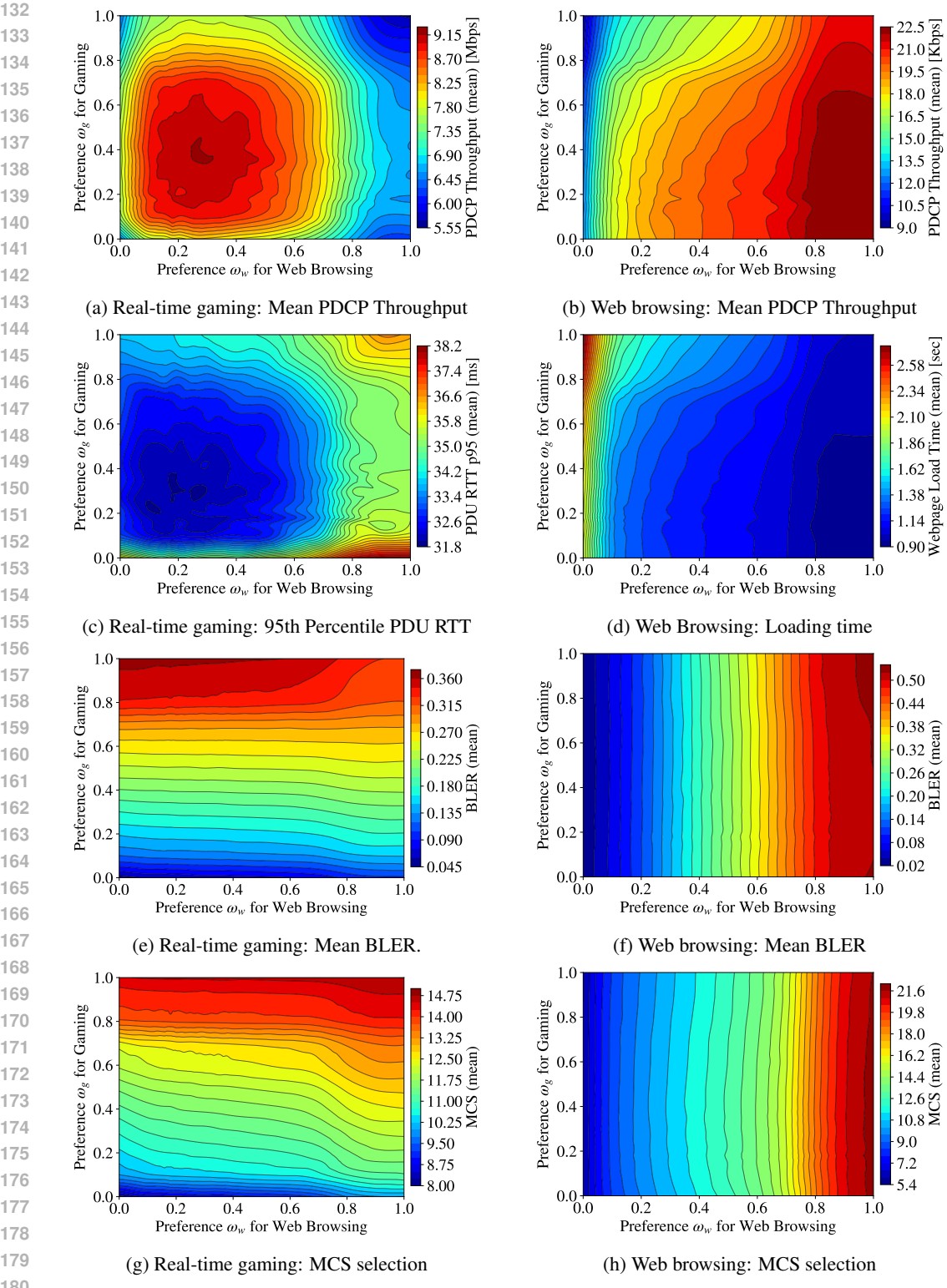

(a) Real-time gaming: Mean PDCP Throughput

(b) Web browsing: Mean PDCP Throughput

(c) Real-time gaming: 95th Percentile PDU RTT

(d) Web Browsing: Loading time

(e) Real-time gaming: Mean BLER.

(f) Web browsing: Mean BLER

(g) Real-time gaming: MCS selection

(h) Web browsing: MCS selection

Figure 15: **Impact of user preference weights on the performance of real-time gaming and browsing users in very high network load conditions.** Each subplot shows a distinct QoS metric for real-time gaming users (left column) and for web browsing users (right column) under varying preference weights $(\omega_g, \omega_w)$ reflecting resource allocation priorities for the two connectivity services. Metrics include: (a)-(b) mean user throughput, (c)-(d) mean latency (defined according to the service), (e)-(f) mean BLER. Furthermore, (g)-(h) show the action (mean MCS) distribution under $(\omega_g, \omega_w)$.

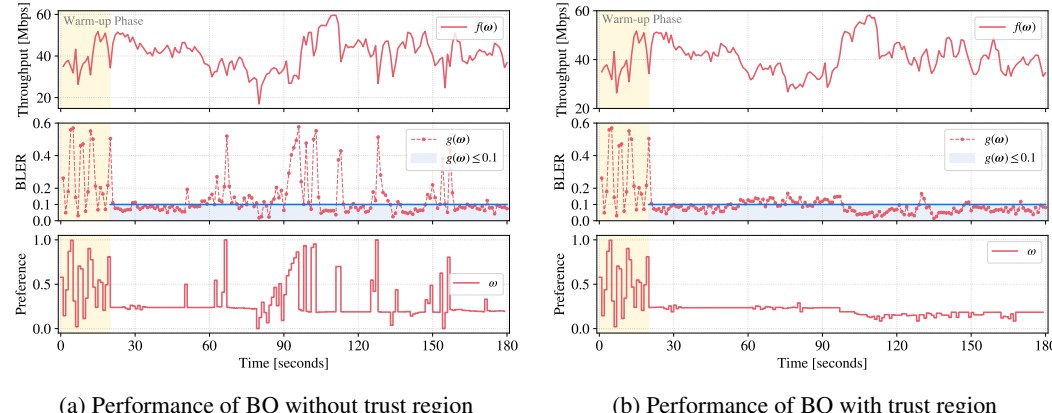

(a) Performance of BO without trust region          (b) Performance of BO with trust region

Figure 16: Performance comparison of optimizer agent when using the PAX-BO algorithm (a) without trust region and (b) with trust region enabled.

### H.4 ONLINE PREFERENCE OPTIMIZATION

Appendix D presented the Preference-Aligned eXploration Bayesian Optimization (PAX-BO) algorithm for the optimizer agent (cf.Algorithm 2). A key design feature of PAX-BO is the integration of trust regions to stabilize the selection of the preference values $\omega$ for the downstream controller agent.

Figure 16 compares the performance of PAX-BO considering an intent definition that requires to maximize the aggregate system throughput while keeping the BLER of each user below $10\%$ – which corresponds to the typical configuration of link adaptation in 4G/5G RAN systems. Figure 16a shows the performance of PAX-BO without trust region, whereas in Figure 16b we enabled the trust region. The top panel of both figures shows the aggregate system throughput $f(\omega)$, the middle panel shows BLER constraint evaluations $g(\omega)$ with threshold $g(\omega) \leq 0.1$, and the bottom panel shows the evolution of the preference parameter $\omega$. Compared to Figure 16a, the trust region stabilizes the optimization, reducing constraint violations and leading to smoother preference adaptation and improved system throughput (with smoother degradations).

# I COMPUTE RESOURCES AND HYPERPARAMETERS

All MORL training runs were performed on a high-performance computing (HPC) cluster. The main training node was equipped with an NVIDIA A100-PCIE-40GB GPU and 48 CPU cores, which hosted the learner, actors, and replay memory. The replay buffer was partitioned into four independently prioritized shards, each pinned to a dedicated CPU core to support parallelized access. Co-locating the learner, actors, and replay shards on the same node minimized intra-node communication latency.

We used 40 actors for each experiment, and each actor launched two threads that interacted with 14 simulator instances in parallel. The simulators were distributed across multiple compute nodes, totaling 560 CPU cores. Each simulator ran in a separate process and communicated with its assigned actor via ZeroMQ, enabling scalable multi-node environment interaction.

Cluster job scheduling and resource management were handled by the Load Sharing Facility (LSF), which managed job queueing, monitoring, and node allocation according to the experiments' resource specifications.

Table 12: Hyperparameters Used for Adaptor in Interpreter.

| Monitor | |
| --- | --- |
| Window size (W) | 12 |
| Alert-on ratio ($\rho_{\text{on}}$) | 0.55 |
| Alert-off ratio ($\rho_{\text{off}}$) | 0.45 |
| **Adjust** | |
| Step (Mbps) | 0.08 |
| Lifetime (Mbps) | 0.40 |
| Floor | 5.00 |
| Ceiling | 9.00 |
| Cooldown steps | 2 |
| Deadband | 0.05 |
| Gain (up) | 1.0 |
| Gain (down) | 1.0 |

Table 13: Hyperparameters Used for Supervised Fine-Tuning of the Intent-to-OTM Translator.

| Component | Setting |
| --- | --- |
| Base model | Qwen2.5-7B-Instruct |
| Parameter-efficient tuning | LoRA (rank 64, $\alpha = 16$, dropout $= 0.05$) |
| LoRA target modules | `q_proj, k_proj, v_proj, o_proj` |
| Precision | bfloat16 |
| Epochs | 2 |
| Batch size (per device) | 2 |
| Gradient accumulation steps | 8 |
| Optimizer | AdamW (Torch fused) |
| Learning rate | $2 \times 10^{-4}$ |
| Scheduler | Cosine decay |
| Warmup ratio | 0.03 |
| Weight decay | 0.01 |
| Gradient clipping | 1.0 |
| Gradient checkpointing | Enabled |
| Max sequence format | Qwen chat template (intent $\rightarrow$ OTM pair) |
| Evaluation frequency | Every 200 steps |
| Checkpoint frequency | Every 200 steps (max 5 checkpoints) |

Table 14: Bayesian Optimization Hyperparameters Used in the Actor.

| Hyperparameter | Value / Description |
|---|---|
| Acquisition function | qLogEI |
| MC samples for acquisition | 256 |
| Raw samples for optimization | 512 |
| Number of restarts | 10 |
| Batch size ($q$) | 1 |
| GP refit frequency | Every 1 observation |
| Training window size | 60 most recent samples |
| Input scaling (normalize) | Yes (Normalize transform) |
| Output scaling | Standardize outcomes |
| Trust region initial radius | 0.15 |
| Minimum trust region radius | 0.05 |
| Trust region shrink factor | 0.7 |
| Infeasible patience | 2 consecutive infeasible samples |
| No-improvement patience | 5 evaluations |
| Initial preference samples | 20 Sobol samples (fixed list) |
| Preference domain | $[0, 1]^2$ |

Table 15: Hyperparameters used for Multi-Objective Reinforcement Learning (MORL).

| Learner | |
|---|---|
| Optimizer | Adam Kingma & Ba (2017) |
| Learning rate | $5 \times 10^{-5}$ |
| $\beta_1$ (Adam momentum term) | 0.9 |
| $\beta_2$ (Adam second moment term) | 0.999 |
| $\epsilon$ (Adam numerical stability) | $1.5 \times 10^{-4}$ |
| Weight decay | $0.02/512$ |
| Gradient norm | 20 |
| Target update period | Every 1 gradient updates |
| Target update policy | Soft |
| Target update factor | $1.0 \times 10^{-3}$ |
| Model update interval | Every 200 gradient updates |
| Prefetched batches | 16 |
| Batch size (experience) | 512 |
| Batch size (preference) | 128 |
| Warm-up phase | 50,000 samples |
| Loss function | MSE |
| **Actor** | |
| Number of actors | 40 |
| Local buffer capacity | 2,500 |
| Discount factor ($\gamma$) | 1.0 |
| $\epsilon$-greedy (linear decay) | $0.8 \to 0.05$ |
| Timesteps | 5,500,000 |
| **Replay Memory** | |
| Number of shards | 4 |
| Capacity of each shard | 4,000,000 |
| Prioritization exponent ($\alpha$) | 0.6 |
| Importance sampling exponent ($\beta$) | 0.4 |
| **Model** | |
| Activation function | ReLU |
| Number of blocks | 6 |
| Number of layers per block | 2 |
| units per layer | 128 |
| Dropout probability | 0.1 |
| Layer normalization | True |

