# OpenReview forum: "From Intents to Actions: Agentic AI in Autonomous Networks"
_ICLR.cc/2026/Conference — Submitted to ICLR 2026_

### Official Review · Reviewer_35Gm · 2025-10-27

**Soundness:** 3
**Presentation:** 3
**Contribution:** 3
**Rating:** 4
**Confidence:** 4

**Summary:**

This paper introduces an Agentic AI system for autonomous networks, aimed at translating high-level network intents (such as low latency, high throughput, and energy efficiency optimization) into concrete control actions. The system consists of three main specialized agents: the Interpreter, the Optimizer, and the Controller. The Interpreter uses a large language model (LLM) to parse network intents and generate optimization templates; the Optimizer converts these templates into constrained optimization problems and adjusts preferences based on network conditions; the Controller utilizes multi-objective reinforcement learning (MORL) to implement adaptive policies that optimize network performance. The system has been validated through simulations in a 5G-compliant network, demonstrating its feasibility and effectiveness in real-world network conditions.

**Strengths:**

1. The Agentic AI system proposed in the paper is innovative in its application to autonomous network control, especially in transforming high-level intents into low-level control actions through the collaboration of multiple agents. The system is capable of handling complex multi-objective optimization problems and making adaptive decisions in dynamic environments.
2. The modular design of the system (including the Interpreter, Optimizer, and Controller) is well-suited for large-scale communication networks, efficiently addressing multiple objectives related to network performance and service quality. Each agent has a clear responsibility, and they achieve the goals through efficient collaboration mechanisms.
3. The paper validates the effectiveness of the Agentic AI system through high-fidelity 5G network simulations, providing rich experimental data that demonstrates the system's potential in real-world applications, particularly in dynamic and changing wireless environments.

**Weaknesses:**

1. Although the system is innovative, its complexity may result in high computational overhead, especially when dealing with large-scale network simulations. The interactions between each agent require processing a large amount of real-time data, which could place high demands on computational resources.
2. The paper mentions the current limitations of 4G/5G RAN hardware in terms of computational power, which poses challenges for the application of large language models (LLMs) due to insufficient hardware resources. While the authors propose using a dual-LLM architecture to mitigate this issue, this approach may still face feasibility problems in large-scale deployments, especially in low-latency and low-computation resource environments.
3. Although the paper demonstrates the effectiveness of the system, it lacks an in-depth comparison with existing autonomous network control methods (such as rule-based optimization or traditional reinforcement learning algorithms). Further comparative analysis could better assess the advantages and limitations of Agentic AI in handling complex intents.

**Questions:**

Please refer to the weaknesses.

---

> ### Author Response · Authors · 2025-11-21
> **Response Part 1**
>
> ***Comment 1: Although the system is innovative, its complexity may result in high computational overhead, especially when dealing with large-scale network simulations. The interactions between each agent require processing a large amount of real-time data, which could place high demands on computational resources.***
>
> **Response:** We thank the reviewer for highlighting this important concern. Computational overhead is central to our design considerations—as contributors from a major RAN vendor organization, our design must remain compatible with the computational and memory constraints available in RAN products.
>
> As elaborated in answers to other reviewers, besides using small-scale LLMs, a key principle of our design that enables us to significantly reduce computational demands is neat separation of timescales across the three agents. The controller’s operating timescale serves as the reference timescale for the entire agentic loop, while optimizer operates at a slower adaptation rhythm and the interpreter acts on an even slower supervisory timescale. We refer to the responses 3 and 4 to Reviewer 1 (hNgE) for an extended description.
>
> The case study considered in this paper is among the most challenging to assess the design feasibility, as the controller agent operates at a **sub-millisecond** timescale.
>
> Under the conditions of our case study:
> - **Controller (fast timescale)** uses a lightweight ResNet model that allows to perform inference in the typical latency bounds of link adaptation.
> - **Optimizer (intermediate timescale)** adjusts the controller’s policy at a much slower rate (e.g., once per second in our experiments). Because updates are infrequent, their computational footprint is manageable.
> - **Interpreter (slow timescale):** The interpreter—the only LLM-based agent—operates at even lower frequency (multi-seconds in our study), with inference triggered by persistent violations of the intent goals or requirement. This eliminates any latency pressure and allows the use of small LLMs that can fits comfortably within the RAN hardware capabilities.
>
> We revised Section 4.1 to clarify that the overall system introduces minimal overhead beyond existing RAN workflows.

---

> ### Author Response · Authors · 2025-11-21
> **Response Part 2**
>
> ***Comment 2: The paper mentions the current limitations of 4G/5G RAN hardware in terms of computational power, which poses challenges for the application of large language models (LLMs) due to insufficient hardware resources. While the authors propose using a dual-LLM architecture to mitigate this issue, this approach may still face feasibility problems in large-scale deployments, especially in low-latency and low-computation resource environments.***
>
> **Response:** We thank the reviewer for this insightful observation. While we partly addressed this concern in the response to the previous comment with respect to computational overhead, here we focus on the feasibility of this design with low-latency decision-making problems.
>
> The neat separation between the three agents timescales if what makes the design suitable for different applications in RAN systems. In this paper, we exemplified a case study representing the worst case scenario in terms of low-latency demands for decision-making at the controller agent.
>
> The controller agent is a More generally, as a drop-in replacement of a RAN control function, thus it inherits it native timescale. As long as the controller agent respects the inherited latency budget, loop stability and latency guarantees are preserved by design.
> - Our MORL link adaptation agent uses a lightweight ResNet dimensioned to perform inference within the typical latency bounds of link adaptation.
>
> The optimizer agent operates at a far slower rate (e.g., once per second in our study), to avoid interference with the faster control loop. As such, is less latency critical and poses minimal computational overhead relative to real-time RAN processes.
>
> The interpreter agent, as stated before, runs on an even slower timescale, placing very loose latency requirements on LLM inference. For a RAN management control function that operates at timescales of tens of minutes, he interpreter agent inference may occur at hourly timescales.
>
> Importantly, we do not deploy LLM inference frequently. The interpreter is invoked only:
>
> 1. **When an intent is first instantiated,** and
>
> 2. **When the intent must be refined,** which occurs infrequently (e.g., every few seconds at most in our case study, and often much slower for other RAN applications).
>
> This is why the proposed **dual-LLM architecture,** based on **small LLMs,** is viable even under today’s hardware limitations: LLM inference is infrequent, non-real-time, and completely decoupled from fast RAN control loops.
>
> Finally, while our design explicitly targets the constraints of current 4G/5G hardware, we also note that 6G platforms are already being planned for native AI workloads, making more computational resource available.
>
> We have revised the manuscript to clarify the importance of timescale separation (Section 3) for ensuring feasibility in low-latency decision-making, and how this helps reducing computational overhead of the overall design.

---

> ### Author Response · Authors · 2025-11-21
> **Response Part 3**
>
> ***Comment 3: Although the paper demonstrates the effectiveness of the system, it lacks an in-depth comparison with existing autonomous network control methods (such as rule-based optimization or traditional reinforcement learning algorithms). Further comparative analysis could better assess the advantages and limitations of Agentic AI in handling complex intents.***
>
> **Response:** We thank the reviewer for raising this important point. While we were unable to provide an extended comparison at the time of submitting this response, we are currently working to provide an extended set of results addressing the reviewer’s concern before the Dec 3rd deadline.
>
> We will concurrently revise the manuscript to explain why existing legacy network control methods—such as rule-based optimization—and traditional RL are fundamentally insufficient for intent-driven RAN operations.
>
> Rule-based controllers encode fixed logic and therefore cannot autonomously adjust their objective or behavior when service intents evolve. Intent-driven operation requires a controller whose optimization target can shift at runtime based on user requirements, traffic variations, or operator policies—capabilities that fixed rules cannot offer.
>
> Furthermore, in traditional RL, a single-policy RL model is trained for a single reward function. Handling multiple intents would therefore require
> - Training and storing multiple RL models, each optimized for a different KPI or KPI trade-off;
> - Swapping models at runtime whenever user requirements change (e.g., between reliability-oriented and throughput-oriented services).
>
> In our case study, transmissions occur every 0.5 ms and link adaptation has atypical time budget of a few tens of microseconds per scheduled users. Therefore, switching models and loading model weights within a few microseconds is not feasible in latency-critical RAN control.
>
> Our MORL controller design requires instead a single unified model trained against combination of rewards tradeoffs and diverse network conditions. Steering the MORL controller policy to achieve different tradeoffs of rewards, reflecting different intents objective, only requires feeding the proper set of preference values as input, without switching models on the fly.

---

### Official Review · Reviewer_u6JQ · 2025-10-28

**Soundness:** 2
**Presentation:** 2
**Contribution:** 2
**Rating:** 4
**Confidence:** 4

**Summary:**

The paper proposes an Agentic AI system that maps high‑level network intents to concrete RAN control actions through three cooperating agents: (i) an interpreter that turns natural‑language intents into an Optimization Template Model (OTM); (ii) an optimizer that converts OTMs into constrained optimization over a preference space and adapts preferences online using Bayesian optimization (PAX‑BO); and (iii) a controller that uses multi‑objective RL (MORL) to operate near the Pareto front under changing preferences. The system is explicitly two‑timescale: a slower intent‑management loop (interpreter↔optimizer) and a fast intent‑fulfillment loop (optimizer↔controller) suitable for sub‑millisecond RRM decisions.

**Strengths:**

1. The triadic agent design—with explicit separation of interpretation, preference planning, and control—fits the realities of RAN timescales.
2. Key technical elements include (a) a dual‑LLM interpreter for schema‑compliant intent parsing and constraint reasoning under tight hardware budgets; (b) PAX‑BO, a preference‑aligned constrained BO routine for steering the controller; and (c) D‑EQL, a distributed envelope Q‑learning algorithm that combines actor–learner decoupling, sharded prioritized replay, distributed exploration across the preference simplex, vector TD targets with envelope updates, and an auxiliary cosine‑stability loss.

**Weaknesses:**

1. D‑EQL essentially marries EQL with APE‑X/distributed PER and a cosine‑stability term; PAX‑BO employs standard GP‑BO with a trust region. These are well‑engineered combinations, but theoretical or algorithmic novelty appears limited. A stronger case would include ablations showing which D‑EQL elements (preference partitioning, hindsight priority refresh, cosine loss) are necessary for the RAN workload and why.
2. All results are from a single high‑fidelity simulator; there is little coverage of (i) different channel models/mobility patterns, (ii) multi‑cell interference/traffic mixes, or (iii) robustness to non‑stationarity. The workflow is persuasive, but broader stress tests are needed to establish generality.
3. The dual‑LLM interpreter is central, yet there is no quantitative assessment of parsing accuracy, OTM schema validity rates, or guard‑rail efficacy.
4. Reproducibility. Code/OTM schemas and simulator configurations are not (yet) available. Given the complexity of the stack, artifacts would materially increase impact.

**Questions:**

See my main concerns above.

---

> ### Author Response · Authors · 2025-11-20
> **Response Part 1**
>
> ***Comment 1: D EQL essentially marries EQL with APE X/distributed PER and a cosine stability term; PAX BO employs standard GP BO with a trust region. These are well engineered combinations, but theoretical or algorithmic novelty appears limited. A stronger case would include ablations showing which D EQL elements (preference partitioning, hindsight priority refresh, cosine loss) are necessary for the RAN workload and why.***
>
> **Response:** We thank the reviewer for the thoughtful assessment. The contribution of this paper is twofold: it is partly application-oriented; and partly theoretical and algorithmic work.
>
> We agree that D-EQL combines elements from EQL, Ape-X/distributed PER, and a cosine consistency term. However, this combination is novel in its integration and purpose, and directly addresses the core limitation of existing MORL methods: scalability of exploration in large state–action and preference spaces. Appendix D shows that D-EQL significantly improves over the state-of-the-art MORL baselines (Yang et al., 2019; Basaklar et al., 2023), in both coverage ratio and hypervolume, when the exploration space grows. Such scalability is instrumental to apply MORL approaches to RAN control functions.
>
> The key contribution of the paper, however, is an end-to-end framework that integrates a MORL-based controller with a supervisory LLM for setting and refining goals, and an optimizer for adapting over time the MORL preference values and align them towards the prescribed goal. Most work on MORL, instead, focuses primarily on the RL design itself, and statically evaluate the Pareto front achieved by the model across sweeping preference values.
>
> Finally, we agree with the reviewer that an ablation analysis is important to understand which parts of the D-EQL algorithm contribute the most. To this end, in Table 3 of appendix D.5.2, we adopted the approach of Basaklar et al., 2023, presenting performance improvements with respect to Yang et al., 2019; Basaklar et al., 2023 as each incrementally adds design features:
> - Basaklar et al., 2023 introduced preference partitioning, distributed but synchronous actors, and a new mechanism for hindsight priority the replay memory, with no update of priority weights and uniform sampling. This improved CFR1 by 12.3% over Yang et al., 2019
> - D-EQL introduces different preference partitioning, distributed asynchronous actors and a extends the mechanism for insight priority the replay memory, with updated priority weights and prioritized sampling. This improved CFR1 by 22.1% over Yang et al., 2019 and 8% over Basaklar et al., 2023.
>
> We have revised Section 5 to better highlight the novelty and strengths of D-EQL, **including a table that compares design aspects and performance of the three algorithms.**

---

> ### Author Response · Authors · 2025-11-20
> **Response Part 2**
>
> ***Comment 2: All results are from a single high fidelity simulator; there is little coverage of (i) different channel models/mobility patterns, (ii) multi cell interference/traffic mixes, or (iii) robustness to non stationarity. The workflow is persuasive, but broader stress tests are needed to establish generality.***
>
> **Response:** We thank the reviewer for raising this important point. We would like to clarify that our results are not evaluated under fixed network conditions – from training to testing. **Appendix F.2** describes how the training process of the MORL controller incorporates **extensive domain randomization,** covering a broad spectrum of:
> - channel models and fading conditions,
> - UE mobility patterns,
> - heterogeneous traffic types and load levels,
> - multi-cell interference settings (including different site layouts and transmission configurations), and
> - non-stationary dynamics inherent to realistic RAN environments.
>
> These characteristics generate diverse and dynamic training environments to ensure robustness and generality. Furthermore, **Appendix F.3 (Figures 9–15)** documents and extensive evaluation of the MORL controller under multiple network conditions and diverse traffic combinations.
>
> Due to space constraints in the main body, Section 7 only summarizes the primary case study, but the broader stress-testing—including variations in interference, mobility, traffic composition, and non-stationary dynamics—is fully documented in the appendix. We will further clarify this in the revised manuscript so the reader can easily locate these results.

---

> ### Author Response · Authors · 2025-11-20
> **Response Part 3**
>
> ***Comment 3: The dual LLM interpreter is central, yet there is no quantitative assessment of parsing accuracy, OTM schema validity rates, or guard rail efficacy.***
>
> **Response:** We thank the reviewer for this remark. We believe it is valid point, and we are currently working to provide a quantitative assessment of parsing accuracy, OTM schema validity rates with a revision that will be uploaded in the coming days.
>
> Regarding the guard-rail efficacy,
> - Figure 4 indicates that we are possibly using a rather conservative guardrail. We are currently working on extending these results with less stringent guard-rail.
> - Figures 16, on the other hand, shows the efficacy of the trust region used by te PAX-BO algorithm.

---

> ### Author Response · Authors · 2025-11-21
> **Response Part 4**
>
> ***Comment 4: Reproducibility. Code/OTM schemas and simulator configurations are not (yet) available. Given the complexity of the stack, artifacts would materially increase impact.***
>
> **Response:** We thank the reviewer for highlighting the importance of reproducibility and we apologize for being able to share only part of our work. As researchers with a major RAN vendor organization, our simulator integrates proprietary network models and near-product algorithmic components. Therefore, we are **not permitted to release the simulator or full codebase,** as these constitute core commercial assets of our organization.
>
> That said, we have made a concerted effort to maximize transparency within these constraints. The manuscript already provides:
> - Complete OTM schemas,
> - Detailed algorithmic descriptions of D-EQL and the overall agentic workflow,
> - A detailed description of the use case design, and
> - Extensive simulator configuration details in the appendices, enabling researchers to reconstruct equivalent environments using open-source tools.
>
> In the revision, we will further ensure that all necessary parameters and workflow steps are clearly documented to support independent re-implementation to the greatest extent possible.

---

> > ### Comment · Reviewer_u6JQ · 2025-11-25
> >
> > I appreciate that the authors provided detailed responses to my questions. After reading the reviews from other reviewers, I plan to maintain my current score at this time.

---

> > > ### Author Response · Authors · 2025-12-02
> > >
> > > We appreciate the reviewer’s follow-up note and we respect the reviewer’s decision. However, we respectfully observe that the reviewer’s comment does not engage with the detailed, point-by-point clarifications and experimental additions that we provided in direct response to the reviewer’s initial concerns.
> > >
> > > Given the depth of our revision and response to the reviewer’s original comments, we would have welcomed further technical discussion or feedback on whether the new evidence and analyses adequately resolved those concerns. Instead, the **decision to maintain the score based solely on reading the opinions of other reviewers**—without commentary on the merit of our specific responses—is rather unconventional and in breach with the rigor of the OpenReview process, making it difficult for us to understand which aspects of our revisions remain unaddressed.

---

### Official Review · Reviewer_EUHn · 2025-10-30

**Soundness:** 3
**Presentation:** 3
**Contribution:** 2
**Rating:** 4
**Confidence:** 3

**Summary:**

This paper introduces an autonomous framework for next-generation, large-scale, real-time distributed Radio Access Networks (RANs) by **incorporating large language models (LLMs) as a human-machine interface** for translating ambiguous high-level intent into executable optimization templates. A similar ideology has been applied in existing works from the past several years in areas such as autonomous driving, robotics, and many others. From my perspective, the primary contribution of this work is its application of this idea in the specific application of RANs with a particular focus on preference optimization and multi-objective control.

**Strengths:**

I particularly enjoy the organization of this paper. It comes with
1. a clear organization of motivations, problem settings, related concepts, and experiments; and
2. a detailed supplementary material which itself can be treated as a distinct technical report.

**Weaknesses:**

From my perspective, this is a typical **application-oriented work** that presents details about incorporating LLMs into a well-established large-scale RANs for human intent translation. In this case, it raises several concerns:
1. **Marginal novelty**. As stated earlier, the concept of using LLMs as a human-machine interface has been broadly explored and well implemented in many other areas, which shadows the novelty of this particular paper.
2. **Experiment Design**. From my understanding, the validations showcased in the experiment of Section 7 are primarily case-specific, meaning that for each specific use case, the authors measure achievements with respect to a specific intent. This raises concerns about whether the cases are carefully selected to sell the performance and whether the system is only responsive to these specific prompts.

**Questions:**

1. **Motivation**. Since the use of LLMs is bridging the gap between *intent in natural language* and *existing optimization template models (OTMs)*. I am not fully convinced of the necessity of incorporating LLMs in real-life operations, given that we already have these OTMs.
2. **Coverage**. Following the previous question, the system's capability is highly constrained by the diversity and generalizability of available OTMs. In this sense, it is not fully exploiting the common sense learned by the LLMs
3. **Benefits**. I am wondering about the cost efficiency of the new framework, given that we have an LLM in the system, which brings additional computational and memory costs. I would love to see the ratio of performance gain vs. computation/memory increase.

---

> ### Author Response · Authors · 2025-11-20
> **Response Part 1: Marginal Novelty**
>
> ***Comment: From my perspective, this is a typical application-oriented work that presents details about incorporating LLMs into a well-established large-scale RANs for human intent translation. In this case, it raises several concerns:
> 1.	Marginal novelty. As stated earlier, the concept of using LLMs as a human-machine interface has been broadly explored and well implemented in many other areas, which shadows the novelty of this particular paper.***
>
> **Response:** We thank the reviewer for the insightful comment. Our work lies at the intersection of being application-oriented while providing new theoretical contributions.
>
> While we agree that the use of LLMs as human–machine interfaces has been widely explored, the core contributions of our work extend well beyond this aspect and, in fact, lie in areas that have not been addressed in prior communication-network research.
>
> First, our paper is not a proposal to simply “incorporate LLMs into RAN systems.” Instead, we define the first end-to-end agentic architecture for intent-driven RAN operation, covering the full chain from:
>
> i.	intent parsing and decomposition,
>
> ii.	structured translation into optimization templates
>
> iii.	multi-timescale optimization and control, and
>
> iv.	Recursive intent refinement based on system feasibility.
>
> To the best of our knowledge, this is the first demonstration in the networking literature that closes the loop from natural-language/
> service-level intents all the way to RAN control actions, validated through system-level simulations with realistic mobility, scheduling, radio models, and resource constraints.
>
> Second, a major contribution—independent of LLM usage—is our distributed multi-objective reinforcement learning (MORL) framework developed in Appendix D. This design addresses the well-known scalability bottleneck of MORL training in large state–action spaces. Our results in Appendix D, clearly demonstrate our approach significantly improves over the state-of-the art MORL approaches, namely Yang et al. (2019) and Basaklar et al. (2023), in environment with even increasing exploration space size. We now transferred some of these results in Section 5.
>
> Finally, we highlight that our architecture explicitly resolves a key research challenge: enabling agentic/LLM-based intelligence in RAN without imposing latency-critical inference, thereby staying within the computational and timing constraints of contemporary RAN hardware.
>
> For these reasons, we believe the novelty and technical contributions of the paper extend meaningfully beyond prior work.

---

> ### Author Response · Authors · 2025-11-20
> **Response Part 2: Experiment Design**
>
> ***2.	Experiment Design. From my understanding, the validations showcased in the experiment of Section 7 are primarily case-specific, meaning that for each specific use case, the authors measure achievements with respect to a specific intent. This raises concerns about whether the cases are carefully selected to sell the performance and whether the system is only responsive to these specific prompts.***
>
> **Response:** We thank the reviewer for raising this important point. We acknowledge that the experiments in Section 7 focus on particular use cases; however, this choice reflects the need for a concrete, measurable, and latency-critical scenario to demonstrate the feasibility of the proposed architecture—not an attempt to sell the performance.
>
> Specifically, we purposely selected a study case—link adaptation—that provides very challenging conditions to test the triadic agentic system:
>
> - Fine-grained latency-constraints (µs–ms) for the controller agent, forcing a meaningful test of controller feasibility under a very fast timescale;
> - A clear test of recursive refinement when intent requirements are infeasible;
> - A realistic demonstration of differentiated connectivity services, a core 6G requirement.
>
> These properties make the scenario a strong proof of concept, not a cherry-picked example.
> The proposed system is fundamentally intent-agnostic and can be instantiated across a broad range of RAN functionalities. For instance, its application to slower control functionalities, such as those of the RAN management layer that reconfigure cells parameters for coverage or capacity trade-off, or network energy savings (that operate on timescale of hours), would have been easier picks for showing performance.
>
> Nonetheless, achievements always need to be measured with respect to intents that are meaningful to the case study at hand. For instance, in terms of energy savings gains for an intent that prescribes such objectives. Section 7 evaluates performance for intents that reflect meaningful differentiated connectivity services.
>
> We revised Section 6 and 7 to better highlight the complexity of the case study.

---

> ### Author Response · Authors · 2025-11-20
> **Response Part 3: Motivation**
>
> ***Comment 1: Motivation. Since the use of LLMs is bridging the gap between intent in natural language and existing optimization template models (OTMs). I am not fully convinced of the necessity of incorporating LLMs in real-life operations, given that we already have these OTMs.***
>
> **Response:** We thank the reviewer for this comment, as we realized that we had poorly described the LLM role(s) in our initial submission, giving the impression that is needed merely for lexical parsing.
>
> Before addressing that, we have to admit that we did not entirely understand the reviewer’s premise on “existing OTMs” and “given that we already have these OTMs”.  The OTM is a formalism that we introduced in this paper to systematically transform generic intents into machine-interpretable optimization structures that a downstream agent can act upon. We are unaware of similar applications to network management operations.
>
> More importantly, however, the role of the LLM-based extends far beyond the initial translation of an intent into an OTM. **Its primary contribution is cognitive, not lexical:** It enables the interpreter agent to reason over the optimizer feedback and network observations to evaluate intent feasibility, diagnose violations, and recursively refine the OTM when needed by relaxing over-tight requirements, proposing alternative trade-offs, or adapting objectives to network dynamics.
>
> We have entirely revised Section 4.1 to clarify this aspect and to better connect it with our choice of a dual-LLM architecture.

---

> ### Author Response · Authors · 2025-11-20
> **Response Part 4: Coverage**
>
> ***Comment 2: Coverage. Following the previous question, the system's capability is highly constrained by the diversity and generalizability of available OTMs. In this sense, it is not fully exploiting the common sense learned by the LLMs.***
>
> **Response:** We thank the reviewer for this insightful comment which indicates that our initial submission did not properly describe two important aspects of the design.
>
> Firstly, we would like to clarify that there is no fixed catalogue of “available OTMs” in our system. OTMs are not predefined templates. Instead, each OTM is constructed on the fly by the interpreter based on the given intent following a sufficiently general OTM schema—identifying objectives, constraints, requirements, metadata of an intent. The current OTM schema design, described Appendix A.3, is sufficiently general to scale the LLM interaction to multiple optimizer-controller pairs solving different network management problems. However, extensions can be considered.
>
> Secondly, following up on the previous response, the interpreter agent’s primary function is cognitive reasoning for recursive intent refinement, not simply mapping text to a template. This requires an LLM that retains common sense and domain awareness. To enable this, our architecture uses a dual-LLM design:
>
> 1.	**A fine-tuned LLM** with narrow scope that parses intents into syntactically correct OTM structures.
>
> 2.	**A more general-purpose, in-context learning (ICL), LLM – yest of small size –** that preserves its breadth of reasoning and common sense to:
>
> - reason over KPI statistics and violation alerts,
> - diagnose feasibility issues,
> - propose remedies (e.g., constraint relaxation or reformulation), and
> - recursively revise the OTM during closed-loop operation.
>
> We substantially revised Section 4.1 to clarify these aspects.

---

> ### Author Response · Authors · 2025-11-20
> **Response Part 5: Benefits**
>
> ***Comment 3: Benefits. I am wondering about the cost efficiency of the new framework, given that we have an LLM in the system, which brings additional computational and memory costs. I would love to see the ratio of performance gain vs. computation/memory increase.***
>
> **Response:** We appreciate the reviewer’s concern, and we fully agree that computational and memory overhead must be carefully justified. As researchers affiliated with a major RAN vendor organization, resource efficiency is a central design constraint, not an afterthought.
>
> While it is not simple to quantify a ratio of performance gain vs. computation/memory increase, our design is engineered to introduce minimal computational and memory overhead relative to existing RAN implementations. We address this aspect in the revised Section 4.1.
>
> Firstly, our dual-LLM architecture is deliberately based on lightweight models dimensioned to comply with our systems memory and computational capabilities. These models require orders of magnitude less computational and memory resources than frontier, large-scale LLMs.
>
> Secondly, the interpreter agent is the only component using LLM inference, and it operates on a slow, non–latency-critical timescale. In our case study, the interpreter issues updates only every few “simulated seconds”, as showed in Figure 4. For use cases with slower decision-making loops, where system dynamics evolve on timescales of minutes to hours (e.g., cell reconfiguration), the LLM-inference would very infrequent.
>
> Therefore, because LLM inference is infrequent and models are small, the computational and memory cost required by this approach is marginal and dimensioned to fit current RAN systems.
>
> Finally, we note that future 6G architectures are expected to be provisioned with native AI acceleration, possibly enabling larger models to be adopted more efficiently. Our proposal remains forward-compatible while being deployable with today’s constraints.

---

> ### Comment · Reviewer_hNgE · 2025-11-26
>
> I appreciate the detailed response from the authors. However, based on my understanding, SA5 focuses mainly on orchestration and management, rather than on the control aspects considered in this paper. SA2 is the main 3GPP standardization body for the core network and, to the best of my knowledge, there is currently no consensus on the use of an agentic system for the core network. On the contrary, there is a clear emphasis that UE–core network signaling is not required to be intent-based and that the core network is not required to deploy agentic solutions. Specifically, the decision of whether to use AI-based agentic solutions for the core network is left to operators and future studies.
>
> Here are some notes from the 3GPP document S2-2522308, which summarize the status of the different work tasks in SA2 (from Section 3.1 WT#3.1: AI for 6G architecture)
>
> 1.	enable the 6G CN to leverage AI capabilities and technologies in the 6G CN (e.g. AI agent), subject to operator policies and configuration and using 6G CN functionalities available in the network
>
> NOTE 3:	Whether to support intents in a 6G CN is up to operator choice. It is assumed it is not required for the MT stack of UE to produce nor understand the intent. The MT stack of UE is assumed to be agnostic to whether or not the network uses 6G CN AI capable entities (e.g. AI agent, AI-enabled NFs) to address UE requests not including intent.
>
> 6.	enable the operator to control the network's use of AI capabilities in its 6G CN, i.e. support different operator-configurable levels of autonomy based on operational needs, including the option to use or not use any AI capabilities.
>
> These points all indicate that whether 6G core networks will adopt an agentic architecture is still subject to study and debate. Therefore, the authors should provide evidences that using such an agentic architecture can bring gains (higher throughput, better QoS guarantees, lower carbon footprint, etc.) compared with more traditional approaches.

---

> ### Author Response · Authors · 2025-12-02
> **Response to Reviewer hNgE**
>
> We thank the reviewer for the thoughtful discussion. We now better understand that the remaining concern centers on the relevance of our work to the 6G core network. **To clarify, the paper does not claim applicability to the core network**. Rather, it explicitly targets network management and RAN control (e.g., via rApps), where we provide concrete evidence of the benefits of our agentic architecture—namely improved QoS and higher network throughput, demonstrated in Figures 3 and 4.
>
> More broadly, we believe that immediate applicability to 3GPP 6G standardization—although already evidenced through SA5 and RAN3–aligned use cases, as noted in our previous response—should not be viewed as a prerequisite for relevance in the ICLR research context, nor as grounds for rejection. The contribution stands on its technical merit and demonstrated effectiveness within its intended scope.
>
> ---
>
> In an abundance of clarity, we offer the following additional points:
>
> ### 1. General relevance
> The paper sits within a rapidly growing research area: agentic AI for RAN management, control, and optimization. Several works published during our submission and review period—e.g., [Pellejero (2025)](https://arxiv.org/pdf/2511.02532), [Bimo (2025)](https://arxiv.org/pdf/2507.14230), [Qayyum (2025)](https://ieeexplore.ieee.org/stamp/stamp.jsp?arnumber=11169757), [Xiong (2025)](https://www.arxiv.org/pdf/2508.13732), [Jolicoeur-Martineau (2025)](https://arxiv.org/abs/2510.04871)—demonstrate strong academic momentum in this direction. Like these references, ours is a research paper and not a direct contribution to 3GPP standardization. Nonetheless, similarly to the cited art, this paper can provide technical foundations that can inform future 6G standardization efforts in RAN management and optimization (e.g., in SA5 and RAN3).
>
> ### 2. Relevance to the 6G standardization
> We agree that the design proposed in the paper is not meant for core-network applications. However, the manuscript never claimed such applicability. Our focus is network management, optimization and control. As such, the work is framed within the RAN orchestration and management domain (e.g., SA5) and network function domain (i.e. RAN3). An application example may consist of the interpreter being embedded within an intent-management function of the RAN orchestration and management domain, while the optimizer-controller could be embedded in a network control rApp (e.g., configuring cells for coverage and capacity).
>
> In our first revision, we referenced a recent SA2 contribution only to clarify that agentic AI is indeed discussed within 3GPP, contrary to the reviewer’s earlier statement that it was not under consideration at all. This was not meant to suggest that SA2 has adopted this direction, nor is such a decision relevant to the contributions of this paper. As noted above, our work (along with cited prior art) is most pertinent to future directions in RAN-side management and optimization.
>
> ### 3. Relevance to RAN products.
> Finally, even beyond standardization, commercial RAN vendors routinely develop proprietary features that go well beyond 3GPP specifications to differentiate their products. Regardless of whether 3GPP adopts agentic-AI-based RAN management, the proposed approach remains directly relevant for proprietary implementations, where its performance and flexibility could provide competitive advantages.

---

### Official Review · Reviewer_hNgE · 2025-10-31

**Soundness:** 2
**Presentation:** 2
**Contribution:** 2
**Rating:** 2
**Confidence:** 4

**Summary:**

The paper investigates the use of AI agents for service provisioning in telecom operator networks. Specifically, it examines how high-level intents can be interpreted and executed by an agentic system. While the topic is timely, there is currently no consensus on adopting agentic systems in future telecom networks. Moreover, the paper remains at a high level—developing the core concept and providing only a few experimental examples. The experimental results are not convincing and therefore cannot serve as a proof of concept.

**Strengths:**

The idea is interesting, especially given the current momentum behind agentic systems.

**Weaknesses:**

The use of agentic systems—and, in particular, intent-based expressions of service requirements—has been discussed in 3GPP SA2. However, there is no consensus to adopt this approach in 6G systems. Even at this early stage, the indications suggest that intent-based service provisioning is not a requirement for 6G.

It is therefore necessary to provide stronger justification for why future telecom networks should be agentic and intent-based. What are the pros and cons of such a design? Is flexibility/programmability the only benefit, or can it also improve resource and energy efficiency at the network level? Note that agentic systems are not “free”: they require substantial computation to operate and may significantly increase operators’ energy consumption. What about decision-making latency and any control-loop stability considerations? Please include analysis (and, where possible, measurements) on these aspects.

Besides, the experiments show that the proposed approach works to some extent, but they do not demonstrate strict QoS guarantees. For example, in Figure 3(a), many users experience throughput below 7 Mbps. While the agents are able to detect requirement violations, detection does not imply recovery. Can the system adapt quickly and improve the situation? This is not shown in Figure 3. Please clarify whether detection triggers effective adaptation, quantify recovery times, and show post-mitigation performance.

**Questions:**

Please check the questions in the Weakness section above.

---

> ### Author Response · Authors · 2025-11-20
> **Response Part 1**
>
> ***Comment: The use of agentic systems—and, in particular, intent-based expressions of service requirements—has been discussed in 3GPP SA2. However, there is no consensus to adopt this approach in 6G systems. Even at this early stage, the indications suggest that intent-based service provisioning is not a requirement for 6G. It is therefore necessary to provide stronger justification for why future telecom networks should be agentic and intent-based.***
>
> **Response:** We appreciate the reviewer for the time, effort, and valuable feedback in evaluating our research. However, we respectfully disagree with the assertion that intent-based or agentic approaches lack relevance or traction for future 6G systems.
> As contributors to 3GPP and O-RAN standardization activities and members of a major RAN vendor organization, we can confirm with certainty that intent-driven network management—and AI/agentic AI as enablers—are already established priorities for 6G, as proved in the first meeting of the 6G cycle in October.  Hereafter, we articulate our response in two parts:
>
> **1. Intent-based management in 6G.**
>
> Intent-based management is already part of the existing 3GPP framework for 5GA (e.g., 3GPP TS 28.312), building on the TM Forum intent architecture (see, e.g., IG1251x, IG1230).
> The 3GPP SA5 workgroup is still actively working on intent management, with proposals to continue intent-driven operations into Release 20 and to extend it into 6G being supported by a broad cross-section of the industry. For example, the Rel-20 intent management study contribution [SP-250861](https://www.3gpp.org/ftp/tsg_sa/TSG_SA/TSGS_108_Prague_2025-06/Docs/SP-250861.zip) from last June is backed by 21 major companies—including Huawei, Ericsson, Nokia, Samsung, AT&T, Verizon, CMCC, China Telecom, Orange, Deutsche Telekom, NTT DOCOMO—reflecting strong industry alignment. Discussions to agree such study item are ongoing this week at the SA5#164 meeting.
>
> **2. The potential role Agentic AI in intent-driven operations is widely recognized.**
>
> Both 3GPP and O-RAN explicitly position AI, and increasingly agentic AI, as key to realizing intent-based automation. For example, the 3GPP public technology outlook on “AI/ML for NG-RAN & 5G-Advanced towards 6G” highlights:
> “How to integrate developing AI technologies, such as Agentic AI, with 6G networks is a promising topic to be investigated when 3GPP starts the discussion on the overall 6G RAN architecture.”
>
> More specifically, the SA5 workshop held last June identified AI agent/Agentic AI for intent-based network management as the key areas for the 6G management architecture. The [workshop summary](https://www.3gpp.org/ftp/Email_Discussions/SA5/OAM%20rapporteur%20calls/Rapporteur%20call%20%23161/SA5_NWM_Discussion_for_Rel-20_6G_OAM_Work_Areas-v0.0.7.pdf), as well as several contributions to it [available here](https://www.3gpp.org/ftp/Email_Discussions/SA5/SA5-level%20discussions/SA5_Workshop_on_6G_Rel20), report strong support for:
>
> *“WT-5 Study the adoption of agentic autonomous management in 6G management architecture towards Autonomous Networks, including agent discovery, agent management and orchestration, multi-agent collaboration and interactions, enabling agent to utilize/access 6G management provisions.”*
>
> As a result, this topic is among the objectives of a new SA5 study proposal on 6G Management and Orchestration (see, e.g., [S5-255123](https://www.3gpp.org/ftp/TSG_SA/WG5_TM/TSGS5_164/Docs/S5-255123.zip)). Concurrently, RAN3 contributors have proposed Agentic AI a new 6G use case on Intent-driven collaborative tasks (see, e.g. [R3-256538](https://www.3gpp.org/ftp/TSG_RAN/WG3_Iu/TSGR3_129-bis/Docs/R3-256538.zip)), while SA2 is discussing an agentic 6G core network (see, e.g., [S2-2507223](https://www.3gpp.org/ftp/tsg_sa/WG2_Arch/TSGS2_170_Goteborg_2025-08/Docs/S2-2507223.zip)).
>
> For these reasons, we maintain that the topic addressed in our paper is not only relevant but timely and aligned with ongoing standardization and industry priorities for 6G. Following the reviewer’s remark, we revised Section 2 of the paper with a new paragraph motivating why agentic and intent-based operation is not an isolated academic proposal but a well-established direction in ongoing academic and industry research.

---

> ### Author Response · Authors · 2025-11-20
> **Response Part 2**
>
> ***Comment: What are the pros and cons of such a design? Is flexibility/programmability the only benefit, or can it also improve resource and energy efficiency at the network level?***
>
> **Response:** We thank the reviewer for the valuable comment. While we present a single case study, the system proposed in this paper serves as a blueprint for the integration of agentic AI across different domains of the RAN architecture for intent-driven network management.
>
> Our case study embeds agentic AI within the RAN network function domain (i.e., the domain represented by RAN access nodes), where we consider the interaction of an LLM-based interpreter with the real-time RAN control functionality of the data layer (i.e., link adaptation). However, the same triadic design principle can also be embedded within the RAN management domain to steer the behavior of higher-layer management functions—such as optimizing coverage and capacity trade-offs or improving network energy efficiency of a certain area (e.g., by switching on/off radio cells or frequency layers). The key design enabler for this versatility is the separation of timescales among the agents, as we elaborate in our responses to subsequent comments.
>
> Based on the reviewer’s feedback, we revised Section 3 to introduce a dedicated discussion highlighting the effects of timescale separation, including how this enables the deployment of this system for managing other functionalities in different RAN network domains.

---

> ### Author Response · Authors · 2025-11-20
> **Response Part 3**
>
> ***Comment: Note that agentic systems are not “free”: they require substantial computation to operate and may significantly increase operators’ energy consumption.***
>
> **Response:** We agree with the reviewer that “agentic systems are not free”, and we regret that our original submission did not properly elaborate on why our design does not require substantial computation or significantly increase operators’ energy consumption.
>
> Two design aspects make this approach computationally and energy efficient:
>
> 1.	The interpreter agent is the only LLM-based agent, and our dual-LLM architecture allows us to use small-scale LLMs dimensioned to the RAN hardware capacity. The optimizer and controller agents remain lightweight, which limits the computational overhead.
>
> 2.	The interpreter agent operates on a slow supervisory timescale—much slower than that of the optimizer and controller agents. This makes LLM inference infrequent. Our case study exemplifies the worst-case scenario, wherein the controller agent replaces a radio resource management function that operates at a sub-millisecond timescale. Under this condition, the LLM-based interpreter agent may act once every few seconds or at a slower cadence when intent violations are reported. When the controller agent replaces network reconfiguration functions that operate on a minute-to-hour timescale, the interpreter agent would react on an even slower cadence.
>
> Therefore, the incremental computational and energy cost of running small LLMs at a slow operational timescale is affordable compared to the cost of continuously executing computationally intense operations—like resource scheduling and beamforming—on a sub-millisecond timescale for a large amount of users.
>
> Based on the reviewer’s feedback, we revised Section 4.1 to clarify why our design does not require substantial computation nor significantly increase operators’ energy consumption compared to other operations of the RAN system.

---

> ### Author Response · Authors · 2025-11-20
> **Response Part 4**
>
> ***Comment: What about decision-making latency and any control-loop stability considerations? Please include analysis (and, where possible, measurements) on these aspects.***
>
> **Response:** We thank the reviewer for raising this important point. We have revised Section 3 to clarify how decision-making latency and control-loop stability are addressed in our design through a clear separation of timescales among the three agents.
> Here, we discuss this in the context of the case study presented in the paper. The controller establishes the reference timescale; the optimizer operates at a slower adaptation rhythm; and the interpreter acts on an even slower supervisory timescale.
>
> **1.	Controller (decision-making – reference timescale):**
> The controller replaces an existing RAN control function and therefore inherits its native latency budget. In our case study, the LA MORL controller is realized with a relatively small MLP that executes within a few microseconds (10 and 20 microseconds)—fully compatible with the real-world time budgets of link adaptation. More generally, as a drop-in replacement, as long as the controller agent respects the inherited latency budget, loop stability and latency guarantees are preserved by design.
>
> **2.	Optimizer (adaptive – medium timescale):**
> The optimizer adapts the controller policy at a slower timescale—in our case study, once per second. The optimizer timescale is dimensioned to ensure that optimization dynamics do not interfere with the fast(er) downstream decision-making loop.
>
> **3.	Interpreter (supervisory – slow timescale):**
> The interpreter refines the intent formulation at a slower supervisory timescale (seconds to minutes in our case study), far removed from real-time control. As discussed earlier, this also contributes to reducing computational demands and energy costs.
>
> This multi-timescale decomposition is not specific to our link adaptation case study but extends to any RAN function that would be involved in this triadic system, ensuring that stability and latency bounds are preserved regardless of which control function the agentic system is applied to.

---

> ### Author Response · Authors · 2025-11-20
> **Response Part 5**
>
> ***Comment: Besides, the experiments show that the proposed approach works to some extent, but they do not demonstrate strict QoS guarantees. For example, in Figure 3(a), many users experience throughput below 7 Mbps. While the agents are able to detect requirement violations, detection does not imply recovery.***
>
> **Response:** We appreciate the reviewer’s comment and have clarified these aspects in the revised manuscript.
>
> First, we note that the reviewer’s observation refers to **Figure 3(a),** which corresponds to the case with strict QoS guarantees **without intent relaxation.** In that experiment, the minimum-rate requirement is held fixed—**even when it becomes infeasible.** This was intentional to illustrate the behavior of the optimizer-controller loop when strict constraints cannot be satisfied.
>
> Our simulation environment models high-fidelity radio conditions, including realistic channels, link adaptation, scheduling, mobility, and decoding. Under such conditions, QoS guarantees are inherently constrained by physics and available resources. When strict per-user constraints are imposed—such as a 7 Mbps minimum rate—these constraints can become infeasible for cell-edge users whose spectral efficiency is too low to support the target, regardless of the controller policy. Therefore, in **Figure 3(a),** when the underlying problem is infeasible (due to poor channel quality or insufficient radio resources), constraint violations will persist. This is not a limitation of the agentic framework—it reflects the nature of wireless systems.
>
> To address this, **Figure 4** evaluates the full agentic loop, where the **interpreter agent is allowed to revise the intent** when persistent infeasibility is detected. The interpreter progressively relaxes the target rate to search for a feasible specification. Nonetheless, even in this case, feasibility cannot always be guaranteed (e.g., extreme cell-edge conditions), but the interpreter enables the system to:
>
> - Avoid wasting large amounts of radio resources on impossible constraints,
> - Redirect resources to users who can meet service requirements, and
> - Maintain statistical QoS performance at the system level.
>
> Our results demonstrate that the proposed agentic AI system identifies infeasibility, adapts the intent when permitted, and improves system-level efficiency—while respecting the fundamental limits of wireless communication.

---

> ### Author Response · Authors · 2025-11-20
> **Response Part 6**
>
> ***Comment: Can the system adapt quickly and improve the situation? This is not shown in Figure 3. Please clarify whether detection triggers effective adaptation, quantify recovery times, and show post-mitigation performance.***
>
> **Response:** We thank the reviewer for raising this point. The system’s ability to detect violations and subsequently improve performance is demonstrated in **Figure 4,** which shows the complete adaptation loop (interpreter → optimizer → controller) when the interpreter can apply cognitive reasoning to refine the intent formulation. As stated in an earlier response, the chosen case study represents the worst condition in terms of decision-making timescales for the controller agent.
>
> In this case, our design allows us to detect intent violations or identify infeasible requirements due to changing network conditions with a 1 second timescale (although it could be reduced to a few hundreds of ms). The subsequence intent refinement and control policy adaptation require approximately another second. Therefore, the control agent acts with a new policy within two seconds. For typical connectivity services, which define KPIs in statistical sense, such timescale is reasonably fast, and the user would not experience a perceivable service degradation.
>
> We revised Section 6 to better explain these dynamics.

---

### Author Response · Authors · 2025-11-20
**General Response**

We sincerely thank all reviewers for their thorough evaluation of our manuscript and for the insightful comments and constructive suggestions. The feedback significantly improved the clarity, rigor, and completeness of our work. Below, we summarize how we addressed the main concerns raised by the reviews.

**1. Relevance of intent-driven network management and Agentic AI in 6G**

The revised manuscript strengthens and documents the motivation for this work, clarifying that intent-based network management is already supported in modern 5G-Advanced systems. Furthermore, there is strong alignment between academia and industry indicating that agentic AI is emerging as a key enabler for several 6G functionalities.
Specifically, we clarify that 3GPP SA5 is currently discussing a new study proposal on 6G Management and Orchestration (see, e.g., [S5-255123](https://www.3gpp.org/ftp/TSG_SA/WG5_TM/TSGS5_164/Docs/S5-255123.zip)) to investigate the adoption of agentic autonomous management within the 6G management architecture, while SA2 is discussing an agentic 6G core network (see, e.g., [S2-2507223](https://www.3gpp.org/ftp/tsg_sa/WG2_Arch/TSGS2_170_Goteborg_2025-08/Docs/S2-2507223.zip)).

**2. Clarification of the Agentic architecture and timescales**

We significantly revised Sections 3 and 4.1 to clarify the roles of the interpreter, optimizer, and controller agents and associated timescales. We now more explicitly state that the controller defines the reference timescale for the three agents, while the optimizer operates on a slower (adaptation) timescale, and the interpreter on an even slower (supervisory) timescale.

**3. Computational/memory overhead and low-latency feasibility**

Based on the timescale separation just described, we revised Section 4.1 to clarify that:
- The controller is the agent who inherits latency bounds from the RAN function it replaces. Such latency bound can be very stringent in some network operations, and not in others.
- The optimizer and interpreter are each progressively less latency-critical.
Furthermore, we clarify that the low inference frequency requirements of the LLM-based interpreter agent, together with the use of a small-scale LLM in our design, ensure that:
- The overall design induces minimal computational and energy cost overhead.
- The memory footprint of the design is reasonable, even for current 5G architectures.

**4. Role of the LLM and OTMs**

We clarified in Section 4.1 that the primary role of LLM is cognitive, not only lexical parsing, leveraging a small-scale LLM with general knowledge for closed-loop reasoning over KPI deviations, intent-fulfilment feedback, and evolving network conditions. This enables the autonomous refinement of optimization objectives in response to the changing network conditions. Furthermore, we clarify that the interpreter agent does not rely on a fixed catalogue of OTMs.

**5. Comparative analysis with existing approaches**

We are currently working to expand the experiments in both Figure 3 and Figure 4 to compare our architecture with the 5G legacy rule-based baseline and classical RL approaches before the December 3rd deadline. Furthermore, we will include an analysis of the parsing accuracy and OTM schema validity rates for LLM-based interpreter.

**6. Reproducibility**

As researchers affiliated with a major RAN vendor organization, we regret that our simulator code cannot be released due to industrial IP constraints. However, the paper includes extensive appendices with full OTM schemas, algorithmic details, and configuration parameters to maximize transparency.

---

### Author Response · Authors · 2025-12-02
**Additional Revision and Analysis Extension**

Following the reviewers’ recommendations, we have further revised the manuscript to address every remaining concern and significantly extended our analysis in three key directions.

### 1. Quantitative assessment of parsing accuracy and OTM schema validity rates
We revised Section 4.1 and added Appendix C.1 to assess intent parsing accuracy and OTM schema validity rates. We Compare an intent translator module realized with (a) a fine-tuned Qwen2.5-Instrctuct model and (b) an off-the-shelf general-purpose Qwen2.5-Instrctuct model. The results indicate that the few-shot fine tuning methodology described in Appendix C provides significant higher intent accuracy and schema validity rate.

### 2. D-EQL algorithm analysis
We revised Section 5.1 to better highlight the technical novelty and scalability of the D-EQL algorithm compared to the de facto established MORL designs Yang et al. (2019) and Basaklar et al. (2023). Our analysis demonstrates that D-EQL outperforms these methods algorithms already in environments with marginally large exploration spaces, both in FR1 score and hypervolume.

### 3. Comparison with existing autonomous network control methods
We significantly extended Section 7 by benchmarking our approach against (a) the state-of-the art link adaptation (LA) used in 5G/5G-A systems and (b) the RL-based LA design of Demirel et al. (2025). We show that our agentic system outperforms both benchmarks in a complex scenario requiring the system to support multiple connectivity services with diverse (conflicting) goals.

Furthermore, we stress that traditional RL cannot adapt policies across service goals in real-time.
- Our approach leverages a **single D-EQL model** trained across multi-reward tradeoffs. At inference, the goals of each connectivity service are met by adapting the D-EQL policy on the fly through a dedicated set of input preference values reflecting the service goals.
- On the contrary, **traditional RL would require multiple models**, each trained with a reward tailored to the specific goal of an individual service. Besides the evident scalability ussie of training one model for every connectivity service, this approach would also require switching models within the microsecond-level budget of LA (and most RRM functions in general), which is impractical.

---

### Meta-Review · Area_Chair_ajhY · 2025-12-03

**Summary:**

Based on the reviewers’ feedback and my own reading of the paper, the overall quality still needs improvement. We regret to inform you that this paper has not been accepted for this year’s conference. We hope the authors can address the relevant issues in subsequent revisions and achieve acceptance in future submissions.

**Reviewer Concerns:**

The revised manuscript strengthens and documents the motivation for this work, clarifying that intent-based network management is already supported in modern 5G-Advanced systems. However, the authors should also provide evidence that using such an agentic architecture can bring gains, since whether 6G core networks will adopt an agentic architecture is still subject to study and debate.

**Reviewer Scores:**

If the authors can provide evidence that using such an agentic architecture can bring gains (higher throughput, better QoS guarantees, lower carbon footprint, etc.)  for 6G core networks, the reviewer would have changed their score.

---

### Decision · Program_Chairs · 2026-01-26

Reject